# Transformers learn to implement preconditioned gradient descent for in-context learning

**Kwangjun Ahn***
MIT EECS/LIDS
kjahn@mit.edu

**Xiang Cheng***
MIT LIDS
chengx@mit.edu

**Hadi Daneshmand***
MIT LIDS/FODSI
hdanesh@mit.edu

**Suvrit Sra**
TU Munich / MIT
suvrit@mit.edu

## Abstract

Several recent works demonstrate that transformers can implement algorithms like gradient descent. By a careful construction of weights, these works show that multiple layers of transformers are expressive enough to simulate iterations of gradient descent. Going beyond the question of expressivity, we ask: *Can transformers learn to implement such algorithms by training over random problem instances?* To our knowledge, we make the first theoretical progress on this question via an analysis of the loss landscape for linear transformers trained over random instances of linear regression. For a single attention layer, we prove the global minimum of the training objective implements a single iteration of preconditioned gradient descent. Notably, the preconditioning matrix not only adapts to the input distribution but also to the variance induced by data inadequacy. For a transformer with $L$ attention layers, we prove certain critical points of the training objective implement $L$ iterations of preconditioned gradient descent. Our results call for future theoretical studies on learning algorithms by training transformers.

## 1 Introduction

In-context learning (ICL) is the striking capability of large language models: Given a prompt containing examples and a query, the transformer produces the correct output based on the context provided by the examples, *without adapting its parameters* (Brown et al., 2020; Lieber et al., 2021; Rae et al., 2021; Black et al., 2022). This property has become the focus of body of recent research that aims to shed light on the underlying mechanism of large language models (Garg et al., 2022; Akyürek et al., 2022; von Oswald et al., 2023; Li and Malik, 2017; Min et al., 2021; Xie et al., 2021; Elhage et al., 2021; Olsson et al., 2022).

A line of research studies ICL via the expressive power of transformers. Transformer architectures are powerful Turing machines, capable of implementing various algorithms (Pérez et al., 2021; Wei et al., 2022). Given an in-context prompt, Edelman et al. (2022); Olsson et al. (2022) argue that transformers are able to implement algorithms through the recurrence of multi-head attentions to extract coarse information from raw input prompts. Akyürek et al. (2022); von Oswald et al. (2023) assert that transformers can implement gradient descent on linear regression encoded in a given input prompt. It is thought provoking that transformers can implement such algorithms.

Although transformers are universal machines to implement algorithms, they need specific parameter configurations for achieving these implementations. In practice, their parameters are adjusted via training using non-convex optimization over random problem instances. Hence, it remains unclear whether this non-convex optimization can be used to learn algorithms. The present paper investigates *the possibility of learning algorithms via training over random problem instances.*

---

*Equal contribution, alphabetical order.

37th Conference on Neural Information Processing Systems (NeurIPS 2023).

More specifically, we investigate the learning of gradient-based methods. It is hard to mathematically formulate what it means to learn gradient descent for general functions with transformers. Yet, Garg et al. (2022) elegantly examine it in the specific setting of ICL for learning functions. Empirical evidence suggests that transformers indeed learn to implement gradient descent, after training on random instances of linear regression (Garg et al., 2022; Akyürek et al., 2022; von Oswald et al., 2023). Motivated by these observations, we theoretically investigate the loss landscape of a simple transformer architecture based on ***attention without softmax*** (Schlag et al., 2021; von Oswald et al., 2023) (see Section 2 for details).

***Summary of our main results.*** Our main contributions are the following:

▶ We provide a complete characterization of the global optimum of a single-layer linear transformer. In particular, we observe that, with the optimal parameters, the transformer implements a single step of preconditioned gradient descent. Notably, the preconditioning matrix not only adapts to the distribution of input data but also to the variance caused by data inadequacy. We present this result in Theorem 1 in Section 3.

▶ Next, we focus on a subset of the transformer parameter space, defined by a special sparsity condition (8). Such a parameter configuration allows us to formulate training transformers as a search over $k$-*step adaptive gradient-based algorithms*. Theorem 2 characterizes the global minimizers of the training objective of a two-layer linear transformer over isotropic regression instances, and shows that the optima correspond to gradient descent with adaptive stepsizes. For multilayer transformers, Theorem 3 demonstrates that gradient descent, with a data-dependent preconditioning, can be derived from a critical point of the training objective.

▶ Finally, we study the loss landscape in the absence of the sparsity condition (8), which goes beyond searching over conventional gradient-based optimization methods. In this case, we prove and interpret the structure of a critical point of the training objective. We show that a certain critical point in parameter space leads to an intriguing gradient-based algorithm that simultaneously takes gradient steps preconditioned by data covariance, and applies a linear transformation to further improve the conditioning. In the specific case when data covariance is isotropic, this algorithm corresponds to the GD++ algorithm of von Oswald et al. (2023) which is experimentally observed to be the outcome of training.

We empirically validate the critical points analyzed in Theorem 3 and Theorem 4. For a transformer with three layers, our experimental results confirm the structural of critical points. Furthermore, we observed the objective value associated with these critical points is close to $0$, suggesting that the critical points might be global optima. These experiments substantiate our theoretical analysis and suggests that our theory indeed *aligns with practice*. Code for our experiments is available at https://github.com/chengxiang/LinearTransformer.

## 1.1 Related works

The ability of neural network architectures to implement algorithms has been investigated in various context. The seminal work by Siegelmann and Sontag (1992) investigate the Turing completeness of recurrent neural networks. Despite this computational power, training recurrent networks remains a challenge. Graves et al. (2014) design an alternative neural architecture known as the *neural Turing machine*, building on *attention layers* introduced by Hochreiter and Schmidhuber (1997). Leveraging attention, Vaswani et al. (2017) propose transformers as powerful neural architectures, capable of solving various tasks in natural language processing (Devlin et al., 2019). This capability inspired a line of research that examines the algorithmic power of transformers (Pérez et al., 2021; Wei et al., 2022; Giannou et al., 2023; Akyürek et al., 2022; Olsson et al., 2022). What sets transformers apart from conventional neural networks is their impressive performance after training. In this work, we focus on understanding *how transformers learn to implement algorithms* by training over problem instances.

A line of research investigates how deep neural networks process data across their layers. The seminal work by Jastrzebski et al. (2018) observes that hidden representations across the layers of deep neural networks approximately implement gradient descent. Recent observations provide novel insights into the working mechanism of ICL for large language models, showing they can implement optimization algorithms across their layers (Garg et al., 2022; Akyürek et al., 2022; von Oswald et al., 2023). Moreover, Zhao et al. (2023); Allen-Zhu and Li (2023) observe transformer perform

dynamic programming to generate text. In this work, we theoretically study how transformer learns gradient-based algorithms for ICL.

We discuss here two related works (Zhang et al., 2023; Mahankali et al., 2023) that appeared shortly after publication of our original draft. Both of these studies focus on a single layer attention network (see Section 3). Zhang et al. (2023) prove the global convergence of gradient descent to the global optimum whose structure is analyzed independently from this study and it the same as that in Theorem 1. Mahankali et al. (2023) also characterize the global minimizer of a single layer attention without softmax for a different data distribution. In addition to results for a single-layer attention, we analyze the landscape of two and multi-layer transformers.

## 2 Setting: training linear transformers over random linear regression

In order to understand the mechanism of ICL, we consider the setting of training transformers over the random instances of linear regression, following (Garg et al., 2022; Akyürek et al., 2022; von Oswald et al., 2023). In particular, the random instances of linear regression are formalized as follows.

***Data distribution: random linear regression instances.*** Let $x^{(i)} \in \mathbb{R}^d$ be the covariates drawn i.i.d. from a distribution $D_{\mathcal{X}}$, and $w_\star \in \mathbb{R}^d$ be drawn from $D_{\mathcal{W}}$. Let $X \in \mathbb{R}^{(n+1) \times d}$ be the matrix of covariates whose row $i$ contains tokens $x^{(i)}$. Given $x^{(i)}$'s and $w_\star$, the responses are defined as $y = [\langle x^{(1)}, w_\star \rangle, \dots, \langle x^{(n)}, w_\star \rangle] \in \mathbb{R}^n$. Define the ***input matrix*** $Z_0$ as

$$Z_0 = \begin{bmatrix} z^{(1)} \ z^{(2)} \ \cdots \ z^{(n)} \ z^{(n+1)} \end{bmatrix} = \begin{bmatrix} x^{(1)} & x^{(2)} & \cdots & x^{(n)} & x^{(n+1)} \\ y^{(1)} & y^{(2)} & \cdots & y^{(n)} & 0 \end{bmatrix} \in \mathbb{R}^{(d+1) \times (n+1)}, \quad (1)$$

where zero in the above matrix is used to replace the unknown response variable corresponding to $x^{(n+1)}$. Then, our goal is to predict $w_\star^\top x^{(n+1)}$ given $Z_0$. In other words, the training data consists of pairs $(Z_0, w_\star^\top x^{(n+1)})$ for $x^{(i)} \sim D_{\mathcal{X}}$ and $w_\star \sim D_{\mathcal{W}}$. We then consider training transformers over this data distribution.

***Self-attention layer without softmax.*** Following (Schlag et al., 2021; von Oswald et al., 2023), we consider the linear self-attention layer. To motivate, we first briefly review the standard self-attention layer (Vaswani et al., 2017). Letting $Z \in \mathbb{R}^{(d+1) \times (n+1)}$ be the input matrix with $n + 1$ tokens in $\mathbb{R}^{d+1}$, a single-head self-attention layer denoted by $\mathrm{Attn}^{\mathsf{smax}}$ is a parametric map defined as

$$\mathrm{Attn}^{\mathsf{smax}}_{W_{k,q,v}}(Z) = W_v Z M \cdot \mathsf{smax}(Z^\top W_k^\top W_q Z), \quad M := \begin{bmatrix} I_n & 0 \\ 0 & 0 \end{bmatrix} \in \mathbb{R}^{(n+1) \times (n+1)}, \quad (2)$$

where $W_v, W_k, W_q \in \mathbb{R}^{(d+1) \times (d+1)}$ are the (value, key and query) weight matrices, and $\mathsf{smax}(\cdot)$ is the softmax operator which applies softmax operation to each column of the input matrix. Note that the prompt is asymmetric since the label for $x^{(n+1)}$ is excluded from the input. To reflect this asymmetric structure, the mask matrix $M$ is included in the attention. In our setting, we consider the self-attention layer that omits the softmax operation in (2). In particular, we reparameterize weights as $P := W_v \in \mathbb{R}^{(d+1) \times (d+1)}$ and $Q := W_k^\top W_q \in \mathbb{R}^{(d+1) \times (d+1)}$ and consider

$$\mathrm{Attn}_{P,Q}(Z) = PZM(Z^\top QZ). \quad (3)$$

At first glance, the omission of the softmax operation (3) might seem over-simplified. But, (von Oswald et al., 2023) proves such attention can implement gradient descent, and we will prove in Lemma 1 that it can also implement various algorithms to solve linear regression in-context.

***Architecture for prediction.*** We now present the neural network architecture that will be used throughout this paper. For the number of layers $L$, we define an ***$L$-layer transformer*** as a stack of $L$ linear self-attention blocks. Formally, denoting by $Z_\ell$ the output of the $\ell^{\text{th}}$ layer attention, we define

$$Z_{\ell+1} = Z_\ell + \frac{1}{n} \mathrm{Attn}_{P_\ell, Q_\ell}(Z_\ell) \quad \text{for } \ell = 0, 1, \dots, L-1, \quad (4)$$

The scaling factor $1/n$ is used only for ease of notation and does not influence the expressive power of the transformer. Given $Z_L$, we define $\mathsf{TF}_L(Z_0; \{P_\ell, Q_\ell\}_{\ell=0,1,\dots L-1}) = -[Z_L]_{(d+1),(n+1)}$, i.e., the $(d+1, n+1)$-th entry of $Z_L$. The reason for the minus sign is to be consistent with (von Oswald

et al., 2023), and we will clarify such a choice in Lemma 1. For training, the parameters are optimized to minimize in-context loss as

$$f\left(\{P_\ell, Q_\ell\}_{\ell=0}^L\right) = \mathbb{E}_{(Z_0, w_\star)}\left[\left(\mathsf{TF}_L(Z_0, \{P_\ell, Q_\ell\}_{\ell=0}^L) + w_\star^\top x^{(n+1)}\right)^2\right]. \tag{5}$$

***Goal: the landscape analysis of the training objective functions.*** We are interested in understanding how the optimization of $f$ leads to in-context learning. We investigate this question by analyzing its loss landscape. Such analysis is challenging due to two major reasons: *(i) $f$ is non-convex in parameters $\{P_i, Q_i\}$ even for a single layer transformer. (ii) The cross-product structures in attention makes $f$ a highly nonlinear function in its parameters.* Hence, we analyze a spectrum of settings from single-layer transformers to multi-layer transformers. For simpler settings such as single-layer transformers, we prove stronger results such as the full characterization of the global minimizers. For networks with more layers, we characterize the structure of critical points. Furthermore, we provide algorithmic interpretations of the critical points. Table 1 summarizes our results for various parameteric models.

| Results | $x^{(i)}$ | $w_\star$ | Setting | Guarantees |
|---|---|---|---|---|
| Theorem 1 | $\mathcal{N}(0, \Sigma)$ | $\mathcal{N}(0, I)$ | single-layer | global minimizers |
| Theorem 2 | $\mathcal{N}(0, I)$ | $\mathcal{N}(0, I)$ | two-layer + symmetric (8) | global minimizers |
| Theorem 3 | $\mathcal{N}(0, \Sigma)$ | $\mathcal{N}(0, \Sigma^{-1})$ | multi-layer + (8) | critical points |
| Theorem 4 | $\mathcal{N}(0, \Sigma)$ | $\mathcal{N}(0, \Sigma^{-1})$ | multi-layer + (11) | critical points |
| Theorem 5 | $\mathcal{N}(0, I)$ | $\mathcal{N}(0, I)$ | single-layer + ReLU activation | global minimizers |

Table 1: Summary of our analyses for various models and input distributions. The additional conditions (8) and (11) are about the sparsity structure of parameters. In addition, "symmetric (8)" means we additionally impose the weights to be symmetric.

**Remark 1** (***Optimizing*** (5) ***vs. practical transformer optimization***). *Interestingly, a recent work by Ahn et al. (2023) reports that common optimization algorithms such as SGD/ADAM behave remarkably similarly on the (linear Transformers + linear regression) problem as they do on (practical transformers + real language modeling tasks). In particular, they reproduce several distinctive features of transformer optimization under a simple shallow linear transformer. This work suggests that (linear transformer + linear regression) may serve as a good proxy for understanding practical transformer optimization.*

## 3 The global optimum for a single-layer transformer

For the single layer case of $L = 1$, the following result characterizes the optimal parameters $P_0$ and $Q_0$ for the in-context loss (5).

**Theorem 1** (**Single-layer; non-isotropic data**). *Assume that vector $x^{(i)}$ is sampled from $\mathcal{N}(0, \Sigma)$, i.e., a Gaussian with covariance $\Sigma = U\Lambda U^\top$ where $\Lambda = \mathrm{diag}(\lambda_1, \ldots, \lambda_d)$. Moreover, assume that $w_\star$ is sampled from $\mathcal{N}(0, I_d)$. Then, the following choice of parameters*

$$P_0 = \begin{bmatrix} 0_{d\times d} & 0 \\ 0 & 1 \end{bmatrix}, \quad Q_0 = -\begin{bmatrix} U\mathrm{diag}\left(\left\{\frac{1}{\frac{n+1}{n}\lambda_i + \frac{1}{n}\cdot(\sum_k \lambda_k)}\right\}_{i=1,\ldots,d}\right)U^\top & 0 \\ 0 & 0 \end{bmatrix}. \tag{6}$$

*is a global minimizer of $f(P, Q)$ up to re-scaling, i.e., $P_0 \leftarrow \gamma P_0$ and $Q_0 \leftarrow \gamma^{-1} Q_0$ for a scalar $\gamma$.*

See Appendix A for the proof of Theorem 1. In the specific case when the Gaussian is isotropic, i.e., $\Sigma = I_d$, the optimal $Q_0$ has the following simple form

$$Q_0 = -\frac{1}{\left(\frac{n-1}{n} + (d+2)\frac{1}{n}\right)}\begin{bmatrix} I_d & 0 \\ 0 & 0 \end{bmatrix}. \tag{7}$$

Up to scaling, the above parameter configuration is equivalent to the parameters used by von Oswald et al. (2023) to perform one step of gradient descent. Thus, in the single-layer setting, the in-context loss is indeed minimized by a transformer that implements the gradient descent algorithm.

More generally, when the in-context samples are non-isotropic, the transformer learns to implement one step of a *preconditioned* gradient descent as we shall detail in Lemma 1. Here the "preconditioning matrix" given in (6) has interesting properties:

- When the number of samples $n$ is large, the first $d \times d$ submatrix of $Q_0$ approximates $\Sigma^{-1}$, the inverse of the data covariance matrix, which is also close to the Gram matrix formed from $x^{(1)}, \ldots, x^{(n)}$. Hence the preconditioning can lead to considerably faster convergence rate when $\Sigma$ is ill-conditioned.
- Moreover, $\frac{1}{n} \sum_k \lambda_k$ in (6) acts as a regularizer. It becomes more significant when $n$ is small and variance of the $x^{(i)}$'s is high. Such an adjustment resembles structural risk minimization (Vapnik, 1999) where the regularization strength is adapted to the sample size.

## 4 Multi-layer transformers with sparse parameters

Theorem 1 proves a single layer of linear attention can implement a single step of preconditioned gradient descent. Inspired by this result, we investigate the algorithmic power of the linear transformer architecture. We show that the model can implement various optimization methods even under sparsity constraints. In particular, we impose the following restrictions on the parameters:

$$P_i = \begin{bmatrix} 0_{d \times d} & 0 \\ 0 & 1 \end{bmatrix}, \quad Q_i = - \begin{bmatrix} A_i & 0 \\ 0 & 0 \end{bmatrix} \quad \text{where } A_i \in \mathbb{R}^{d \times d}. \tag{8}$$

The next lemma proves that a forward-pass of a $L$-layer transformer, with the parameter configuration (8) is the same as taking $L$ steps of gradient descent, preconditioned by $A_\ell$.

**Lemma 1** (*Forward pass as a preconditioned gradient descent*). *Consider the $L$-layer linear transfomer parameterized by $A_0, \ldots, A_{L-1}$ as in (8). Let $y_\ell^{(n+1)}$ be the $(d+1, n+1)$-th entry of the $\ell$-th layer output, i.e., $y_\ell^{(n+1)} = [Z_\ell]_{(d+1),(n+1)}$ for $\ell = 1, \ldots, L$. Then, it holds that $y_\ell^{(n+1)} = -\langle x^{(n+1)}, w_\ell^{\mathsf{gd}} \rangle$ where $\{w_\ell^{\mathsf{gd}}\}$ is defined as $w_0^{\mathsf{gd}} = 0$ and as follows for $\ell = 1, \ldots, L-1$:*

$$w_{\ell+1}^{\mathsf{gd}} = w_\ell^{\mathsf{gd}} - A_\ell \nabla R_{w_\star} \left( w_\ell^{\mathsf{gd}} \right) \quad \text{where} \quad R_{w_\star}(w) \coloneqq \frac{1}{2n} \sum_{i=1}^{n} (w^\top x_i - w_\star^\top x_i)^2. \tag{9}$$

See Subsection C.1 for a proof. The iterative scheme (9) includes various optimization methods including gradient descent with $A_\ell = \gamma_\ell I_d$, and (adaptive) preconditioned gradient descent, where the preconditioner $A_\ell$ depends on the time step. In the upcoming sections, we characterize how the optimal $\{A_\ell\}$ are linked to the input distribution.

### 4.1 Warm-up: optimal two-layer transformer with symmetric weights

For the rest of this section, we will study the optimal parameters for the in-context loss under the constraint of Eq. (8). Later in Section 5, we analyze the optimal model for a more general parameters. For a two-layer transformer, the next Theorem proves the optimal in-context loss obtains the simple gradient descent with adaptive coordinate-wise stepsizes.

**Theorem 2** (Global optimality for the two-layer (symmetric) transformer). *Consider the optimization of in-context loss for a two-layer transformer with the parameter configuration in Eq. (8), and additionally assume that $A_1, A_2$ are symmetric matrices. More formally, consider*

$$\min_{A_1, A_2 \text{ are symmetric}} f \left\{ P_\ell = \begin{bmatrix} 0_{d \times d} & 0 \\ 0 & 1 \end{bmatrix}, Q_\ell = \begin{bmatrix} -A_\ell & 0 \\ 0 & 0 \end{bmatrix} \right\}_{\ell=1,2}.$$

*Assume $x^{(i)} \overset{i.i.d.}{\sim} N(0, I_d)$ and $w_\star \sim N(0, I_d)$; then, there are diagonal matrices $A_1$ and $A_2$ that are a global minimizer of $f$.*

Combining the above result with Lemma 1 concludes that the two iterations of gradient descent with *coordinate-wise adaptive stepsizes* achieve the minimal in-context loss for isotropic Gaussian inputs. Gradient descent with adaptive stepsizes such as Adagrad (Duchi et al., 2011) are widely used in machine learning. While Adagrad adjusts its stepsize based on the individual problem instance, the algorithm learned adjusts its stepsize to the underlying data distribution.

## 4.2 Multi-layer transformers

We now turn to the setting of general $L$-layer transformers, for any positive integer $L$. The next theorem proves that certain critical points of the in-context loss effectively implement a specific preconditioned gradient algorithm, where the preconditioning matrix is the inverse covariance of the input distribution. Before stating this result, let us first consider a motivating scenario in which the data-covariance matrix is non-identity:

***Linear regression with distorted view of the data:*** Suppose that $\overline{w}_\star \sim \mathcal{N}(0, I)$ and the *latent* covariates are $\overline{x}^{(1)}, \dots, \overline{x}^{(n+1)}$, drawn i.i.d from $\mathcal{N}(0, I)$. We are given $y^{(1)}, \dots, y^{(n)}$, with $y^{(i)} = \langle \overline{x}^{(i)}, \overline{w}_\star \rangle$. However, we *do not observe* the latent covariates $\overline{x}^{(i)}$. Instead, we observe the *distorted* covariates $x^{(i)} = W\overline{x}^{(i)}$, where $W \in \mathbb{R}^{d \times d}$ is a distortion matrix. Thus the prompt consists of $(x^{(1)}, y^{(1)}), \dots, (x^{(n)}, y^{(n)})$, as well as $x^{(n+1)}$. The goal is still to predict $y^{(n+1)}$. Note that this setting is quite common in practice, when covariates are often represented in an arbitrary basis.

Assume that $\Sigma := WW^\top \succ 0$. We verify from our definitions that for $w_\star := \Sigma^{-1/2}\overline{w}_\star$, $y^{(i)} = \langle x^{(i)}, w_\star \rangle$. Furthermore, $x^{(i)} \sim \mathcal{N}(0, \Sigma)$ and $w_\star \sim \mathcal{N}(0, \Sigma^{-1})$. From Lemma 1, the transformer with weight matrices $\{A_0, \dots, A_{L-1}\}$ implements preconditioned gradient descent with respect to $R_{w_\star}(w) = \frac{1}{2n}(w - w_\star)^T XX^\top (w - w_\star)$, with $X = [x^{(1)}, \dots, x^{(n)}]$. Under this loss, the Hessian matrix $\nabla^2 R_{w_\star}(w) = \frac{1}{2n}XX^\top$ (at least in the case of large $n$). For any fixed prompt, Newton's method corresponds to $A_i \propto (XX^\top)^{-1}$, which makes the problem well-conditioned even if $\Sigma$ is very degenerate. As we will see in Theorem 3 below, the choice of $A_i \propto \Sigma^{-1} = \mathbb{E}[XX^\top]^{-1}$ appears to be a *stationary point* of the loss landscape, in expectation over prompts.

Before stating the theorem, we introduce the following simplified notation: let $A := \{A_i\}_{i=0}^{L-1} \in \mathbb{R}^{L \times d \times d}$. We use $f(A)$ to denote the in-context loss of $f(\{P_i, Q_i\}_{i=0}^{L-1})$ as defined in (5), when $Q_i$ depends on $A_i$, and $P_i$ is a constant matrix, as described in (8).

**Theorem 3.** *Assume that $x^{(i)} \overset{iid}{\sim} \mathcal{N}(0, \Sigma)$ and $w_\star \sim \mathcal{N}(0, \Sigma^{-1})$, for $i = 1, \dots, n$, and for some $\Sigma \succ 0$. Consider the optimization of in-context loss for a $k$-layer transformer with the the parameter configuration in Eq. (8) given by:*

$$\min_{\{A_i\}_{i=0}^{L-1}} f(A).$$

*Let $\mathcal{S} \subset \mathbb{R}^{L \times d \times d}$ be defined as follows: $A \in \mathcal{S}$ if and only if for all $i = 0, \dots, L-1$, there exists scalars $a_i \in \mathbb{R}$ such that $A_i = a_i \Sigma^{-1}$. Then*

$$\inf_{(A,B) \in \mathcal{S}} \sum_{i=0}^{L-1} \|\nabla_{A_i} f(A, B)\|_F^2 = 0,$$

*where $\nabla_{A_i} f$ denotes derivative wrt the Frobenius norm $\|A_i\|_F$.*

As discussed in the motivation above, under the setting of $A_i = a_i \Sigma^{-1}$, the linear transformer implements an algorithm that is reminiscent of Newton's method (as well as a number of other adaptive algorithms such as the full-matrix variant of Adagrad); these can converge significantly faster than vanilla gradient descent when the problem is ill-conditioned. The proposed parameters $A_i$ in Theorem 3 are also similar to $A_i$'s in Theorem 1 when $n$ is large. However, in contrast to Theorem 1, there is no trade-off with statistical robustness; this is because $w_\star$ has covariance matrix $\Sigma^{-1}$ in the Theorem 3, while Theorem 1 has isotropic $w_\star$.

Unlike our prior results, Theorem 3 only guarantees that the set $\mathcal{S}$ of transformer prameters satisfying $\{A_i \propto \Sigma^{-1}\}_{i=0}^{L-1}$ essentially[2] contains critical points of the in-context loss. However, in the next section, we show experimentally that this choice of $A_i$'s does indeed seem to be recovered by training.

We defer the proof of Theorem 3 to Subsection B.2. Due to the complexity of the transformer function, even verifying critical points can be challenging. We show that the in-context loss can be equivalently written as (roughly) a matrix polynomial involving the weights at each layer. By

---

[2] A subtle issue is that the infimum may not be attained, so it is possible that $\mathcal{S}$ contains points with arbitrarily small gradient, but does not contain a point with exactly 0 gradient.

exploiting invariances in the underlying distribution of prompts, we construct a flow, contained entirely in $\mathcal{S}$, whose objective value decreases as fast as gradient flow. Since $f$ is lower bounded, we conclude that there must be points in $\mathcal{S}$ whos gradient is arbitrarily small.

### 4.3 Experimental validations for Theorem 3

We present here an empirical verification of our results in Theorem 3. We consider the ICL loss for linear regression. The dimension is $d = 5$, and the number of training samples in the prompt is $n = 20$. Both $x^{(i)} \sim \mathcal{N}(0, \Sigma)$ and $w_\star \sim \mathcal{N}(0, \Sigma^{-1})$, where $\Sigma = U^T D U$, where $U$ is a uniformly random orthogonal matrix, and $D$ is a fixed diagonal matrix with entries $(1, 1, 0.25, 0.0625, 1)$.

We optimizes $f$ for a three-layer linear transformer using ADAM, where the matrices $A_0, A_1$, and $A_2$ are initialized by i.i.d. Gaussian matrices. Each gradient step is computed from a minibatch of size 20000, and we resample the minibatch every 100 steps. We clip the gradient of each matrix to 0.01. All plots are averaged over 5 runs with different $U$ (i.e. $\Sigma$) sampled each time.

Figure 1d plots the average loss. We observe that the training converges to an almost $0$ value, suggesting the convergence to global minimum. The parameters at convergence match the stationary point introduced in Theorem 3, and indeed appear to be globally optimal.

To quantify the similarity between $A_0, A_1, A_2$ and $\Sigma^{-1}$ (up to scaling), we use the *normalized Frobenius norm distance*: $\mathrm{Dist}(M, I) := \min_\alpha \frac{\|M - \alpha \cdot I\|}{\|M\|_F}$, (equivalent to choosing $\alpha := \frac{1}{d} \sum_{i=1}^d M[i, i]$). This is essentially the projection distance of $M/\|M\|_F$ onto the space of scaled identity matrices.

We plot $\mathrm{Dist}(A_i, I)$, averaged over 5 runs, against iteration in Figures 1a,1b,1c. In each plot, the blue line represents $\mathrm{Dist}(\Sigma^{1/2} A_i \Sigma^{1/2}, I)$, and we verify that the optimal parameters are converging to the critical point introduced in Theorem 3, which implements preconditioned gradient descent. The red line represents $\mathrm{Dist}(A_i, I)$; it remains constant indicating that the trained transformer is not implementing plain gradient descent. Figures 2a–2c visualize each $\Sigma^{1/2} A_i \Sigma^{1/2}$ matrix at the end of training to further validate that the learned parameter is as described in Theorem 3.

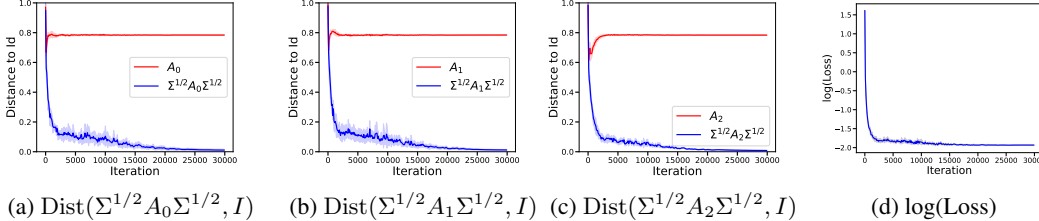

(a) $\mathrm{Dist}(\Sigma^{1/2} A_0 \Sigma^{1/2}, I)$    (b) $\mathrm{Dist}(\Sigma^{1/2} A_1 \Sigma^{1/2}, I)$    (c) $\mathrm{Dist}(\Sigma^{1/2} A_2 \Sigma^{1/2}, I)$      (d) $\log(\mathrm{Loss})$

Figure 1: Plots for verifying convergence of general linear transformer, defined in Theorem 3. Figure (d) shows convergence of loss to 0. Figures (a),(b),(c) illustrate convergence of $A_i$'s to identity. More specifically, the blue line represents $\mathrm{Dist}(\Sigma^{1/2} A_i \Sigma^{1/2}, I)$, which measures the convergence to the critical point introduced in Theorem 3 (corresponding to $\Sigma^{-1}$-preconditioned gradient descent). The red line represents $\mathrm{Dist}(A_i, I)$; it remains constant indicating that the trained transformer is not implementing plain gradient descent.

## 5 Multi-layer transformers beyond standard optimization methods

In this section, we study the more general setting of

$$ P_i = \begin{bmatrix} B_i & 0 \\ 0 & 1 \end{bmatrix}, \quad Q_i = \begin{bmatrix} A_i & 0 \\ 0 & 0 \end{bmatrix} \quad \text{where } A_i, B_i \in \mathbb{R}^{d \times d}. \tag{11} $$

Note that $A_i, B_i$ are not constrained to be symmetric. Similar to Section 4, we introduce the following simplified notation: let $A := \{A_i\}_{i=0}^{L-1} \in \mathbb{R}^{L \times d \times d}$ and $B := \{B_i\}_{i=0}^{L-1} \in \mathbb{R}^{L \times d \times d}$. We use $f(A, B)$ to denote the in-context loss of $f\left(\{P_i, Q_i\}_{i=0}^{L-1}\right)$ as defined in (5), when $P_i$ and $Q_i$ depend on $B_i$ and $A_i$ as described in (11).

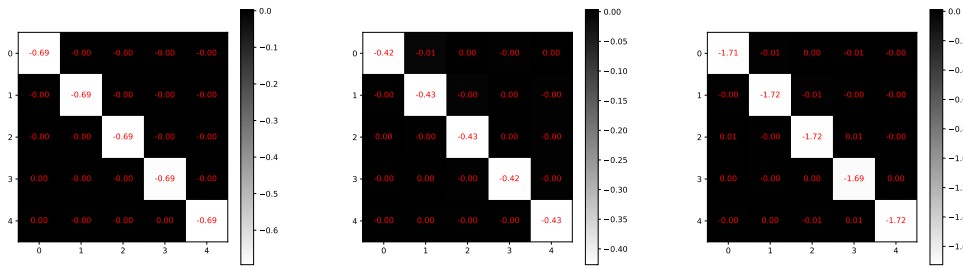

(a) Visualization of $\Sigma^{1/2} A_0 \Sigma^{1/2}$    (b) Visualization of $\Sigma^{1/2} A_1 \Sigma^{1/2}$    (c) Visualization of $\Sigma^{1/2} A_2 \Sigma^{1/2}$

Figure 2: Visualization of learned weights for the setting of Theorem 3. We visualize each $\Sigma^{1/2} A_i \Sigma^{1/2}$ matrix at the end of training. Note that the optimized weights match the stationary point discussed in Theorem 3.

With this relaxed parameter configuration, it turns out transformers can learn algorithms beyond the conventional preconditioned gradient descent. The next theorem asserts the possibility of learning a novel preconditioned gradient method. Let $L$ be a fixed but arbitrary number of layers.

**Theorem 4.** *Let $\Sigma$ denote any PSD matrix. Assume that $x^{(i)} \stackrel{iid}{\sim} \mathcal{N}(0, \Sigma)$ and $w_\star \sim \mathcal{N}(0, \Sigma^{-1})$, for $i = 1, \ldots, n$, and for some $\Sigma \succ 0$. Consider the optimization of in-context loss for a $L$-layer linear transformer with the the parameter configuration in Eq. (11) given by:*

$$\min_{\{A_i, B_i\}_{i=0}^{L-1}} f(A, B).$$

*Let $\mathcal{S} \subset \mathbb{R}^{2 \times L \times d \times d}$ be defined as follows: $(A, B) \in \mathcal{S}$ if and only if for all $i \in \{0, \ldots, k\}$, there exists scalars $a_i, b_i \in \mathbb{R}$ such that $A_i = a_i \Sigma^{-1}$ and $B_i = b_i I$. Then*

$$\inf_{(A,B) \in \mathcal{S}} \sum_{i=0}^{L-1} \|\nabla_{A_i} f(A, B)\|_F^2 + \|\nabla_{B_i} f(A, B)\|_F^2 = 0, \tag{12}$$

*where $\nabla_{A_i} f$ denotes derivative wrt the Frobenius norm $\|A_i\|_F$.*

In words, parameter matrices in $\mathcal{S}$ implement the following algorithm: $\left\{ A_i = a_i \Sigma^{-1} \right\}_{i=0}^{L-1}$ plays the role of a distribution-dependent preconditioner for the gradient steps. At the same time, $B_i = b_i I$ transforms the covariates themselves to make the Gram matrix have better condition number with each iteration. When the $\Sigma = I$, the algorithm implemented by $A_i \propto I, b_i \propto I$ is exactly the GD++ algorithm proposed in (von Oswald et al., 2023) (up to stepsize).

The result in (12) says that the set $\mathcal{S}$ *essentially*[3] contains critical points of the in-context loss $f(A, B)$. In the next section, we provide empirical evidence that the trained transformer parameters do in fact converge to a point in $\mathcal{S}$.

### 5.1 Experimental validations for Theorem 4

The experimental setup is similar to Subsection 4.3: we consider ICL for linear regression with $n = 10, d = 5$, with $x^{(i)} \sim \mathcal{N}(0, \Sigma)$ and $w_\star \sim \mathcal{N}(0, \Sigma^{-1})$, where $\Sigma = U^T D U$, where $U$ is a uniformly random orthogonal matrix, and $D$ is a fixed diagonal matrix with entries $(1, 1, 0.25, 0.0625, 1)$. We train a three-layer linear transformer, under the constraints in (11) which is less restrictive than (8) in Subsection 4.3. We train the matrices $A_0, A_1, A_2, B_0, B_1$ [4] using ADAM with the same setup as in Section Subsection 4.3. We repeat this experiment 5 times with different random seeds, each time we sample a different $U$ (i.e. $\Sigma$).

---

[3]Once again, similar to the case of Theorem 3, the infimum may not be attained, so it is possible that $\mathcal{S}$ contains points with arbitrarily small gradient, but does not contain a point with exactly 0 gradient.

[4]Note that the objective function does not depend on $B_2$.

In Figure 3c, we plot the in-context loss through the iterations of ADAM; the loss appears to be converging to 0, suggesting that parameters are converging to the global minimum.

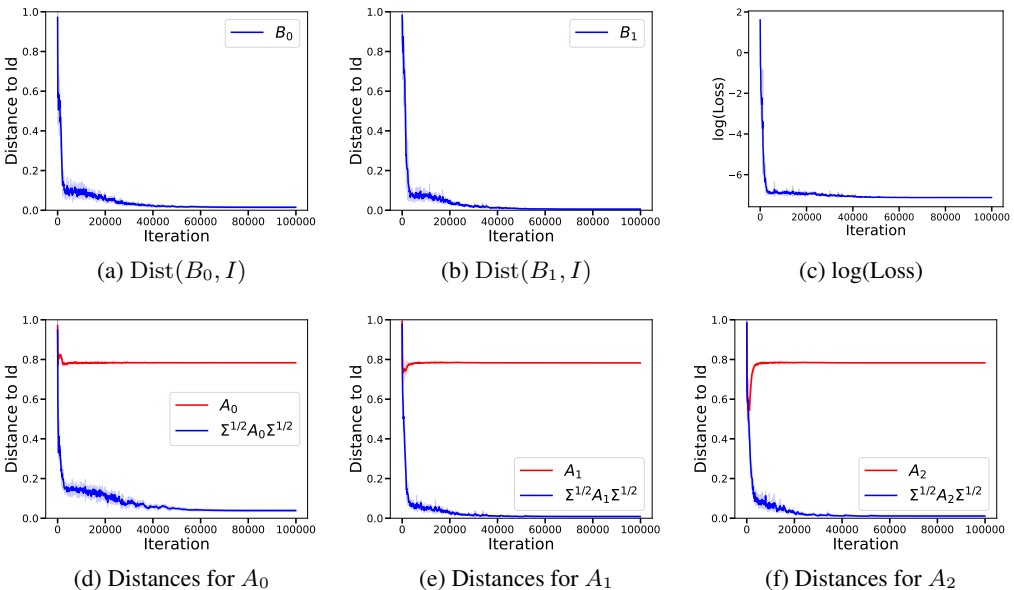

(a) $\mathrm{Dist}(B_0, I)$         (b) $\mathrm{Dist}(B_1, I)$         (c) log(Loss)

(d) Distances for $A_0$       (e) Distances for $A_1$       (f) Distances for $A_2$

Figure 3: Plots for verifying convergence of general linear transformer, defined in Theorem 4. Figure (c) shows convergence of loss to 0. Figures (a),(b) illustrate convergence of $B_0, B_1$ to identity. Figures (d),(e),(f) illustrate convergence of $A_i$'s to $\Sigma^{-1}$.

We next verify that the parameters at convergence are consistent with Theorem 4. We will once again use $\mathrm{Dist}(M, I)$ to measure the distance from $M$ to the identity matrix, up to scaling (see Subsection 4.3 for definition of $Dist$). Figures 3a and 3b show that $B_0$ and $B_1$ are close to identity, as $\mathrm{Dist}(B_i, I)$ appears to be decreasing to 0. Figures 3d, 3e and 3f plot $\mathrm{Dist}(A_i, I)$ (red line) and $\mathrm{Dist}(\Sigma^{1/2} A_i \Sigma^{1/2}, I)$ (blue line); the results here suggest that $A_i$ is converging to $\Sigma^{-1}$, up to scaling. In Figures 3a and 3b, we observe that $B_0$ and $B_1$ also converge to the identity matrix (*without* left and right multiplication by $\Sigma^{1/2}$), consistent with Theorem 4.

We visualize each of $B_0, B_1$ in Figure 4 and $A_0, A_1, A_2$ in Figure 5a-5c at the end of training. We highlight two noteworthy observations:

1. Let $X_k \in \mathbb{R}^{d \times n}$ denote the first $d$ rows of $Z_k$, which are the output at layer $k - 1$ defined in (4). Then the update to $X_k$ is $X_{k+1} = X_k + B_k X_k M X_k^T A_k X_k \approx X_{k+1} = X_k \left( I - |a_k b_k| M X_k^T X_k \right)$, where $M$ is a mask defined in (2). As noted by von Oswald et al. (2023), this may be motivated by curvature correction.

2. As seen in Figures 5a-5c in the Appendix, $\|A_0\| \leq \|A_1\| \leq \|A_2\|$ that implies the transformer implements gradient descent with a small stepsize at the beginning and a large stepsize at the end. This makes intuitive sense as $X_2$ is better-conditioned compared to $X_1$, due to the choice of $B_0, B_1$. This can be contrasted with the plots in Figures (2a)-(2c), where similar trends are not as pronounced because $B_i$'s are constrained to be 0.

## 6 Discussion

We take a first step toward proving that transformers can learn algorithms when trained over a set of random problem instances. Specifically, we investigate the possibility of learning gradient based methods when training on the in-context loss for linear regression. For a single layer transformer, we prove that the global minimum corresponds to a single iteration of preconditioned gradient descent. For multiple layers, we show that certain parameters that correspond to the critical points of the in-context loss can be interpreted as a broad family of adaptive gradient-based algorithms.

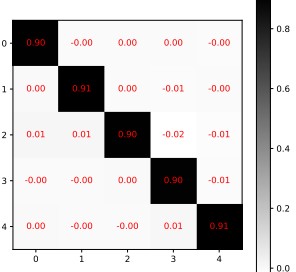 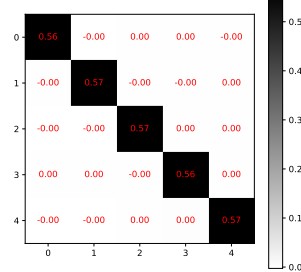

Figure 4: Visualization of optimized weight matrices $B_0$ (left) and $B_1$ (right). One can see that the weight pattern matches the stationary point analyzed in Theorem 4. Matrices $A_0$, $A_1$ and $A_2$ are similar to Figure 2, and are visualized in Figure 5 in Appendix D.

We discuss below two interesting future directions.

**Beyond linear attention.** The standard transformer architecture comes with nonlinear activations in attention. Hence, the natural question here is to ask the effect of nonlinear activations for our main results. Empirically, von Oswald et al. (2023) have observed that for linear regression task, softmax activations generally degrade the prediction performance, and in particular, softmax transformers typically need more attention heads to match their performance with that of linear transformers.

As a first step analysis, we consider the nonlinear attention defined as

$$\text{Attn}^{\sigma}_{P,Q}(Z) \coloneqq PZM\,\sigma(Z^{\top}QZ) \quad \text{where } \sigma : \mathbb{R} \to \mathbb{R} \text{ is applied entry-wise.}$$

The following result is an analog of Theorem 1 for single-layer nonlinear attention. It characterizes a global minimizer for this setting with ReLU activation. Here, our choice of ReLU activation was motivated by Wortsman et al. (2023) who observed that ReLU attention matches the performance of softmax attention for vision transformers.

**Theorem 5.** *Consider the single layer nonlinear attention setting with $\sigma = \text{ReLU}$. Assume that vector $x^{(i)}$ is sampled from $\mathcal{N}(0, I_d)$. Moreover, assume that $w_{\star}$ is sampled from $\mathcal{N}(0, I_d)$. Consider the parameter configuration $P_0, Q_0$ where we additionally assume that the last row of $Q_0$ is zero. Then, the following parameters form a global minimizer of the corresponding in-context loss:*

$$P_0 = \begin{bmatrix} 0_{d \times d} & 0 \\ 0 & 1 \end{bmatrix}, \quad Q_0 = -\frac{1}{\frac{1}{2}\frac{n-1}{n} + (d+2)\frac{1}{n}} \cdot \begin{bmatrix} I_d & 0 \\ 0 & 0 \end{bmatrix}.$$

The proof of Theorem 5 involves an instructive argument and leverages tools from (Erdogdu et al., 2016); we defer it to Subsection A.4. Thus, for isotropic Gaussian data, the structure of global minimum under ReLU attention is similar to the global minimum with linear attention, established in Theorem 1 (specifically the minimizer for the isotropic date given in (7)).

**Refined landscape analysis for multilayer transformer.** Theorem 4 proves that a stationary point of the in-context loss corresponds to implementing a preconditioned gradient method. However, we do not prove that all critical points of the non-convex objective lead to similar optimization methods. In fact, in Lemma 4 in Appendix B, we prove that the in-context loss can have multiple critical points. It will be interesting to analyze the set of all critical points and try to understand their algorithmic interpretations, as well as quantify their (sub)optimality.

## Acknowledgments and Disclosure of Funding

We thank Ekin Akyürek, Johannes von Oswald, Alex Gu and Joshua Robinson for helpful discussions. Kwangjun Ahn was supported by the ONR grant (N00014-20-1-2394) and MIT-IBM Watson as well as a Vannevar Bush fellowship from Office of the Secretary of Defense. Kwangjun Ahn also acknowledges support from the Kwanjeong Educational Foundation. Xiang Cheng acknowledges support from NSF CCF-2112665 (TILOS AI Research Institute). Hadi Daneshmand acknowledges support from NSF TRIPODS program (award DMS-2022448). Suvrit Sra acknowledges support from an NSF CAREER grant (1846088), and NSF CCF-2112665 (TILOS AI Research Institute).

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

# Appendix

## A   Proofs for the single layer case

In this section, we prove our characterization of global minima for the single layer case (Theorem 1). We begin by simplifying the loss into a more concrete form. Throughout the proof, we will write $P, Q$ instead of $P_0, Q_0$ for brevity.

### A.1   Rewriting the loss function

Recall the in-context loss (5) for the single layer case $f(P, Q)$ is defined as:

$$f(P, Q) = \mathbb{E}_{Z_0, w_\star} \left[ \left( Z_0 + \frac{1}{n} \mathrm{Attn}_{P,Q}(Z_0) \right)_{(d+1),(n+1)} + w_\star^\top x^{(n+1)} \right]^2$$

From the definition of attention given in (3), one can further spell out the expression $Z_0 + \frac{1}{n} \mathrm{Attn}_{P,Q}(Z_0)$ using the notation $Z_0 = [z^{(1)} \ z^{(2)} \ \cdots \ z^{(n+1)}]$ as follows:

$$[z^{(1)} \ \cdots \ z^{(n+1)}] + \frac{1}{n} P[z^{(1)} \ \cdots \ z^{(n+1)}] M \left( [z^{(1)} \ \cdots \ z^{(n+1)}]^\top Q[z^{(1)} \ \cdots \ z^{(n+1)}] \right)$$

$$= [z^{(1)} \ \cdots \ z^{(n+1)}] + \frac{1}{n} P \left( \sum_{i=1}^n z^{(i)} z^{(i)\top} \right) Q[z^{(1)} \ \cdots \ z^{(n+1)}].$$

Thus, the last column of the above matrix can be expressed as

$$\begin{bmatrix} x^{(n+1)} \\ 0 \end{bmatrix} + \frac{1}{n} P \left( \sum_{i=1}^n z^{(i)} z^{(i)\top} \right) Q \begin{bmatrix} x^{(n+1)} \\ 0 \end{bmatrix},$$

where note that the summation is for $i = 1, 2, \ldots, n$ due to the mask matrix $M$. Therefore, letting $b^\top$ be the last row of $P$, and $A \in \mathbb{R}^{d+1,d}$ be the first $d$ columns of $Q$ (as we did in (14)), then $\left[ Z_0 + \frac{1}{n} \mathrm{Attn}_{P,Q}(Z_0) \right]_{(d+1),(n+1)}$ can be written as

$$\frac{1}{n} b^\top \left( \sum_{i=1}^n z^{(i)} z^{(i)\top} \right) A \begin{bmatrix} x^{(n+1)} \\ 0 \end{bmatrix}, \tag{13}$$

in other words, $f(P, Q)$ only depends on the parameter $b$ and $A$. Henceforth, we will write $f(P, Q)$ as $f(b, A)$. Let us summarize our conclusion so far since it's crucial for the analysis to follow.

---

**Conclusion so far:** A careful inspection reveals that the in-context loss only depends on the last row of $P$ and the first $d$ columns of $Q$. Thus, consider the following parametrization

$$P = \begin{bmatrix} 0 \\ b^\top \end{bmatrix} \quad \text{and} \quad Q = [A \quad 0] \text{ , where } b \in \mathbb{R}^{d+1} \text{ and } A \in \mathbb{R}^{(d+1) \times d}. \tag{14}$$

Now with this parametrization, the in-context loss can be written as $f(b, A) := f([0 \ b]^\top, [A \ 0])$.

---

Now, let us spell out $f(b, A)$ based on (13) as follows:

$$f(b, A) = \mathbb{E}_{Z_0, w_\star} \left[ b^\top \underbrace{\frac{1}{n} \sum_i z^{(i)} z^{(i)^\top}}_{=:\mathsf{G}} A x^{(n+1)} + w_\star^\top x^{(n+1)} \right]^2$$

$$=: \mathbb{E}_{Z_0, w_\star} \left[ b^\top \mathsf{G} A x^{(n+1)} + w_\star^\top x^{(n+1)} \right]^2 = \mathbb{E}_{Z_0, w_\star} \left[ (b^\top \mathsf{G} A + w_\star^\top) x^{(n+1)} \right]^2, \tag{15}$$

where we used the notation $\mathsf{G} := \frac{1}{n} \sum_i z^{(i)} z^{(i)^\top}$ to simplify. We now analyze the global minima of this loss function. To illustrate the proof idea clearly, we begin with the proof for the simpler case of isotropic data.

### A.2 Warm-up: proof for the isotropic data

As a warm-up, we first prove the result for the special case where $x^{(i)}$ is sampled from $\mathcal{N}(0, I_d)$.

***1. Decomposing the loss function into components.*** Writing $A = [a_1 \ a_1 \ \cdots \ a_d]$, and use the fact that $\mathbb{E}[x^{(n+1)}[j] x^{(n+1)}[j']] = 0$ for $j \neq j'$ and $\mathbb{E}[x^{(n+1)}[j]^2] = 1$, we get

$$f(b, A) = \sum_{j=1}^d \mathbb{E}_{Z_0, w_\star} \left[ b^\top \mathsf{G} \, a_j + w_\star[j] \right]^2 \mathbb{E}[x^{(n+1)}[j]^2] = \sum_{j=1}^d \mathbb{E}_{Z_0, w_\star} \left[ b^\top \mathsf{G} \, a_j + w_\star[j] \right]^2.$$

The key idea is to characterize the global minima of each component in the summation separately. Another key idea is to reparametrize the cost function given the following identity:

$$\mathbb{E}_{Z_0, w_\star} \left[ b^\top \mathsf{G} \, a_j + w_\star[j] \right]^2 = \mathbb{E}_{Z_0, w_\star} \left[ \mathrm{Tr}(\mathsf{G} \, a_j b^\top) + w_\star[j] \right]^2 = \mathbb{E}_{Z_0, w_\star} \left[ \langle \mathsf{G}, b a_j^\top \rangle + w_\star[j] \right]^2,$$

where we use the notation $\langle X, Y \rangle := \mathrm{Tr}(XY^\top)$ for two matrices $X$ and $Y$ here and below. Given the above identity, we define each component in the summation as follows.

$$\boxed{f_j(X) := \mathbb{E}_{Z_0, w_\star} \left[ \langle \mathsf{G}, X \rangle + w_\star[j] \right]^2 \quad \text{for } X \in \mathbb{R}^{(d+1) \times (d+1)}.}$$

***2. Characterizing global minima of each component.*** To characterize the global minima of each objective, we prove the following result.

**Lemma 2 (Global minima of each component).** *Suppose that $x^{(i)}$ is sampled from $\mathcal{N}(0, I_d)$ and $w_\star$ is sampled from $\mathcal{N}(0, I_d)$. Consider the following objective (here, $\langle X, Y \rangle := \mathrm{Tr}(XY^\top)$ for two matrices $X$ and $Y$)*

$$f_j(X) = \mathbb{E}_{Z_0, w_\star} \left[ \langle \mathsf{G}, X \rangle + w_\star[j] \right]^2.$$

*Then a global minimum is given as*

$$X_j = -\frac{1}{\left( \frac{n-1}{n} + (d+2)\frac{1}{n} \right)} E_{d+1, j},$$

*where $E_{i_1, i_2}$ is the matrix whose $(i_1, i_2)$-th entry is 1, and the other entries are zero.*

**Proof of Lemma 2.** Note first that $f_j$ is convex in $X$. Hence, in order to show that matrix $X_j$ is the global optimum of $f_j$, it suffices to show that the gradient vanishes at that point, in other words, $\nabla f_j(X_j) = 0$. To verify this, let us compute the gradient of $f_j$: for a matrix $X$,

$$\nabla f_j(X) = 2\,\mathbb{E}\left[\langle \mathsf{G}, X\rangle\, \mathsf{G}\right] + 2\,\mathbb{E}\left[w_\star[j]\, \mathsf{G}\right],$$

where we recall that $\mathsf{G}$ is defined as

$$\mathsf{G} = \frac{1}{n}\sum_i \begin{bmatrix} x^{(i)}x^{(i)\top} & y^{(i)}x^{(i)} \\ y^{(i)}x^{(i)\top} & y^{(i)2} \end{bmatrix}.$$

To verify that the gradient is equal to zero, let us first compute $\mathbb{E}\left[w_\star[j]\,\mathsf{G}\right]$. For each $i = 1, \ldots, n$, note that $\mathbb{E}[w_\star[j]\, x^{(i)}x^{(i)\top}] = O$ because $\mathbb{E}[w_\star] = 0$. Moreover, $\mathbb{E}[w_\star[j]\, (y^{(i)})^2] = 0$ because $w_\star$ is symmetric, i.e., $w_\star \overset{d}{=} -w_\star$, and $y^{(i)} = \langle w_\star, x^{(i)}\rangle$. Lastly, for $k = 1, 2, \ldots, d$, we have

$$\mathbb{E}[w_\star[j]\, y^{(i)}\, x^{(i)}[k]] = \mathbb{E}[w_\star[j]\, \langle w_\star, x^{(i)}\rangle\, x^{(i)}[k]] = \mathbb{E}\left[w_\star[j]^2\, x^{(i)}[j]\, x^{(i)}[k]\right] = \mathbb{1}_{[j=k]} \quad (16)$$

because $\mathbb{E}[w_\star[i]\, w_\star[j]] = 0$ for $i \neq j$. Combining the above calculations, it follows that

$$\boxed{\mathbb{E}\left[w_\star[j]\,\mathsf{G}\right] = E_{d+1,j} + E_{j,d+1}\,.} \quad (17)$$

In order to compute $\mathbb{E}\left[\langle \mathsf{G}, X\rangle\, \mathsf{G}\right]$, let us compute $\mathbb{E}\left[\langle \mathsf{G}, E_{i,i'}\rangle\, \mathsf{G}\right]$ for $i, i' = 1, \ldots, d+1$. Without loss of generality, $i \geq i'$. First of all We now compute compute $\mathbb{E}\left[\langle \mathsf{G}, E_{d+1,j}\rangle\, \mathsf{G}\right]$. Note first that

$$\langle \mathsf{G}, E_{d+1,j}\rangle = \sum_i \langle w_\star, x^{(i)}\rangle\, x^{(i)}[j]\,.$$

Hence, it holds that

$$\mathbb{E}\left[\langle \mathsf{G}, E_{d+1,j}\rangle \left(\sum_i x^{(i)}x^{(i)\top}\right)\right] = \mathbb{E}\left[\left(\sum_i \langle w_\star, x^{(i)}\rangle\, x^{(i)}[j]\right)\left(\sum_i x^{(i)}x^{(i)\top}\right)\right] = O\,.$$

because $\mathbb{E}[w_\star] = 0$. Next, we have

$$\mathbb{E}\left[\langle \mathsf{G}, E_{d+1,j}\rangle \left(\sum_i y^{(i)2}\right)\right] = \mathbb{E}\left[\left(\sum_i \langle w_\star, x^{(i)}\rangle\, x^{(i)}[j]\right)\left(\sum_i y^{(i)2}\right)\right] = 0$$

because $w_\star \overset{d}{=} -w_\star$. Lastly, we compute

$$\mathbb{E}\left[\langle \mathsf{G}, E_{d+1,j}\rangle \left(\sum_i y^{(i)}x^{(i)\top}\right)\right]\,.$$

To that end, note that for $j \neq j'$,

$$\mathbb{E}\left[\langle w_\star, x^{(i)}\rangle\, x^{(i)}[j]\, \langle w_\star, x^{(i')}\rangle\, x^{(i')}[j']\right] = \begin{cases} \mathbb{E}[\langle x^{(i)}, x^{(i')}\rangle\, x^{(i)}[j]\, x^{(i')}[j']] = 0 & \text{if } i \neq i', \\ \mathbb{E}[\|x^{(i)}\|^2\, x^{(i)}[j]\, x^{(i)}[j']] = 0 & \text{if } i = i', \end{cases}$$

and

$$\mathbb{E}\left[\langle w_\star, x^{(i)}\rangle\, x^{(i)}[j]\, \langle w_\star, x^{(i')}\rangle\, x^{(i')}[j]\right] = \begin{cases} \mathbb{E}[(x^{(i)}[j])^2\, (x^{(i')}[j])^2] = 1 & \text{if } i \neq i', \\ \mathbb{E}\left[\langle w_\star, x^{(i)}\rangle^2\, (x^{(i)}[j])^2\right] = d+2 & \text{if } i = i', \end{cases} \quad (18)$$

where the last case follows from the fact that the fourth moment of Gaussian is 3 and

$$\mathbb{E}\left[\langle w_\star, x^{(i)}\rangle^2\, (x^{(i)}[j])^2\right] = \mathbb{E}\left[\|x^{(i)}\|^2\, (x^{(i)}[j])^2\right] = 3 + d - 1 = d + 2.$$

Combining the above calculations together, we arrive at

$$\mathbb{E}\left[\langle \mathsf{G}, E_{d+1,j}\rangle\, \mathsf{G}\right] = \frac{1}{n^2}\cdot\left(n(n-1) + (d+2)n\right)\left(E_{d+1,j} + E_{j,d+1}\right)$$

$$= \left(\frac{n-1}{n} + (d+2)\frac{1}{n}\right)\left(E_{d+1,j} + E_{j,d+1}\right). \quad (19)$$

Therefore, combining (17) and (19), the results follows. $\qquad\square$

*3. Combining global minima of each component.* From Lemma 2, it follows that

$$X_j = -\frac{1}{\left(\frac{n-1}{n} + (d+2)\frac{1}{n}\right)} E_{d+1,j}\,,$$

is the unique global minimum of $f_j$. Hence, $b$ and $A = [a_1\ a_1\ \cdots\ a_d]$ achieve the global minimum of $f(b, A) = \sum_{j=1}^d f_j(ba_j^\top)$ if they satisfy

$$ba_j^\top = -\frac{1}{\left(\frac{n-1}{n} + (d+2)\frac{1}{n}\right)} E_{d+1,j} \quad \text{for all } i = 1, 2, \ldots, d.$$

This can be achieve by the following choice:

$$b^\top = \mathbf{e}_{d+1}, \quad a_j = -\frac{1}{\left(\frac{n-1}{n} + (d+2)\frac{1}{n}\right)} \mathbf{e}_j \quad \text{for } i = 1, 2, \ldots, d\,,$$

where $\mathbf{e}_j$ is the $j$-th coordinate vector. This choice precisely corresponds to

$$b = \mathbf{e}_{d+1}, \quad A = -\frac{1}{\left(\frac{n-1}{n} + (d+2)\frac{1}{n}\right)} \begin{bmatrix} I_d \\ 0 \end{bmatrix}.$$

We next move on to the non-isotropic case.

### A.3  Proof for the non-isotropic case

*1. Diagonal covariance case.* We first consider the case where $x^{(i)}$ is sampled from $\mathcal{N}(0, \Lambda)$ where $\Lambda = \mathrm{diag}(\lambda_1, \ldots, \lambda_d)$ and $w_\star$ is sampled from $\mathcal{N}(0, I_d)$. We prove the following generalization of Lemma 2.

**Lemma 3.** *Suppose that $x^{(i)}$ is sampled from $\mathcal{N}(0, \Lambda)$ where $\Lambda = \mathrm{diag}(\lambda_1, \ldots, \lambda_d)$ and $w_\star$ is sampled from $\mathcal{N}(0, I_d)$. Consider the following objective*

$$f_j(X) = \mathbb{E}_{Z_0, w_\star} \left[ \langle \mathsf{G}, X \rangle + w_\star[j] \right]^2.$$

*Then a global minimum is given as*

$$X_j = -\frac{1}{\frac{n+1}{n}\lambda_j + \frac{1}{n} \cdot \left(\sum_k \lambda_k\right)} E_{d+1,j}\,,$$

*where $E_{i_1,i_2}$ is the matrix whose $(i_1, i_2)$-th entry is 1, and the other entries are zero.*

**Proof of Lemma 3.** Similarly to the proof of Lemma 2, it suffices to check that

$$2\,\mathbb{E}\left[\langle \mathsf{G}, X_0 \rangle \mathsf{G}\right] + 2\,\mathbb{E}\left[w_\star[j]\, \mathsf{G}\right] = 0\,,$$

where we recall that $\mathsf{G}$ is defined as

$$\mathsf{G} = \frac{1}{n} \sum_i \begin{bmatrix} x^{(i)} x^{(i)\top} & y^{(i)} x^{(i)} \\ y^{(i)} x^{(i)\top} & y^{(i)2} \end{bmatrix}.$$

A similar calculation as the proof of Lemma 2 yields

$$\mathbb{E}\left[w_\star[j]\,\mathsf{G}\right] = \lambda_j (E_{d+1,j} + E_{j,d+1}). \tag{20}$$

Here the factor of $\lambda_j$ comes from the following generalization of (16):

$$\mathbb{E}[w_\star[j]\, y^{(i)}\, x^{(i)}[k]] = \mathbb{E}[w_\star[j]\, \langle w_\star, x^{(i)} \rangle\, x^{(i)}[k]] = \mathbb{E}\left[w_\star[j]^2\, x^{(i)}[j]\, x^{(i)}[k]\right] = \lambda_j \mathbb{1}_{[j=k]}\,.$$

Next, we compute $\mathbb{E}\left[\langle \mathsf{G}, E_{d+1,j} \rangle \mathsf{G}\right]$. Again, we follow a similar calculation to the proof of Lemma 2 except that this time we use the following generalization of (18):

$$\mathbb{E}\left[\langle w_\star, x^{(i)} \rangle\, x^{(i)}[j]\, \langle w_\star, x^{(i')} \rangle\, x^{(i')}[j]\right] = \begin{cases} \mathbb{E}[x^{(i)}[j]^2\, x^{(i')}[j]^2] = \lambda_j^2 & \text{if } i \neq i', \\ \mathbb{E}\left[\langle w_\star, x^{(i)} \rangle^2\, x^{(i)}[j]^2\right] = \lambda_j \sum_k \lambda_k + 2\lambda_j^2 & \text{if } i = i', \end{cases}$$

where the last line follows since

$$\mathbb{E}\left[\langle w_\star, x^{(i)}\rangle^2 \, x^{(i)}[j]^2\right] = \mathbb{E}\left[\|x^{(i)}\|^2 \, x^{(i)}[j]^2\right] = \mathbb{E}\left[x^{(i)}[j]^2 \sum_k x^{(i)}[k]^2\right] = \lambda_j \sum_k \lambda_k + 2\lambda_j^2.$$

Therefore, we have

$$\mathbb{E}\left[\langle \mathsf{G}, E_{d+1,j}\rangle \, \mathsf{G}\right] = \frac{1}{n^2} \cdot \left(n(n-1)\lambda_j^2 + n\lambda_j \sum_k \lambda_k + 2n\lambda_j^2\right)(E_{d+1,j} + E_{j,d+1})$$

$$= \left(\frac{n+1}{n}\lambda_j^2 + \frac{1}{n}(\lambda_j \sum_k \lambda_k)\right)(E_{d+1,j} + E_{j,d+1}). \tag{21}$$

Therefore, combining (20) and (21), the results follows. □

Now we finish the proof. From Lemma 2, it follows that

$$X_j = -\frac{1}{\frac{n+1}{n}\lambda_j + \frac{1}{n} \cdot (\sum_k \lambda_k)} E_{d+1,j}$$

is the unique global minimum of $f_j$. Hence, $b$ and $A = [a_1 \, a_1 \, \cdots \, a_d]$ achieve the global minimum of $f(b, A) = \sum_{j=1}^d f_j(b, A_j)$ if they satisfy

$$ba_j^\top = X_j = -\frac{1}{\frac{n+1}{n}\lambda_j + \frac{1}{n} \cdot (\sum_k \lambda_k)} E_{d+1,j} \quad \text{for all } i = 1, 2, \ldots, d.$$

This can be achieve by the following choice:

$$b^\top = \mathbf{e}_{d+1}, \quad a_j = -\frac{1}{\frac{n+1}{n}\lambda_j + \frac{1}{n} \cdot (\sum_k \lambda_k)} \mathbf{e}_j \quad \text{for } i = 1, 2, \ldots, d,$$

where $\mathbf{e}_j$ is the $j$-th coordinate vector. This choice precisely corresponds to

$$b = \mathbf{e}_{d+1}, \quad A = -\begin{bmatrix} \operatorname{diag}\left(\left\{\frac{1}{\frac{n+1}{n}\lambda_j + \frac{1}{n} \cdot (\sum_k \lambda_k)}\right\}_j\right) \\ 0 \end{bmatrix}.$$

***2. Non-diagonal covariance case (the setting of Theorem 1).*** We finally prove the general result of Theorem 1, namely $x^{(i)}$ is sampled from a Gaussian with covariance $\Sigma = U\Lambda U^\top$ where $\Lambda = \operatorname{diag}(\lambda_1, \ldots, \lambda_d)$ and $w_\star$ is sampled from $\mathcal{N}(0, I_d)$. The proof works by reducing this case to the previous case. For each $i$, define $\widetilde{x}^{(i)} := U^T x^{(i)}$. Then $\mathbb{E}[\widetilde{x}^{(i)}(\widetilde{x}^{(i)})^\top] = \mathbb{E}[U^\top (U\Lambda U^\top) U] = \Lambda$. Now let us write the loss function (15) with this new coordinate system: since $x^{(i)} = U\widetilde{x}^{(i)}$, we have

$$f(b, A) = \mathbb{E}_{Z_0, w_\star}\left[(b^\top \mathsf{G} A + w_\star^\top)U\widetilde{x}^{(n+1)}\right]^2 = \sum_{j=1}^d \lambda_j \mathbb{E}_{Z_0, w_\star}\left[((b^\top \mathsf{G} A + w_\star^\top)U)\,[j]\right]^2.$$

Hence, let us consider the vector $(b^\top \mathsf{G} A + w_\star^\top)U$. By definition of $\mathsf{G}$, we have

$$(b^\top \mathsf{G} A + w_\star^\top)U = \frac{1}{n}\sum_i b^\top \begin{bmatrix} x^{(i)} \\ \langle x^{(i)}, w_\star\rangle \end{bmatrix}^{\otimes 2} AU + w_\star^\top U$$

$$= \frac{1}{n}\sum_i b^\top \begin{bmatrix} U\widetilde{x}_i \\ \langle Ux^{(i)}, w_\star\rangle \end{bmatrix}^{\otimes 2} AU + w_\star^\top U$$

$$= \frac{1}{n}\sum_i b^\top \begin{bmatrix} U & 0 \\ 0 & 1 \end{bmatrix} \begin{bmatrix} \widetilde{x}_i \\ \langle Ux^{(i)}, w_\star\rangle \end{bmatrix}^{\otimes 2} \begin{bmatrix} U^\top & 0 \\ 0 & 1 \end{bmatrix} AU + w_\star^\top U$$

$$= \frac{1}{n}\sum_i \widetilde{b}^\top \begin{bmatrix} \widetilde{x}_i \\ \langle x^{(i)}, \widetilde{w}_\star\rangle \end{bmatrix}^{\otimes 2} \widetilde{A} + \widetilde{w}_\star^\top$$

where we define $\widetilde{b}^\top := b^\top \begin{bmatrix} U & 0 \\ 0 & 1 \end{bmatrix}$, $\widetilde{A} := \begin{bmatrix} U^\top & 0 \\ 0 & 1 \end{bmatrix} AU$, and $\widetilde{w}_\star := U^\top w_\star$. By the rotational symmetry, $\widetilde{w}_\star$ is also distributed as $\mathcal{N}(0, I_d)$. Hence, this reduces to the previous case, and a global minimum is given as

$$\widetilde{b} = \mathbf{e}_{d+1}, \quad \widetilde{A} = - \begin{bmatrix} \mathrm{diag}\left(\left\{\frac{1}{\frac{n+1}{n}\lambda_j + \frac{1}{n}\cdot(\sum_k \lambda_k)}\right\}_j\right) \\ 0 \end{bmatrix}.$$

From the definition of $\widetilde{b}, \widetilde{A}$, it thus follows that a global minimum is given by

$$b^\top = \mathbf{e}_{d+1}, \quad A = - \begin{bmatrix} U\mathrm{diag}\left(\left\{\frac{1}{\frac{n+1}{n}\lambda_i + \frac{1}{n}\cdot(\sum_k \lambda_k)}\right\}_i\right)U^\top \\ 0 \end{bmatrix},$$

as desired.

### A.4 Proof for non-linear attentions (Theorem 5)

As mentioned in Theorem 5, we focus on the setting where the last row of $Q$ is zero, i.e., let

$$Q = \begin{bmatrix} A & a \\ 0^\top & 0 \end{bmatrix} \quad \text{for } A \in \mathbb{R}^{d\times d} \text{ and } a \in \mathbb{R}^d.$$

We first rewrite the loss function and simplify it following Subsection A.1. Moreover, for simple notation we will often write $z, x$ instead of $z^{(n+1)}, x^{(n+1)}$.

#### *1. Rewriting loss function.*

Following Subsection A.1, let us write down the in-context loss (5). for the single-layer nonlinear attention denoted by $f(P, Q)$:

$$f(P, Q) = \mathbb{E}_{Z_0, w_\star}\left(\left[Z_0 + \frac{1}{n}\mathrm{Attn}^\sigma_{P,Q}(Z_0)\right]_{d+1,n+1} + w_\star^\top x\right)^2$$

Recalling the definition of the ReLU attention $\mathrm{Attn}^\sigma_{P,Q}(Z) := PZM\,\sigma(Z^\top QZ)$, the data matrix $Z := [z^{(1)} \cdots z^{(n+1)}]$, and the mask matrix $M$, the term $\mathrm{Attn}^\sigma_{P,Q}(Z_0)$ can be written as:

$$\mathrm{Attn}^\sigma_{P,Q}(Z_0) = \underbrace{PZ_0}_{\mathbb{R}^{(d+1)\times(n+1)}} \cdot \begin{bmatrix} I_{n\times n} & 0 \\ 0 & 0 \end{bmatrix} \cdot \underbrace{\sigma\left(Z_0^\top QZ_0\right)}_{\mathbb{R}^{(n+1)\times(n+1)}}.$$

Hence, it follows that the $(d + 1, n + 1)$-th entry of $\mathrm{Attn}^\sigma_{P,Q}(Z_0)$ is equal to the product of the $(d + 1)$-th row of $PZ_0$, the mask matrix $M$, and the $(n + 1)$-th column of $\sigma\left(Z_0^\top QZ_0\right)$. Hence, let us write them down explicitly:

- Letting $b^\top$ be the last row of the matrix $P$, it holds that the $(d + 1)$-th row of $PZ_0$ is equal to $[\langle b, z^{(i)}\rangle]_{i=1,\ldots,n+1}$.
- The $(n + 1)$-th column of $\sigma\left(Z_0^\top QZ_0\right)$ is equal to $\left[\sigma\left((z^{(i)})^\top Qz^{(n+1)}\right)\right]_{i=1,\ldots,n+1}$. Letting $A$ the first $d$ columns of $Q$, this vector is equal to $\left[\sigma\left((x^{(i)})^\top Ax^{(n+1)}\right)\right]_{i=1,\ldots,n+1}$ because the last row of $Q$ is zero and the last row of $z^{(n+1)}$ is zero (since $(z^{(n+1)})^\top = [(x^{(n+1)})^\top\ 0]$).

Thus, the product of $[\langle b, z^{(i)}\rangle]_{i=1,\ldots,n+1}$, the mask matrix $M$, and $\left[\sigma\left((x^{(i)})^\top Ax^{(n+1)}\right)\right]_{i=1,\ldots,n+1}$ results in the following expression of the attention (writing $z, x$ instead of $z^{(n+1)}, x^{(n+1)}$):

$$\left[\mathrm{Attn}^\sigma_{P,Q}(Z_0)\right]_{d+1,n+1} = \sum_{i=1}^n \left[\langle b, z^{(i)}\rangle \cdot \sigma((x^{(i)})^\top Ax)\right].$$

Since $[Z_0]_{d+1,n+1} = 0$, we therefore have

$$\left[ Z_0 + \frac{1}{n} \mathrm{Attn}_{P,Q}^\sigma(Z_0) \right]_{d+1,n+1} = \frac{1}{n} \sum_{i=1}^{n} \left[ \langle b, z^{(i)} \rangle \cdot \sigma((x^{(i)})^\top A x) \right].$$

Therefore, it follows that the in-context loss $f(P, Q)$ only depends on $b$ and $A$. Henceforth, let us write $f(b, A)$ instead of $f(P, Q)$ following Subsection A.1. In particular, writing $b^\top = [b_0^\top, b_1]$ for $b_0 \in \mathbb{R}^d$ and $b_1 \in \mathbb{R}$, the loss function can be expressed as

$$f(b, A) := \mathbb{E} \left( \frac{1}{n} \sum_{i=1}^{n} \left[ (\langle b_0, x^{(i)} \rangle + b_1 y^{(i)}) \cdot \sigma((x^{(i)})^\top A x) \right] + \langle w_\star, x \rangle \right)^2. \tag{22}$$

## 2. Simplifying the loss function with symmetry.

Now, we use the fact that both $x^{(i)}$'s and $w_\star$ are sampled from the isotropic Gaussian, i.e., $\mathcal{N}(0, I_d)$ in order to further simplify the loss function in (22). In particular, we use the following facts:

(a) For orthonormal matrices $U, V \in \mathbb{R}^{d \times d}$, it holds that $Ux^{(i)}$, $Vx$ and $Uw_\star$ have the same distributions as $\mathcal{N}(0, I_d)$.

(b) Moreover, for a diagonal matrix $\Xi = \mathrm{diag}(\xi_i) \in \mathbb{R}^{d \times d}$ with the diagonal entries being random signs $\xi_i \sim \{\pm 1\}$, it holds that $\Xi x^{(i)}$, $\Xi x$ and $\Xi w_\star$ have the same distributions as $\mathcal{N}(0, I_d)$.

Now let us fix a matrix $A \in \mathbb{R}^{d \times d}$ and $b^\top = [b_0^\top, b_1]$ for $b_0 \in \mathbb{R}^d$ and $b_1 \in \mathbb{R}$. Letting $A = U \Sigma V^\top$ be the SVD of the matrix $A$, it follows that

$$f(b, A) = \mathbb{E} \left[ \frac{1}{n} \sum_{i=1}^{n} \left[ (b_0^\top x^{(i)} + b_1 w_\star^\top x^{(i)}) \cdot \sigma((x^{(i)})^\top A x) \right] + w_\star^\top x \right]^2$$

$$\overset{(a)}{=} \mathbb{E} \left[ \frac{1}{n} \sum_{i=1}^{n} \left[ (b_0^\top U x^{(i)} + b_1 w_\star^\top U^\top U x^{(i)}) \cdot \sigma((x^{(i)})^\top U^\top A V x) \right] + w_\star^\top U^\top V x \right]^2$$

$$\overset{(b)}{=} \mathbb{E} \left[ \frac{1}{n} \sum_{i=1}^{n} \left[ (b_0^\top U \Xi x^{(i)} + b_1 w_\star^\top x^{(i)}) \cdot \sigma((x^{(i)})^\top \Sigma x) \right] + w_\star^\top \Xi U^\top V \Xi x \right]^2$$

$$\geq \mathbb{E} \left[ \mathbb{E}_\Xi \left\{ \frac{1}{n} \sum_{i=1}^{n} \left[ (b_0^\top U \Xi x^{(i)} + b_1 w_\star^\top x^{(i)}) \cdot \sigma((x^{(i)})^\top \Sigma x) \right] + w_\star^\top \Xi U^\top V \Xi x \right\} \right]^2$$

$$= \mathbb{E} \left[ \frac{1}{n} \sum_{i=1}^{n} \left[ b_1 w_\star^\top x^{(i)} \cdot \sigma((x^{(i)})^\top \Sigma x) \right] + w_\star^\top \mathrm{diag}(U^\top V) x \right]^2$$

$$=: f_{\mathsf{lower}}(b_1, \Sigma, D := \mathrm{diag}(U^\top V)).$$

where in the third line we use the fact that $\Xi^\top \Sigma \Xi = \Xi \Sigma \Xi = \Sigma$; and the fourth line follows from the Jensen's inequality. Hence for the remainder of the proof, we will characterize the global minimizer of the lower bound, i.e., $f_{\mathsf{lower}}$ and then we will connect it back to the original objective.

## 3. Computation of the lower bound $f_{\mathsf{lower}}$.

Let us now explicitly compute $f_{\mathsf{lower}}$. Let us rewrite the definition of $f_{\mathsf{lower}}$. In fact since, $\sigma = \mathrm{ReLU}$ is homogenous, one can further simplify the lower bound by pushing the constant $b_1$ inside and write $b_1 \Sigma$ as $\Sigma$. Hence, for two diagonal matrices $\Sigma, D \in \mathbb{R}^{d \times d}$, $f_{\mathsf{lower}}$ is defined as:

$$f_{\mathsf{lower}}(\Sigma, D) := \mathbb{E} \left[ \frac{1}{n} \sum_{i=1}^{n} \left[ \langle w_\star, x^{(i)} \rangle \cdot \sigma(x^\top \Sigma x^{(i)}) \right] + w_\star^\top D x \right]^2.$$

In particular, $D$ is constrained to be the diagonal part of an orthogonal matrix (since $D = \mathrm{diag}(U^\top V)$ in the above derivation). Now we focus on characterizing the global minimizers of $f_{\mathsf{lower}}$.

The main part of the argument is inspired by the elegant observation of Erdogdu et al. (2016), which says that the solution of least squares and generalized linear models are collinear for Gaussian inputs. We leverage the same proof technique (*à la* Stein's Lemma) to prove that the presence of ReLU only changes the scaling of global optimum.

First, since $w_\star$ is isotropic Gaussian, we can take the expectation over $w_\star$ to obtain

$$f_{\text{lower}}(\Sigma, D) = \mathbb{E}\left[\frac{1}{n}\sum_{i=1}^{n}\left[\sigma(x^\top \Sigma x^{(i)})\, x^{(i)}\right] + Dx\right]^2,$$

which after a careful expansion becomes

$$\mathbb{E}\left[\frac{1}{n^2}\sum_{i,j}\sigma(x^\top \Sigma x^{(i)})\sigma(x^\top \Sigma x^{(j)})\langle x^{(i)}, x^{(j)}\rangle + \frac{2}{n}\sum_i \sigma(x^\top \Sigma x^{(i)})\langle x^{(i)}, Dx\rangle\right] + \text{const.} \quad (23)$$

In order to compute (23), we will rely on the aforementioned argument of Erdogdu et al. (2016). In particular, from integration by parts, or Stein's lemma (Erdogdu et al., 2016) (since $x \sim \mathcal{N}(0, I_d)$), we have

$$\mathbb{E}_x[\sigma(x^\top v)x] = \mathbb{E}_x[\sigma'(x^\top v)]v \quad \text{for a fixed } v \in \mathbb{R}^d.$$

We use this to compute all the terms in (23) as follows:

- We first apply Stein's lemma to the first term of (23) for $i \neq j$. This results in

$$\mathbb{E}_{x^{(i)}, x^{(j)}, x}\,\sigma(x^\top \Sigma x^{(i)})\sigma(x^\top \Sigma x^{(j)})\langle x^{(i)}, x^{(j)}\rangle$$
$$= \mathbb{E}_{x^{(j)}, x}[\mathbb{E}_{x^{(i)}}[\sigma'(x^\top \Sigma x^{(i)})]\,\sigma(x^\top \Sigma x^{(j)})\,x^\top \Sigma x^{(j)}]$$

  Using the fact that $x^{(i)}$ is a symmetric random variable, one can compute the expectation above as follows: one the one hand, we know $\mathbb{E}_{x^{(i)}}[\sigma'(x^\top \Sigma x^{(i)})] = \mathbb{E}_{x^{(i)}}[\sigma'(-x^\top \Sigma x^{(i)})]$. On the other hand, we also know that for any scalar $\alpha$, $\sigma'(-\alpha) + \sigma'(\alpha) = 1$. Therefore, we conclude that $\mathbb{E}_{x^{(i)}}[\sigma'(x^\top \Sigma x^{(i)})] = 1/2$. Thus, applying this technique twice, we obtain the following

$$\mathbb{E}_{x^{(i)}, x^{(j)}, x}\,\sigma(x^\top \Sigma x^{(i)})\sigma(x^\top \Sigma x^{(j)})\langle x^{(i)}, x^{(j)}\rangle = \frac{1}{4}\mathbb{E}_x[x^\top \Sigma^2 x] = \frac{1}{4}\operatorname{Tr}(\Sigma^2).$$

- Similarly, we can use Stein's lemma to the second term of (23) to conclude

$$\mathbb{E}\,\sigma(x^\top \Sigma x^{(i)})\langle x^{(i)}, Dx\rangle = \frac{1}{2}\,\mathbb{E}_x\,x^\top \Sigma Dx = \frac{1}{2}\operatorname{Tr}(\Sigma D).$$

- Lastly, the computation of the first term of (23) for $i = j$ is straightforward. Using the fact that $\forall \alpha \in \mathbb{R}, \sigma^2(\alpha) + \sigma^2(-\alpha) = a^2$, we get

$$\mathbb{E}\left[\sigma^2(x^\top \Sigma x^{(i)})\,\|x^{(i)}\|^2\right] = \frac{1}{2}\mathbb{E}[(x^\top \Sigma x^{(i)})^2\,\|x^{(i)}\|^2]$$
$$= \frac{1}{2}\,\mathbb{E}\left[(x^{(i)})^\top \Sigma^2 x^{(i)}\,\|x^{(i)}\|^2\right] = \frac{d+2}{2}\operatorname{Tr}(\Sigma^2).$$

Putting things all together (and ignoring the constant part in (23)), we have

$$f_{\text{lower}}(\Sigma, D) = \frac{2(d+2) + (n-1)}{4n}\operatorname{Tr}(\Sigma^2) + \operatorname{Tr}(\Sigma D). \quad (24)$$

### *4. Connecting back to the original loss function.*

One can in fact write (24) solely in terms of $b_1$ and $A$ as follows:

$$\frac{2(d+2) + (n-1)}{4n}\operatorname{Tr}(\Sigma^2) + \operatorname{Tr}(\Sigma D) = \frac{2(d+2) + (n-1)}{4n}\|b_1 A\|_F^2 + \operatorname{Tr}(b_1 A).$$

Since the latter is a convex function in the matrix $b_1 A$, it follows that the minimizer corresponds to

$$b_1 A = -\frac{2n}{2(d+2) + (n-1)} \cdot I_d$$

In fact the choice $b_1 = 1$, $b_0 = 0$, and $A = -\frac{2n}{2(d+2)+(n-1)} \cdot I_d$ achieves this, and more crucially, satisfies the property that $f_{\text{lower}} = f$ for the corresponding parameters. Therefore, this shows that such choice is a global minimizer.

# B Proofs for the multi-layer case

## B.1 Proof of Theorem 2

The proof is based on probabilistic methods (Alon and Spencer, 2016). According to Lemma 5, the objective function can be written as (for more details check the derivations in (25))

$$f(A_1, A_2) = \mathbb{E} \operatorname{Tr} \left( \mathbb{E} \left[ \prod_{i=1}^{2} (I - X_0^\top A_i X_0 M) X_0^\top w_\star w_\star^\top X_0 \prod_{i=1}^{2} (I - M X_0^T A_i X_0) \right] \right)$$

$$= \mathbb{E} \operatorname{Tr} \left( \mathbb{E} \left[ \prod_{i=2}^{1} (I - X_0^\top A_i X_0 M) X_0^\top X_0 \prod_{j=1}^{2} (I - M X_0^T A_j X_0) \right] \right),$$

where we use the isotropy of $w_\star$ and the linearity of trace to get the last equation. Suppose that $A_0^*$ and $A_1^*$ denote the global minimizer of $f$ over symmetric matrices. Since $A_1^*$ is a symmetric matrix, it admits the spectral decomposition $A_1 = U D_1 U^\top$ where $D_1$ is a diagonal matrix and $U$ is an orthogonal matrix. Remarkably, the distribution of $X_0$ is invariant to a linear transformation by an orthogonal matrix, i.e, $X_0$ has the same distribution as $X_0 U^\top$. This invariance yields

$$f(U D_1 U^\top, A_2^*) = f(D_1, U^\top A_2^* U).$$

Thus, we can assume $A_1^*$ is diagonal without loss of generality. To prove $A_2^*$ is also diagonal, we leverage a probabilistic proof technique. Consider the random diagonal matrix $S$ whose diagonal elements are either 1 or $-1$ with probability $\frac{1}{2}$. Since the input distribution is invariant to orthogonal transformations, we have

$$f(D_1, A_2^*) = f(S D_1 S, S A_2^* S) = f(D_1, S A_2^* S).$$

Note that we use $S D_1 S = D_1$ in the last equation, which holds due to $D_1$ and $S$ are diagonal matrices and $S$ has diagonal elements in $\{+1, -1\}$. Since $f$ is convex in $A_2$, a straightforward application of Jensen's inequality yields

$$f(D_1, A_2^*) = \mathbb{E} \left[ f(D_1, S A_2^* S) \right] \geq f(D_1, \mathbb{E} \left[ S A_2^* S \right]) = f(D_1, \operatorname{diag}(A_2^*)).$$

Thus, there are diagonal $D_1$ and $\operatorname{diag}(A_2^*)$ for which $f(D_1, \operatorname{diag}(A_2^*)) \leq f(A_1^*, A_2^*)$ holds for an optimal $A_1^*$ and $A_2^*$. This concludes the proof.

## B.2 Proof of Theorem 3

Let us drop the factor of $1/n$ which was present in the original update (56). This is because the constant $1/n$ can be absorbed into $A_i$'s. Doing so does not change the theorem statement, but reduces notational clutter.

Let us consider the reformulation of the in-context loss $f$ presented in Lemma 5. Specifically, let $\overline{Z}_0$ be defined as

$$\overline{Z}_0 = \begin{bmatrix} x^{(1)} & x^{(2)} & \cdots & x^{(n)} & x^{(n+1)} \\ y^{(1)} & y^{(2)} & \cdots & y^{(n)} & y^{(n+1)} \end{bmatrix} \in \mathbb{R}^{(d+1) \times (n+1)},$$

where $y^{(n+1)} = \langle w_\star, x^{(n+1)} \rangle$. Let $\overline{Z}_i$ denote the output of the $(i-1)^{th}$ layer of the linear transformer (as defined in (56), initialized at $\overline{Z}_0$). For the rest of this proof, we will drop the bar, and simply denote $\overline{Z}_i$ by $Z_i$.[5] Let $X_i \in \mathbb{R}^{d \times (n+1)}$ denote the first $d$ rows of $Z_i$ and let $Y_i \in \mathbb{R}^{1 \times (n+1)}$ denote the $(d+1)^{th}$ row of $Z_k$. Under the sparsity pattern enforced in (8), we verify that, for any $i \in \{0, \ldots, k\}$,

$$X_i = X_0,$$

$$Y_{i+1} = Y_i + Y_i M X_i^\top A_i X_i = Y_0 \prod_{\ell=0}^{i} \left( I + M X_0^\top A_\ell X_0 \right). \tag{25}$$

---

[5]This use of $Z_i$ differs the original definition in (1). But we will not refer to the original definition anywhere in this proof.

where $M = \begin{bmatrix} I_{n\times n} & 0 \\ 0 & 0 \end{bmatrix}$. We adopt the shorthand $A = \{A_i\}_{i=0}^k$.

We adopt the shorthand $A = \{A_i\}_{i=0}^k$. Let $\mathcal{S} \subset \mathbb{R}^{(k+1)\times d\times d}$, and $A \in \mathcal{S}$ if and only if for all $i \in \{0, \ldots, k\}$, there exists scalars $a_i \in \mathbb{R}$ such that $A_i = a_i \Sigma^{-1}$ and $B_i = b_i I$. We use $f(A)$ to refer to the in-context loss of Theorem 3, that is,

$$f(A) := f\left(\left\{Q_i = \begin{bmatrix} A_i & 0 \\ 0 & 0 \end{bmatrix}, P_i = \begin{bmatrix} 0_{d\times d} & 0 \\ 0 & 1 \end{bmatrix}\right\}_{i=0}^k\right).$$

Throughout this proof, we will work with the following formulation of the *in-context loss* from Lemma 5:

$$f(A) = \mathbb{E}_{(X_0, w_\star)}\left[\text{Tr}\left((I - M) Y_{k+1}^\top Y_{k+1} (I - M)\right)\right]. \tag{26}$$

The theorem statement is equivalent to the following:

$$\inf_{A\in\mathcal{S}} \sum_{i=0}^k \|\nabla_{A_i} f(A)\|_F^2 = 0, \tag{27}$$

where $\nabla_{A_i} f$ denotes derivative wrt the Frobenius norm $\|A_i\|_F$. Towards this end, we establish the following intermediate result: if $A \in \mathcal{S}$, then for any $R \in \mathbb{R}^{(k+1)\times d\times d}$, there exists $\tilde{R} \in \mathcal{S}$, such that, at $t = 0$,

$$\frac{d}{dt} f(A + t\tilde{R}) \leq \frac{d}{dt} f(A + tR). \tag{28}$$

In fact, we show that $\tilde{R}_i := r_i I$, for $r_i = \frac{1}{d} \text{Tr}\left(\Sigma^{1/2} R_i \Sigma^{1/2}\right)$. This implies (27) via the following simple argument: Consider the "$\mathcal{S}$-constrained gradient flow": let $A(t) : \mathbb{R}^+ \to \mathbb{R}^{(k+1)\times d\times d}$ be defined as

$$\frac{d}{dt} A_i(t) = -r_i(t)\Sigma^{-1}, \quad r_i(t) := \text{Tr}(\Sigma^{1/2}\nabla_{A_i} f(A(t))\Sigma^{1/2})$$

for $i = 0, \ldots, k$. By (28), we verify that

$$\frac{d}{dt} f(A(t)) \leq -\sum_{i=0}^k \|\nabla_{A_i} f(A(t))\|_F^2. \tag{29}$$

We verify from its definition that $f(A) \geq 0$; if the infimum in (27) fails to be zero, then inequality (29) will ensure unbounded descent as $t \to \infty$, contradicting the fact that $f(A)$ is lower-bounded. This concludes the proof.

***Proof outline.*** The remainder of the proof will be devoted to showing (28), which we outline as follows:

- In Step 1, we reduce the condition in (29) to a more easily verified *layer-wise* condition. Specifically, we only need to verify (29) when $R_i$ are all zero except for $R_j$ for some fixed $j$ (see (30))

  At the end of Step 1, we set up some additional notation, and introduce an important matrix $G$, which is roughly "a product of attention layer matrices". In (31), we study the evolution of $f(A(t))$ when $A(t)$ moves in the direction of $R$, as $X_0$ is (roughly speaking) randomly transformed.

- In Step 2, we use the results of Step 2 to to study $G$ (see (32)) and $\frac{d}{dt} G(A(t))$ (see (33)) under random transformation of $X_0$. The idea in (33) is that "randomly transforming $X_0$" has the same effect as "randomly transforming $S$" (recall $S$ is the perturbation to $B$).

- In Step 3, we apply the result from Step 2 to the expression of $\frac{d}{dt} f(A(t))$ in (31). We verify that $\tilde{R}$ in (28) is exactly the expected matrix after "randomly transforming $S$". This concludes our proof.

***1. Reduction to layer-wise condition.*** To prove (28), it suffices to show the following simpler condition: Let $j \in \{0, \ldots, k\}$. Let $R_j \in \mathbb{R}^{d\times d}$ be arbitrary matrices. For $C \in \mathbb{R}^{d\times d}$, let $A(tC, j)$ denote

the collection of matrices, where $[A(tC, j)]_j = A_j + tC$, and for $i \neq j$, $A(tC, j)_i = A_i$. We show that for all $j \in \{0, \ldots, k\}$, $R_j \in \mathbb{R}^{d \times d}$, there exists $\tilde{R}_j = r_j \Sigma^{-1}$, such that, at $t = 0$,

$$\frac{d}{dt} f(A(t\tilde{R}_j, j)) \leq \frac{d}{dt} f(A(tR_j, j)) \tag{30}$$

We can verify that (28) is equivalent to (30) by noticing that for any $R$, at $t = 0$, $\frac{d}{dt} f(A + tR) = \sum_{j=0}^{k} \frac{d}{dt} f(A(tR_j, j))$. We will now work towards proving (30) for some index $j$ that is arbitrarily chosen but fixed throughout.

Let us define, for any $C \in \mathbb{R}^{d \times d}$, $G(X, A_j + C) := X \prod_{i=0}^{k} \left( I - M X^\top [A(C, j)]_i X \right)$. By (25) and (26),

$$
\begin{aligned}
&f(A(tR_j, j)) \\
&= \mathbb{E}\left[ \mathrm{Tr}\left( (I - M) Y_{k+1}^\top Y_{k+1} (I - M) \right) \right] \\
&= \mathbb{E}\left[ \mathrm{Tr}\left( (I - M) G(X_0, A_j + tR_j)^\top w_\star^\top w_\star G(X_0, A_j + tR_j) (I - M) \right) \right] \\
&= \mathbb{E}\left[ \mathrm{Tr}\left( (I - M) G(X_0, A_j + tR_j)^\top \Sigma^{-1} G(X_0, A_j + tR_j) (I - M) \right) \right]
\end{aligned}
$$

The second equality follows from plugging in (25). For the rest of this proof, let $U$ denote a uniformly randomly sampled orthogonal matrix. Let $U_\Sigma := \Sigma^{1/2} U \Sigma^{-1/2}$. Using the fact that $X_0 \overset{d}{=} U_\Sigma X_0$, we can verify

$$
\begin{aligned}
&\left. \frac{d}{dt} f(A(tR_j, j)) \right|_{t=0} \\
&= \left. \frac{d}{dt} \mathbb{E}\left[ \mathrm{Tr}\left( (I - M) G(X_0, A_j + tR_j)^\top \Sigma^{-1} G(X_0, A_j + tR_j) (I - M) \right) \right] \right|_{t=0} \\
&= \left. \frac{d}{dt} \mathbb{E}_{X_0, U}\left[ \mathrm{Tr}\left( (I - M) G(U_\Sigma X_0, A_j + tR_j)^\top \Sigma^{-1} G(U_\Sigma X_0, A_j + tR_j) (I - M) \right) \right] \right|_{t=0} \\
&= 2 \mathbb{E}_{X_0, U}\left[ \mathrm{Tr}\left( (I - M) G(U_\Sigma X_0, A_j)^\top \Sigma^{-1} \left. \frac{d}{dt} G(U_\Sigma X_0, A_j + tR_j) \right|_{t=0} (I - M) \right) \right]. \quad (31)
\end{aligned}
$$

**2. $G$ and $\frac{d}{dt} G$ under random transformation of $X_0$.** We will now verify that $G(U_\Sigma X_0, A_j) = U_\Sigma G(X_0, A_j)$:

$$
\begin{aligned}
&G(U_\Sigma X_0, A_j) \\
&= U_\Sigma X_0 \prod_{i=0}^{k} \left( I + M X_0^T U_\Sigma^\top A_i U_\Sigma X_0 \right) \\
&= U_\Sigma G(X_0, A_j),
\end{aligned} \tag{32}
$$

where we use the fact that $U_\Sigma^\top A_i U_\Sigma = U_\Sigma^\top (a_i \Sigma^{-1}) U_\Sigma = A_i$. Next, we verify that

$$
\begin{aligned}
&\left. \frac{d}{dt} G(U_\Sigma X_0, A + tR_j) \right|_{t=0} \\
&= U_\Sigma X_0 \left( \prod_{i=0}^{j-1} (I + M X_0^T A_i X_0) \right) M X_0^T U_\Sigma^\top R_j U_\Sigma X_0 \prod_{i=j+1}^{k} (I + M X_0^T A_i X_0) \\
&= U_\Sigma \frac{d}{dt} G(X_0, A_j + t U_\Sigma^\top R_j U_\Sigma)
\end{aligned} \tag{33}
$$

where the first equality again uses the fact that $U_\Sigma^\top A_i U_\Sigma = A_i$.

**3. _Putting everything together._** Let us continue from (31). Plugging (32) and (33) into (31),

$$\left. \frac{d}{dt} f(A(tR_j, j)) \right|_{t=0}$$

$$= 2\, \mathbb{E}_{X_0, U}\left[\mathrm{Tr}\left((I-M)\,G(U_\Sigma X_0, A_j)^\top \Sigma^{-1}\, \frac{d}{dt} G(U_\Sigma X_0, A_j + tR_j)\Big|_{t=0}\,(I-M)\right)\right]$$

$$\overset{(i)}{=} 2\, \mathbb{E}_{X_0, U}\left[\mathrm{Tr}\left((I-M)\,G(X_0, A_j)^\top \Sigma^{-1}\, \frac{d}{dt} G(X_0, A_j + tU_\Sigma^\top R_j U_\Sigma)\Big|_{t=0}\,(I-M)\right)\right]$$

$$= 2\, \mathbb{E}_{X_0}\left[\mathrm{Tr}\left((I-M)\,G(X_0, A_j)^\top \Sigma^{-1}\, \mathbb{E}_U\left[\frac{d}{dt} G(X_0, A_j + tU_\Sigma^\top R_j U_\Sigma)\Big|_{t=0}\right]\,(I-M)\right)\right]$$

$$\overset{(ii)}{=} 2\, \mathbb{E}_{X_0}\left[\mathrm{Tr}\left((I-M)\,G(X_0, A_j)^\top \Sigma^{-1}\, \frac{d}{dt} G(X_0, A_j + t\,\mathbb{E}_U\left[U_\Sigma^\top R_j U_\Sigma\right])\Big|_{t=0}\,(I-M)\right)\right]$$

$$= 2\, \mathbb{E}_{X_0}\left[\mathrm{Tr}\left((I-M)\,G(X_0, A_j)^\top \Sigma^{-1}\, \frac{d}{dt} G(X_0, A_j + t\cdot r_j \Sigma^{-1})\Big|_{t=0}\,(I-M)\right)\right]$$

$$= \frac{d}{dt} f(A(t\cdot r_j \Sigma^{-1}, j))\Big|_{t=0},$$

where $r_j := \frac{1}{d}\,\mathrm{Tr}\left(\Sigma^{1/2} R_j \Sigma^{1/2}\right)$. In the above, $(i)$ uses 1. (32) and (33), as well as the fact that $U_\Sigma^\top \Sigma^{-1} U_\Sigma = \Sigma^{-1}$. $(ii)$ uses the fact that $\frac{d}{dt} G(X_0, A_j + tC)\big|_{t=0}$ is affine in $C$. To see this, one can verify from the definition of $G$, e.g. using similar algebra as (33), that $\frac{d}{dt} G(X_0, A_j + C)$ is affine in $C$. Thus $\mathbb{E}_U\left[G(X_0, A_j + tU_\Sigma^\top R_j U_\Sigma)\right] = G(X_0, A_j + t\,\mathbb{E}_U\left[U_\Sigma^\top R_j U_\Sigma\right])$.

### B.3   Proof of Theorem 4

The proof of Theorem 4 is similar to that of Theorem 3, and with a similar setup. However to keep the proof self-contained, we will restate the setup. Once again, we drop the factor of $\frac{1}{n}$ which was present in the original update (56). This is because the constant $1/n$ can be absorbed into $A_i$'s. Doing so does not change the theorem statement, but reduces notational clutter.

Let us consider the reformulation of the in-context loss $f$ presented in Lemma 5. Specifically, let $\overline{Z}_0$ be defined as

$$\overline{Z}_0 = \begin{bmatrix} x^{(1)} & x^{(2)} & \cdots & x^{(n)} & x^{(n+1)} \\ y^{(1)} & y^{(2)} & \cdots & y^{(n)} & y^{(n+1)} \end{bmatrix} \in \mathbb{R}^{(d+1)\times(n+1)},$$

where $y^{(n+1)} = \langle w_\star, x^{(n+1)}\rangle$. Let $\overline{Z}_i$ denote the output of the $(i-1)^{th}$ layer of the linear transformer (as defined in (56), initialized at $\overline{Z}_0$). For the rest of this proof, we will drop the bar, and simply denote $\overline{Z}_i$ by $Z_i$.[6] Let $X_i \in \mathbb{R}^{d\times n+1}$ denote the first $d$ rows of $Z_i$ and let $Y_i \in \mathbb{R}^{1\times n+1}$ denote the $(d+1)^{th}$ row of $Z_k$. Under the sparsity pattern enforced in (11), we verify that, for any $i \in \{0, \ldots, k\}$,

$$X_{i+1} = X_i + B_i X_i M X_i^\top A_i X_i$$

$$Y_{i+1} = Y_i + Y_i M X_i^\top A_i X_i = Y_0 \prod_{\ell=0}^{i} \left(I + M X_\ell^T A_\ell X_\ell\right). \tag{34}$$

We adopt the shorthand $A = \{A_i\}_{i=0}^k$ and $B = \{B_i\}_{i=0}^k$. Let $\mathcal{S} \subset \mathbb{R}^{2\times(k+1)\times d\times d}$, and $(A, B) \in \mathcal{S}$ if and only if for all $i \in \{0, \ldots, k\}$, there exists scalars $a_i, b_i \in \mathbb{R}$ such that $A_i = a_i \Sigma^{-1}$ and $B_i = b_i I$. Throughout this proof, we will work with the following formulation of the *in-context loss* from Lemma 5:

$$f(A, B) := \mathbb{E}_{(X_0, w_\star)}\left[\mathrm{Tr}\left((I-M)\,Y_{k+1}^\top Y_{k+1}\,(I-M)\right)\right]. \tag{35}$$

(note that the only randomness in $Z_0$ comes from $X_0$ as $Y_0$ is a deterministic function of $X_0$). The theorem statement is equivalent to the following:

$$\inf_{(A,B)\in\mathcal{S}} \sum_{i=0}^{k} \|\nabla_{A_i} f(A, B)\|_F^2 + \|\nabla_{B_i} f(A, B)\|_F^2 = 0 \tag{36}$$

---

[6] This use of $Z_i$ differs the original definition in (1). But we will not refer to the original definition anywhere in this proof.

where $\nabla_{A_i} f$ denotes derivative wrt the Frobenius norm $\|A_i\|_F$.

Our goal is to show that, if $(A, B) \in \mathcal{S}$, then for any $(R, S) \in \mathbb{R}^{2 \times (k+1) \times d \times d}$, there exists $(\tilde{R}, \tilde{S}) \in \mathcal{S}$, such that, at $t = 0$,

$$\frac{d}{dt} f(A + t\tilde{R}, B + t\tilde{S}) \leq \frac{d}{dt} f(A + tR, B + tS). \tag{37}$$

In fact, we show that $\tilde{R}_i := r_i I$, for $r_i = \frac{1}{d} \operatorname{Tr}\left(\Sigma^{1/2} R_i \Sigma^{1/2}\right)$ and $\tilde{S}_i = s_i I$, for $s_i = \frac{1}{d} \operatorname{Tr}\left(\Sigma^{-1/2} S_i \Sigma^{1/2}\right)$. This implies (36) via the following simple argument: Consider the "$\mathcal{S}$-constrained gradient flow": let $A(t) : \mathbb{R}^+ \to \mathbb{R}^{(k+1) \times d \times d}$ and $B(t) : \mathbb{R}^+ \to \mathbb{R}^{(k+1) \times d \times d}$ be defined as

$$\frac{d}{dt} A_i(t) = -r_i(t) \Sigma^{-1}, \quad r_i(t) := \operatorname{Tr}(\Sigma^{1/2} \nabla_{A_i} f(A(t), B(t)) \Sigma^{1/2})$$

$$\frac{d}{dt} B_i(t) = -s_i(t) \Sigma^{-1}, \quad s_i(t) := \operatorname{Tr}(\Sigma^{-1/2} \nabla_{B_i} f(A(t), B(t)) \Sigma^{1/2}),$$

for $i = 0, \ldots, k$. By (37), we verify that

$$\frac{d}{dt} f(A(t), B(t)) \leq -\left(\sum_{i=0}^{k} \|\nabla_{A_i} f(A(t), B(t))\|_F^2 + \|\nabla_{B_i} f(A(t), B(t))\|_F^2\right). \tag{38}$$

We verify from its definition that $f(A, B) \geq 0$; if (36) does not hold then (38) will ensure unbounded descent as $t \to \infty$, contradicting the fact that $f(A, B)$ is lower-bounded. This concludes the proof.

***Proof outline.*** The remainder of the proof will be devoted to showing (37), which we outline as follows:

- In Step 1, we reduce the condition in (37) to a more easily verified *layer-wise* condition. Specifically, we only need to verify (37) in one of the two cases: (I) when $R_i, S_i$ are all zero except for $R_j$ for some fixed $j$ (see (40)), or (II) when $R_i, S_i$ are all zero except for $S_j$ for some fixed $j$ (see (39)).
  We focus on the proof of (II), as the proof of (I) is almost identical. At the end of Step 1, we set up some additional notation, and introduce an important matrix $G$, which is roughly "a product of attention layer matrices". In (41), we study the evolution of $f(A, B(t))$ when $B(t)$ moves in the direction of $S$, as $X_0$ is (roughly speaking) randomly transformed. This motivates the subsequent analysis in Steps 2 and 3 below.

- In Step 2, we study how outputs of each layer (34) changes when $X_0$ is randomly transformed. There are two main results here: First we provide the expression for $X_i$ in (42). Second, we provide the expression for $\frac{d}{dt} X_i(B(t))$ in (43).

- In Step 3, we use the results of Step 2 to to study $G$ (see (47)) and $\frac{d}{dt} G(B(t))$ (see (48)) under random transformation of $X_0$.
  The idea in (48) is that "randomly transforming $X_0$" has the same effect as "randomly transforming $S$" (recall $S$ is the perturbation to $B$).

- In Step 4, we use the results from Steps 2 and 3 to the expression of $\frac{d}{dt} f(A, B(t))$ in (41). We verify that $\tilde{S}$ in (37) is exactly the expected matrix after "randomly transforming $S$". This concludes our proof of (II).

- In Step 5, we sketch the proof of (I), which is almost identical to Steps 2-4.

***1. Reduction to layer-wise condition.*** To prove (37), it suffices to show the following simpler condition: Let $j \in \{0, \ldots, k\}$. Let $R_j, S_j \in \mathbb{R}^{d \times d}$ be arbitrary matrices. For $C \in \mathbb{R}^{d \times d}$, let $A(tC, j)$ denote the collection of matrices, where $A(tC, j)_j = A_j + tC$, and for $i \neq j$, $A(tC, j)_i = A_i$. Define $B(tC, j)$ analogously. We show that for all $j \in \{0, \ldots, k\}$ and all $R_j, S_j \in \mathbb{R}^{d \times d}$, there exists $\tilde{R}_j = r_j \Sigma^{-1}$ and $\tilde{S}_j = s_j \Sigma^{-1}$, such that, at $t = 0$,

$$\frac{d}{dt} f(A(t\tilde{R}_j, j), B) \leq \frac{d}{dt} f(A(tR_j, j), B) \tag{39}$$

$$\text{and} \quad \frac{d}{dt} f(A, B(t\tilde{S}_j, j)) \leq \frac{d}{dt} f(A, B(tS_j, j)). \tag{40}$$

We can verify that (37) is equivalent to (39)+(40) by noticing that for any $(R, S) \in \mathbb{R}^{2 \times (k+1) \times d \times d}$, at $t = 0$, $\frac{d}{dt} f(A + tR, B + tS) = \sum_{j=0}^{k} \left( \frac{d}{dt} f(A(tR_j, j), B) + \frac{d}{dt} f(A, B(tS_j, j)) \right)$.

We will first focus on proving (40) (the proof of (39) is similar, and we present it in Step 5 at the end), for some index $j$ that is arbitrarily chosen but fixed throughout. Notice that $X_i$ and $Y_i$ in (34) are in fact functions of $A, B$ and $X_0$. For most of our subsequent discussion, $A_i$ (for all $i$) and $B_i$ (for all $i \neq j$) can be treated as constant matrices. We will however make the dependence on $X_0$ and $B_j$ explicit (as we consider the curve $B_j + tS$), i.e. we use $X_i(X, C)$ (resp $Y_i(X, C)$) to denote the value of $X_i$ (resp $Y_i$) from (34), with $X_0 = X$, and $B_j = C$.

By (35) and (34),

$$
\begin{aligned}
&f(A, B(tS_j, j)) \\
&= \mathbb{E} \left[ \text{Tr} \left( (I - M) Y_{k+1}(X_0, B_j + tS)^\top Y_{k+1}(X_0, B_j + tS_j) (I - M) \right) \right] \\
&= \mathbb{E} \left[ \text{Tr} \left( (I - M) G(X_0, B_j + tS_j)^\top w_\star^\top w_\star G(X_0, B_j + tS_j) (I - M) \right) \right] \\
&= \mathbb{E} \left[ \text{Tr} \left( (I - M) G(X_0, B_j + tS_j)^\top \Sigma^{-1} G(X_0, B_j + tS_j) (I - M) \right) \right]
\end{aligned}
$$

where $G(X, C) := X \prod_{i=0}^{k} \left( I - M X_i(X, C)^T A_i X_i(X, C) \right)$. The second equality follows from plugging in (34).

For the rest of this proof, let $U$ denote a uniformly randomly sampled orthogonal matrix. Let $U_\Sigma := \Sigma^{1/2} U \Sigma^{-1/2}$. Using the fact that $X_0 \stackrel{d}{=} U_\Sigma X_0$, we can verify

$$
\begin{aligned}
&\left. \frac{d}{dt} f(A, B(tS_j, j)) \right|_{t=0} \\
&= \left. \frac{d}{dt} \mathbb{E}_{X_0} \left[ \text{Tr} \left( (I - M) G(X_0, B_j + tS_j)^\top \Sigma^{-1} G(X_0, B_j + tS_j) (I - M) \right) \right] \right|_{t=0} \\
&= \left. \frac{d}{dt} \mathbb{E}_{X_0, U} \left[ \text{Tr} \left( (I - M) G(U_\Sigma X_0, B_j + tS_j)^\top \Sigma^{-1} G(U_\Sigma X_0, B_j + tS_j) (I - M) \right) \right] \right|_{t=0} \\
&= 2 \mathbb{E}_{X_0, U} \left[ \text{Tr} \left( (I - M) G(U_\Sigma X_0, B_j)^\top \Sigma^{-1} \left. \frac{d}{dt} G(U_\Sigma X_0, B_j + tS_j) \right|_{t=0} (I - M) \right) \right]. \quad (41)
\end{aligned}
$$

**2. $X_i$ and $\frac{d}{dt} X_i$ _under random transformation of_ $X_0$.** In this step, we prove that when $X_0$ is transformed by $U_\Sigma$, $X_i$ for $i \geq 1$ are likewise transformed in a simple manner. The first goal of this step is to show

$$
X_i(U_\Sigma X_0, B_j) = U_\Sigma X_i(X_0, B_j). \quad (42)
$$

We will prove this by induction. When $i = 0$, this clearly holds by definition. Suppose that (42) holds for some $i$. Then

$$
\begin{aligned}
&X_{i+1}(U_\Sigma X_0, B_j) \\
&= X_i(U_\Sigma X_0, B_j) + B_i X_i(U_\Sigma X_0, B_j) M X_i(U_\Sigma X_0, B_j)^T A_i X_i(U_\Sigma X_0, B_j) \\
&= U_\Sigma X_i(X_0, B_j) + U_\Sigma B_i X_i(X_0, B_j) M X_i(X_0, B_j)^T A_i X_i(X_0, B_j) \\
&= U_\Sigma X_{i+1}(X_0, B_j)
\end{aligned}
$$

where the second equality uses the inductive hypothesis, and the fact that $A_i = a_i \Sigma^{-1}$, so that $U_\Sigma^T A_i U_\Sigma = A_i$, and the fact that $B_i = b_i I$, from the definition of $\mathcal{S}$ and our assumption that $(A, B) \in \mathcal{S}$. This concludes the proof of (42).

We now present the second main result of this step. Let $U_\Sigma^{-1} := \Sigma^{1/2} U^T \Sigma^{-1/2}$, so that it satisfies $U_\Sigma U_\Sigma^{-1} = U_\Sigma^{-1} U_\Sigma = I$. For all $i$,

$$
\left. U_\Sigma^{-1} \frac{d}{dt} X_i(U_\Sigma X_0, B_j + tS_j) \right|_{t=0} = \left. \frac{d}{dt} X_i(X_0, B_j + tU_\Sigma^{-1} S_j U_\Sigma) \right|_{t=0}. \quad (43)
$$

To reduce notation, we will not write $\cdot|_{t=0}$ explicitly in the subsequent proof. We first write down the dynamics for the right-hand-side term of (43): From (34), for any $\ell \leq j$, and for any $i \geq j + 1$, and

for any $C \in \mathbb{R}^{d \times d}$,

$$\frac{d}{dt} X_\ell (X_0, B_j + tC) = 0$$

$$\frac{d}{dt} X_{j+1} (X_0, B_j + tC) = C X_j (X_0, B_j) M X_j (X_0, B_j)^\top A_j X_j (X_0, B_j)$$

$$\frac{d}{dt} X_{i+1} (X_0, B_j + tC) = \frac{d}{dt} X_i (X_0, B_j + tC)$$

$$+ B_i \left( \frac{d}{dt} X_i (X_0, B_j + tC) \right) M X_i (X_0, B_j)^\top A_i X_i (X_0, B_j)$$

$$+ B_i X_i (X_0, B_j) M \left( \frac{d}{dt} X_i (X_0, B_j + tC) \right)^\top A_i X_i (X_0, B_j)$$

$$+ B_i X_i (X_0, B_j) M X_i (X_0, B_j)^\top A_i \left( \frac{d}{dt} X_i (X_0, B_j + tC) \right) \qquad (44)$$

We are now ready to prove (43) using induction. For the base case, we verify that for $\ell \leq j$, $U_\Sigma^{-1} \frac{d}{dt} X_\ell (U_\Sigma X_0, B_k + tS_j) = 0 = \frac{d}{dt} X_\ell \left( X_0, B_j + tU_\Sigma^{-1} S_j U_\Sigma \right)$ (see first equation in (44)). For index $j + 1$, we verify that

$$U_\Sigma^{-1} \frac{d}{dt} X_{j+1} (U_\Sigma X_0, B_j + tS_j)$$

$$= U_\Sigma^{-1} S_j U_\Sigma X_j (X_0, B_j) M X_j (U_\Sigma X_0, B_j)^\top A_j X_j (U_\Sigma X_0, B_j)$$

$$= \frac{d}{dt} X_{j+1} \left( X_0, B_j + tU_\Sigma^{-1} S_j U_\Sigma \right) \qquad (45)$$

where we use two facts: 1. $X_i(U_\Sigma X_0, B_j) = U_\Sigma X_i(X_0, B_j)$ from (42), 2. $A_i = a_i \Sigma^{-1}$, so that $U_\Sigma^\top A_i U_\Sigma = A_i$. We verify by comparison to the second equation in (44) that $U_\Sigma^{-1} \frac{d}{dt} X_j (U_\Sigma X_0, B_j + tS_j) = 0 = \frac{d}{dt} X_j \left( X_0, B_j + tU_\Sigma^{-1} S_j U_\Sigma \right)$. These conclude the proof of the base case.

Now suppose that (43) holds for some $i$. We will now prove (43) holds for $i + 1$. From (34),

$$U_\Sigma^{-1} \frac{d}{dt} X_{i+1} (U_\Sigma X_0, B_j + tS_j)$$

$$= U_\Sigma^{-1} \frac{d}{dt} \left( X_i (U_\Sigma X_0, B_j + tS_j) \right)$$

$$+ U_\Sigma^{-1} \frac{d}{dt} \left( B_i X_i (U_\Sigma X_0, B_j + tS_j) M X_i (U_\Sigma X_0, B_j + tS_j)^\top A_i X_i (U_\Sigma X_0, B_j + tS_j) \right)$$

$$= U_\Sigma^{-1} \frac{d}{dt} \left( X_i (U_\Sigma X_0, B_j + tS_j) \right)$$

$$+ U_\Sigma^{-1} B_i \left( \frac{d}{dt} X_i (U_\Sigma X_0, B_j + tS_j) \right) M X_i (U_\Sigma X_0, B_j)^\top A_i X_i (U_\Sigma X_0, B_j)$$

$$+ U_\Sigma^{-1} B_i X_i (U_\Sigma X_0, B_j) M \left( \frac{d}{dt} X_i (U_\Sigma X_0, B_j + tS_j) \right)^\top A_i X_i (U_\Sigma X_0, B_j)$$

$$+ U_\Sigma^{-1} B_i X_i (U_\Sigma X_0, B_j) M X_i (U_\Sigma X_0, B_j)^\top A_i \left( \frac{d}{dt} X_i (U_\Sigma X_0, B_j + tS_j) \right)$$

$$\overset{(i)}{=} U_\Sigma^{-1} \frac{d}{dt} X_i (U_\Sigma X_0, B_j + tS_j)$$

$$+ B_i \left( U_\Sigma^{-1} \frac{d}{dt} X_i (U_\Sigma X_0, B_j + tS_j) \right) M X_i (X_0, B_j)^\top A_i X_i (X_0, B_j)$$

$$+ B_i X_i (X_0, B_j) M \left( U_\Sigma^{-1} \frac{d}{dt} X_i (U_\Sigma X_0, B_j + tS_j) \right)^\top A_i X_i (X_0, B_j)$$

$$- B_i X_i (X_0, B_j) M X_i (X_0, B_j)^\top A_i \left( U_\Sigma^{-1} \frac{d}{dt} X_i (U_\Sigma X_0, B_j + tS_j) \right)$$

$$\overset{(ii)}{=} \frac{d}{dt} X_i \left( X_0, B_j + tU_\Sigma^{-1} S_j U_\Sigma \right)$$

$$+ B_i \left( \frac{d}{dt} X_i \left( X_0, B_j + tU_\Sigma^{-1} S_j U_\Sigma \right) \right) M X_i \left( X_0, B_j \right)^\top A_i X_i \left( X_0, B_j \right)$$

$$+ B_i X_i \left( X_0, B_j \right) M \left( \frac{d}{dt} X_i \left( X_0, B_j + tU_\Sigma^{-1} S_j U_\Sigma \right) \right)^\top A_i X_i \left( X_0, B_j \right)$$

$$+ B_i X_i \left( X_0, B_j \right) M X_i \left( X_0, B_j \right)^\top A_i \left( \frac{d}{dt} X_i \left( X_0, B_j + tU_\Sigma^{-1} S_j U_\Sigma \right) \right) \tag{46}$$

In $(i)$ above, we crucially use the following facts: 1. $B_i = b_i I$ so that $U_\Sigma^{-1} B_i = B_i U_\Sigma^{-1}$, 2. $X_i(U_\Sigma X_0, B_j) = U_\Sigma X_i(X_0, B_j)$ from (42), 3. $A_i = a_i \Sigma^{-1}$, so that $U_\Sigma^\top A_i U_\Sigma = A_i$, 4. $U_\Sigma U_\Sigma^{-1} = U_\Sigma^{-1} U_\Sigma = I$. $(ii)$ follows from our inductive hypothesis. The inductive proof is complete by verifying that (46) exactly matches the third equation of (44) when $C = U_\Sigma^{-1} S U_\Sigma$.

**3. $G$ and $\frac{d}{dt} G$ under random transformation of $X_0$.** We now verify that $G(U_\Sigma X_0, B_j) = U_\Sigma G(X_0, B_j)$. This is a straightforward consequence of (42) as

$$G(U_\Sigma X_0, B_j)$$

$$= U_\Sigma X_0 \prod_{i=0}^{k} \left( I + M X_i(U_\Sigma X_0, B_j)^T A_i X_i(U_\Sigma X_0, B_j) \right)$$

$$= U_\Sigma X_0 \prod_{i=0}^{k} \left( I + M X_i(X_0, B_j)^T A_i X_i(X_0, B_j) \right)$$

$$= U_\Sigma G(X_0, B_j), \tag{47}$$

where the second equality uses (42), as well as the fact that $U_\Sigma^\top A_i U_\Sigma = A_i$. Next, we will show that

$$U_\Sigma^{-1} \left. \frac{d}{dt} G(U_\Sigma X_0, B_j + tS_j) \right|_{t=0} = \left. \frac{d}{dt} G(X_0, B_j + tU_\Sigma^{-1} S_j U_\Sigma) \right|_{t=0}. \tag{48}$$

To see this, we can expand

$$U_\Sigma^{-1} \frac{d}{dt} G(U_\Sigma X_0, B_j + tS_j)$$

$$= U_\Sigma^{-1} \frac{d}{dt} \left( U_\Sigma X_0 \prod_{i=0}^{k} \left( I + M X_i(U_\Sigma X_0, B_j + tS_j)^T A_i X_i(U_\Sigma X_0, B_j + tS_j) \right) \right)$$

$$= X_0 \sum_{i=0}^{k} \left( \prod_{\ell=0}^{i-1} \left( I + M X_\ell(U_\Sigma X_0, B_j)^T A_\ell X_i(U_\Sigma X_0, B_\ell) \right) \right)$$

$$\cdot M \frac{d}{dt} \left( X_i(U_\Sigma X_0, B_j + tS_j)^T A_i X_i(U_\Sigma X_0, B_j) \right)$$

$$\cdot \left( \prod_{\ell=i+1}^{k} \left( I + M X_\ell(U_\Sigma X_0, B_j)^T A_\ell X_i(U_\Sigma X_0, B_\ell) \right) \right)$$

$$\overset{(i)}{=} X_0 \sum_{i=0}^{k} \left( \prod_{\ell=0}^{i-1} \left( I + M X_\ell(X_0, B_j)^T A_\ell X_\ell(X_0, B_\ell) \right) \right)$$

$$\cdot M \left( \left( U_\Sigma^{-1} \frac{d}{dt} X_i(U_\Sigma X_0, B_j + tS_j) \right)^T A_i X_i(X_0, B_j) + M X_i(X_0, B_j)^T A_i \left( U_\Sigma^{-1} \frac{d}{dt} X_i(U_\Sigma X_0, B_j + tS_j) \right) \right)$$

$$\cdot \left( \prod_{\ell=i+1}^{k} \left( I + M X_\ell(X_0, B_j)^T A_\ell X_\ell(X_0, B_\ell) \right) \right)$$

$$\overset{(ii)}{=} X_0 \sum_{i=0}^{k} \left( \prod_{\ell=0}^{i-1} \left( I + M X_\ell(X_0, B_j)^T A_\ell X_\ell(X_0, B_\ell) \right) \right)$$

$$\cdot M \left( \left( \left( \frac{d}{dt} X_i(X_0, B_j + tU_\Sigma^{-1} S_j U_\Sigma) \right)^T A_i X_i(X_0, B_j) + M X_i(X_0, B_j)^T A_i \left( \frac{d}{dt} X_i(X_0, B_j + tU_\Sigma^{-1} S_j U_\Sigma) \right) \right) \right.$$

$$\left. \cdot \left( \prod_{\ell=i+1}^k \left( I + M X_\ell(X_0, B_j)^T A_\ell X_\ell(X_0, B_\ell) \right) \right) \right)$$

$$\overset{(iii)}{=} \frac{d}{dt} G(X_0, B_j + tU_\Sigma^{-1} S_j U_\Sigma)$$

In $(i)$ above, we the following facts: 1. $X_i(U_\Sigma X_0, B_j) = U_\Sigma X_i(X_0, B_j)$ from (42), 2. $A_i = a_i \Sigma^{-1}$, so that $U_\Sigma^\top A_i U_\Sigma = A_i$, 3. $U_\Sigma U_\Sigma^{-1} = U_\Sigma^{-1} U_\Sigma = I$. $(ii)$ follows from (43). $(iii)$ is by definition of $G$.

**4. Putting everything together.** Let us now continue from (41). We can now plug (47) and (48) into (41):

$$\frac{d}{dt} f(A, B(tS_j, j)) \Big|_{t=0}$$

$$= 2 \, \mathbb{E}_{X_0, U} \left[ \mathrm{Tr} \left( (I - M) \, G(U_\Sigma X_0, B_j)^\top \Sigma^{-1} \frac{d}{dt} G(U_\Sigma X_0, B_j + tS_j) \Big|_{t=0} (I - M) \right) \right]$$

$$\overset{(i)}{=} 2 \, \mathbb{E}_{X_0, U} \left[ \mathrm{Tr} \left( (I - M) \, G(X_0, B_j)^\top \Sigma^{-1} \frac{d}{dt} G(X_0, B_j + tU_\Sigma^{-1} S_j U_\Sigma) \Big|_{t=0} (I - M) \right) \right]$$

$$= 2 \, \mathbb{E}_{X_0} \left[ \mathrm{Tr} \left( (I - M) \, G(X_0, B_j)^\top \Sigma^{-1} \mathbb{E}_U \left[ \frac{d}{dt} G(X_0, B_j + tU_\Sigma^{-1} S_j U_\Sigma) \Big|_{t=0} \right] (I - M) \right) \right]$$

$$\overset{(ii)}{=} 2 \, \mathbb{E}_{X_0} \left[ \mathrm{Tr} \left( (I - M) \, G(X_0, B_j)^\top \Sigma^{-1} \frac{d}{dt} G(X_0, B_j + t \, \mathbb{E}_U \left[ U_\Sigma^{-1} S_j U_\Sigma \right]) \Big|_{t=0} (I - M) \right) \right]$$

$$= 2 \, \mathbb{E}_{X_0} \left[ \mathrm{Tr} \left( (I - M) \, G(X_0, B_j)^\top \Sigma^{-1} \frac{d}{dt} G(X_0, B_j + ts_j I) \Big|_{t=0} (I - M) \right) \right]$$

$$= \frac{d}{dt} f(A, B(ts_j I, j)) \Big|_{t=0}$$

where $s_j := \frac{1}{d} \mathrm{Tr} \left( \Sigma^{-1/2} S_j \Sigma^{1/2} \right)$. In the above, $(i)$ uses 1. (47) and (48), as well as the fact that $U_\Sigma^\top \Sigma^{-1} U_\Sigma = \Sigma^{-1}$. $(ii)$ uses the fact that $\frac{d}{dt} G(X_0, B_j + tC) \big|_{t=0}$ is affine in $C$. To see this, one can verify from (44), using a simple induction argument, that $\frac{d}{dt} X_i(X_0, B_j + tC)$ is affine in $C$ for all $i$. We can then verify from the definition of $G$, e.g. using similar algebra as the proof of (48), that $\frac{d}{dt} G(X_0, B_j + C)$ is affine in $\frac{d}{dt} X_i(X_0, B_j + tC)$. Thus $\mathbb{E}_U \left[ G(X_0, B_j + tU_\Sigma^{-1} S_j U_\Sigma) \right] = G(X_0, B_j + t \, \mathbb{E}_U \left[ U_\Sigma^{-1} S_j U_\Sigma \right])$.

With this, we conclude our proof of (40).

**5. Proof of (39).** We will now prove (39) for fixed but arbitrary $j$, i.e. there is some $r_j$ such that

$$\frac{d}{dt} f(A(t \cdot r_j \Sigma^{-1}, j), B) \leq \frac{d}{dt} f(A(tR_j, j), B).$$

The proof is very similar to the proof of (40) that we just saw, and we will essentially repeat the same steps from Step 2-4 above.

Since we now consider perturbations to $A$ instead of to $B$, we will need to redefine some notation: let $X_i(X, C)$ (resp $Y_i(X, C)$) to denote the value of $X_i$ (resp $Y_i$) from (34), with $X_0 = X$, and $A_j = C$ (previously it was with $B_j = C$). Let $G(X, A_j + C) := X \prod_{i=0}^i \left( I + M \left( X_i(X, A_j + C)^T A(C, j)_i X_i(X, A_j + C) \right) \right)$, where recall that $A(C, j) := A_j + C$, and $A(C, j)_\ell := A_\ell$ for all $\ell \in \{0...k\} \setminus \{j\}$.

We first verify that

$$X_i(U_\Sigma X_0, A_j) = U_\Sigma X_i(X_0, A_j)$$
$$G(U_\Sigma X_0, A_j) = U_\Sigma G(X_0, A_j). \tag{49}$$

The proofs are identical to the proofs of (42) and (47) so we omit them. Next, we show that for all $i$,

$$U_\Sigma^{-1} \frac{d}{dt} X_i(U_\Sigma X_0, A_j + tR_j)\Big|_{t=0} = \frac{d}{dt} X_i(X_0, A_j + tU_\Sigma^\top R_j U_\Sigma)\Big|_{t=0}. \tag{50}$$

We establish the dynamics for the right-hand-side of (50):

$$\frac{d}{dt} X_\ell(X_0, A_j + tC) = 0$$

$$\frac{d}{dt} X_{j+1}(X_0, A_j + tC) = B_j X_j(X_0, A_j) M X_j(X_0, A_j)^\top C X_j(X_0, A_j)$$

$$\frac{d}{dt} X_{i+1}(X_0, A_j + tC) = \frac{d}{dt} X_i(X_0, A_j + tC)$$

$$+ B_i \left( \frac{d}{dt} X_i(X_0, A_j + tC) \right) M X_i(X_0, A_j)^\top A_i X_i(X_0, A_j)$$

$$+ B_i X_i(X_0, A_j) M \left( \frac{d}{dt} X_i(X_0, A_j + tC) \right)^\top A_i X_i(X_0, A_j)$$

$$+ B_i X_i(X_0, A_j) M X_i(X_0, A_j)^\top A_i \left( \frac{d}{dt} X_i(X_0, A_j + tC) \right) \tag{51}$$

Similar to (45), we show that for $i \le j$,

$$U_\Sigma^{-1} \frac{d}{dt} X_i(U_\Sigma X_0, A_j + tR_j) = 0 = U_\Sigma^{-1} \frac{d}{dt} X_i(U_\Sigma X_0, A_j + tU_\Sigma R_j U_\Sigma)$$

and

$$U_\Sigma^{-1} \frac{d}{dt} X_{j+1}(U_\Sigma X_0, A_j + tR_j)$$

$$= U_\Sigma^{-1} B_j U_\Sigma X_j(X_0, A_j) M X_j(U_\Sigma X_0, A_j)^\top A_j X_j(U_\Sigma X_0, A_j)$$

$$= \frac{d}{dt} X_{j+1}(X_0, A_j + tU_\Sigma^\top R_j U_\Sigma).$$

Finally, for the inductive step, we follow identical steps leading up to (46) to show that

$$U_\Sigma^{-1} \frac{d}{dt} X_{i+1}(U_\Sigma X_0, A_j + tR_j)$$

$$= \frac{d}{dt} X_i(X_0, A_j + tU_\Sigma^\top R_j U_\Sigma)$$

$$+ B_i \left( \frac{d}{dt} X_i(X_0, A_j + tU_\Sigma^\top R_j U_\Sigma) \right) M X_i(X_0, A_j)^\top A_i X_i(X_0, A_j)$$

$$+ B_i X_i(X_0, A_j) M \left( \frac{d}{dt} X_i(X_0, A_j + tU_\Sigma^\top R_j U_\Sigma) \right)^\top A_i X_i(X_0, A_j)$$

$$+ B_i X_i(X_0, A_j) M X_i(X_0, A_j)^\top A_i \left( \frac{d}{dt} X_i(X_0, A_j + tU_\Sigma^\top R_j U_\Sigma) \right) \tag{52}$$

The inductive proof is complete by verifying that (52) exactly matches the third equation of (51) when $C = U_\Sigma^{-1} S U_\Sigma$. This concludes the proof of (50).

Next, we study the time derivative of $G(U_\Sigma X_0, A_j + tR_j)$ and show that

$$U_\Sigma^{-1} \frac{d}{dt} G(U_\Sigma X_0, A_j + tR_j) = \frac{d}{dt} G(X_0, A_j + tU_\Sigma^\top R_j U_\Sigma). \tag{53}$$

This proof differs significantly from that of (48) in a few places, so we provide the whole derivation below. By chain-rule, we can write

$$U_\Sigma^{-1} \frac{d}{dt} G(U_\Sigma X_0, A_j + tR_j) = \spadesuit + \heartsuit$$

where

$$\spadesuit := U_\Sigma^{-1} \frac{d}{dt} \left( U_\Sigma X_0 \prod_{i=0}^{k} \left( I + M X_i(U_\Sigma X_0, A_j + tR_j)^T A_i X_i(U_\Sigma X_0, A_j + tR_j) \right) \right)$$

and

$$\heartsuit := U_\Sigma^{-1} U_\Sigma X_0 \left( \prod_{i=0}^{j-1} \left( I + M X_i(U_\Sigma X_0, A_j)^T A_i X_i(U_\Sigma X_0, A_j) \right) \right)$$
$$\cdot M X_j(U_\Sigma X_0, A_j)^T R_j X_j(U_\Sigma X_0, A_j)$$
$$\cdot \left( \prod_{i=j+1}^{k} \left( I + M X_i(U_\Sigma X_0, A_j)^T A_i X_i(U_\Sigma X_0, A_j) \right) \right).$$

We will separately simplify $\spadesuit$ and $\heartsuit$, and verify at the end that summing them recovers the right-hand-side of (53). We begin with $\spadesuit$, and the steps are almost identical to the proof of (48).

$\spadesuit$

$$=U_\Sigma^{-1} \frac{d}{dt} \left( U_\Sigma X_0 \prod_{i=0}^{k} \left( I + M X_i(U_\Sigma X_0, A_j + tR_j)^T A_i X_i(U_\Sigma X_0, A_j + tR_j) \right) \right)$$

$$=X_0 \sum_{i=0}^{k} \left( \prod_{\ell=0}^{i-1} \left( I + M X_\ell(U_\Sigma X_0, A_j)^T A_\ell X_i(U_\Sigma X_0, A_\ell) \right) \right)$$
$$\cdot M \frac{d}{dt} \left( X_i(U_\Sigma X_0, A_j + tR_j)^T A_i X_i(U_\Sigma X_0, A_j + tR_j) \right)$$
$$\cdot \left( \prod_{\ell=i+1}^{k} \left( I + M X_\ell(U_\Sigma X_0, A_j)^T A_\ell X_i(U_\Sigma X_0, A_\ell) \right) \right)$$

$$\overset{(i)}{=} X_0 \sum_{i=0}^{k} \left( \prod_{\ell=0}^{i-1} \left( I + M X_\ell(X_0, A_j)^T A_\ell X_\ell(X_0, A_\ell) \right) \right)$$
$$\cdot M \left( \left( U_\Sigma^{-1} \frac{d}{dt} X_i(U_\Sigma X_0, A_j + tR_j) \right)^T A_i X_i(X_0, A_j) + M X_i(X_0, A_j)^T A_i \left( U_\Sigma^{-1} \frac{d}{dt} X_i(U_\Sigma X_0, A_j + tR_j) \right) \right)$$
$$\cdot \left( \prod_{\ell=i+1}^{k} \left( I + M X_\ell(X_0, A_j)^T A_\ell X_\ell(X_0, A_\ell) \right) \right)$$

$$\overset{(ii)}{=} X_0 \sum_{i=0}^{k} \left( \prod_{\ell=0}^{i-1} \left( I + M X_\ell(X_0, A_j)^T A_\ell X_\ell(X_0, A_\ell) \right) \right)$$
$$\cdot M \left( \left( \frac{d}{dt} X_i(X_0, A_j + tU_\Sigma^\top R_j U_\Sigma) \right)^T A_i X_i(X_0, A_j) + M X_i(X_0, A_j)^T A_i \left( \frac{d}{dt} X_i(X_0, A_j + tU_\Sigma^\top R_j U_\Sigma) \right) \right)$$
$$\cdot \left( \prod_{\ell=i+1}^{k} \left( I + M X_\ell(X_0, A_j)^T A_\ell X_\ell(X_0, A_\ell) \right) \right)$$

$$= X_0 \sum_{i=0}^{k} \left( \prod_{\ell=0}^{i-1} \left( I + M X_\ell(X_0, A_j)^T A_\ell X_\ell(X_0, A_\ell) \right) \right)$$
$$\cdot M \frac{d}{dt} \left( X_i(X_0, A_j + tU_\Sigma^\top R_j U_\Sigma)^T A_i X_i(X_0, A_j + tU_\Sigma^\top R_j U_\Sigma) \right)$$
$$\cdot \left( \prod_{\ell=i+1}^{k} \left( I + M X_\ell(X_0, A_j)^T A_\ell X_\ell(X_0, A_\ell) \right) \right) \tag{54}$$

In $(i)$ above, we the following facts: 1. $X_i(U_\Sigma X_0, B_j) = U_\Sigma X_i(X_0, B_j)$ from (49), 2. $A_i = a_i \Sigma^{-1}$, so that $U_\Sigma^\top A_i U_\Sigma = A_i$, 3. $U_\Sigma U_\Sigma^{-1} = U_\Sigma^{-1} U_\Sigma = I$. $(ii)$ follows from (50).

We will now simplify $\heartsuit$.

$$\heartsuit$$

$$=U_\Sigma^{-1} U_\Sigma X_0 \left( \prod_{i=0}^{j-1} \left( I + M X_i(U_\Sigma X_0, A_j)^T A_i X_i(U_\Sigma X_0, A_j) \right) \right)$$
$$\cdot M X_j(U_\Sigma X_0, A_j)^T R_j X_j(U_\Sigma X_0, A_j)$$
$$\cdot \left( \prod_{i=j+1}^{k} \left( I + M X_i(U_\Sigma X_0, A_j)^T A_i X_i(U_\Sigma X_0, A_j) \right) \right)$$
$$\overset{(i)}{=} X_0 \left( \prod_{i=0}^{j-1} \left( I + M X_i(X_0, A_j)^T A_i X_i(X_0, A_j) \right) \right) M X_j(X_0, A_j)^\top U_\Sigma^\top R_j U_\Sigma X_j(X_0, A_j)$$
$$\cdot \left( \prod_{i=j+1}^{k} \left( I + M X_i(X_0, A_j)^T A_i X_i(X_0, A_j) \right) \right), \tag{55}$$

where $(i)$ uses the fact that $X_i(U_\Sigma X_0, B_j) = U_\Sigma X_i(X_0, B_j)$ from (49) and the fact that $A_i = a_i \Sigma^{-1}$.

By expanding $\frac{d}{dt} G(X_0, A_j + t U_\Sigma^\top R_j U_\Sigma)$, we verify that

$$\frac{d}{dt} G(X_0, A_j + t U_\Sigma^\top R_j U_\Sigma) = (54) + (55) = \spadesuit + \heartsuit = U_\Sigma^{-1} \frac{d}{dt} G(U_\Sigma X_0, A_j + t R_j),$$

this concludes the proof of (53).

The remainder of the proof is similar to what was done in (41) in Step 4:

$$\frac{d}{dt} f(A(tR_j, j), B) \Big|_{t=0}$$
$$= 2 \, \mathbb{E}_{X_0, U} \left[ \mathrm{Tr} \left( (I - M) \, G(U_\Sigma X_0, A_j)^\top \Sigma^{-1} \frac{d}{dt} G(U_\Sigma X_0, A_j + t R_j) \Big|_{t=0} (I - M) \right) \right]$$
$$\overset{(i)}{=} 2 \, \mathbb{E}_{X_0, U} \left[ \mathrm{Tr} \left( (I - M) \, G(X_0, A_j)^\top \Sigma^{-1} \frac{d}{dt} G(X_0, A_j + t U_\Sigma^\top R_j U_\Sigma) \Big|_{t=0} (I - M) \right) \right]$$
$$\overset{(ii)}{=} 2 \, \mathbb{E}_{X_0} \left[ \mathrm{Tr} \left( (I - M) \, G(X_0, A_j)^\top \Sigma^{-1} \frac{d}{dt} G(X_0, A_j + t \, \mathbb{E}_U \left[ U_\Sigma^\top R_j U_\Sigma \right]) \Big|_{t=0} (I - M) \right) \right]$$
$$= 2 \, \mathbb{E}_{X_0} \left[ \mathrm{Tr} \left( (I - M) \, G(X_0, A_j)^\top \Sigma^{-1} \frac{d}{dt} G(X_0, A_j + t \cdot r_j \Sigma^{-1}) \Big|_{t=0} (I - M) \right) \right]$$
$$= \frac{d}{dt} f(A(t \cdot r_j \Sigma^{-1}, j), B) \Big|_{t=0},$$

where $r_j := \frac{1}{d} \mathrm{Tr} \left( \Sigma^{1/2} R_j \Sigma^{1/2} \right)$. In the above, $(i)$ uses 1. (49) and (53), as well as the fact that $U_\Sigma^\top \Sigma^{-1} U_\Sigma = \Sigma^{-1}$. $(ii)$ uses the fact that $\frac{d}{dt} G(X_0, A_j + tC) \big|_{t=0}$ is affine in $C$. To see this, one can verify using a simple induction argument, that $\frac{d}{dt} X_i(X_0, A_j + tC)$ is affine in $C$ for all $i$. We can then verify from the definition of $G$, e.g. using similar algebra as the proof of (53), that $\frac{d}{dt} G(X_0, A_j + C)$ is affine in $\frac{d}{dt} X_i(X_0, A_j + tC)$ and $C$. Thus $\mathbb{E}_U \left[ G(X_0, A_j + t U_\Sigma^\top R_j U_\Sigma) \right] = G(X_0, A_j + t \, \mathbb{E}_U \left[ U_\Sigma^\top R_j U_\Sigma \right])$.

This concludes the proof of (39), and hence of the whole theorem.

## B.4 Equivalence under permutation

**Lemma 4.** *Consider the same setup as [Theorem 3](#). Let $A = \{A_i\}_{i=0}^k$, with $A_i = a_i \Sigma^{-1}$. Let*

$$f(A) := f\left(\left\{Q_i = \begin{bmatrix} A_i & 0 \\ 0 & 0 \end{bmatrix}, P_i = \begin{bmatrix} 0_{d\times d} & 0 \\ 0 & 1 \end{bmatrix}\right\}_{i=0}^k\right).$$

*Let $i, j \in \{0, \ldots, k\}$ be any two arbitrary indices, and let $\tilde{A}_i = A_j$, $\tilde{A}_j = A_i$, and let $\tilde{A}_\ell = A_\ell$ for all $\ell \in \{0, \ldots, k\} \setminus \{i, j\}$. Then $f(A) = f(\tilde{A})$*

*Proof.* Following the same setup leading up to (26) in the proof of [Theorem 3](#), we verify that the in-context loss is

$$f(A) = \mathbb{E}\left[\text{Tr}\left((I - M)\, G(X_0, A)^\top \Sigma^{-1} G(X_0, A)\,(I - M)\right)\right]$$

where $G(X_0, A) := X_0 \prod_{\ell=0}^k \left(I + M X_0^T A_\ell X_0\right)$.

Consider any fixed index $\ell$. We will show that

$$\left(I + M X_0^T A_\ell X_0\right)\left(I + M X_0^T A_{\ell+1} X_0\right) = \left(I + M X_0^T A_{\ell+1} X_0\right)\left(I + M X_0^T A_\ell X_0\right).$$

The lemma can then be proven by repeatedly applying the above, so that indices of $A_i$ and $A_j$ are swapped.

To prove the above equality,

$$\begin{aligned}
&\left(I + M X_0^T A_\ell X_0\right)\left(I + M X_0^T A_{\ell+1} X_0\right) \\
=& I + M X_0^T A_\ell X_0 + M X_0^T A_{\ell+1} X_0 + M X_0^T A_\ell X_0 M X_0^T A_{\ell+1} X_0 \\
=& I + M X_0^T A_\ell X_0 + M X_0^T A_{\ell+1} X_0 + M X_0^T a_\ell \Sigma^{-1} X_0 M X_0^T a_{\ell+1} \Sigma^{-1} X_0 \\
=& I + M X_0^T A_\ell X_0 + M X_0^T A_{\ell+1} X_0 + M X_0^T a_{\ell+1} \Sigma^{-1} X_0 M X_0^T a_\ell \Sigma^{-1} X_0 \\
=& \left(I + M X_0^T A_{\ell+1} X_0\right)\left(I + M X_0^T A_\ell X_0\right).
\end{aligned}$$

This concludes the proof. Notice that we crucially used the fact that $A_\ell$ and $A_{\ell+1}$ are the same matrix up to scaling. $\qquad\square$

# C  Auxiliary Lemmas

## C.1  Proof of Lemma 1 (Equivalence to Preconditioned Gradient Descent)

Consider fixed samples $x^{(1)}, \ldots, x^{(n)}$, and fixed $w_\star$. Let $P = \{P_i\}_{i=0}^k$, $Q = \{Q_i\}_{i=0}^k$ denote fixed weights. Let $Z_i$ evolve as described in (4). Let $X_i$ denote the first $d$ rows of $Z_k$ (under (8), $X_i = X_0$ for all $I$) and let $Y_i$ denote the $(d+1)^{th}$ row of $Z_i$. Let $g(x, y, k) : \mathbb{R}^d \times \mathbb{R} \times \mathbb{Z} \to \mathbb{R}$ be a function defined as follows: let $x^{n+1} = x$ and let $y_0^{n+1} = y$, then $g(x, y, k) := y_k^{n+1}$. Note that $y_k^{n+1} = [Y_k]_{n+1}$.

We verify that, under (8), the formula for updating $y_k^{(n+1)}$ is given by

$$Y_{k+1} = Y_k - \frac{1}{n} Y_k M X_0^\top A_k X_0.$$

where $M$ is a mask given by $\begin{bmatrix} I & 0 \\ 0 & 0 \end{bmatrix}$. We can verify the following facts

1. $g(x, y, k) = g(x, 0, k) + y$. To see this, notice first that for all $i \in \{1, \ldots, n\}$,

$$y_{k+1}^{(i)} = y_k^{(i)} - \frac{1}{n} \sum_{j=1}^n x^{(i)^T} A_k x^{(j)} y_k^{(j)}.$$

In other words, $y_k^{(i)}$ does not depend on $y_t^{(n+1)}$ for any $t$. Next, for $y_k^{(n+1)}$ itself,

$$y_{k+1}^{(n+1)} = y_k^{(n+1)} - \frac{1}{n} \sum_{j=1}^{n} x^{(n+1)^T} A_k x^{(j)} y_k^{(j)},$$

which depends on $y_k^{n+1}$ only additively. We can verify under a simple induction that $g(x, y, k+1) - y = g(x, y, k) - y$.

2. $g(x, 0, k)$ is linear in $x$. To see this, notice first that for $j \neq n+1$, $y_k^{(j)}$ is does not depend on $x_t^{(n+1)}$ for all $t, j, k$. Consequently, the update formula for $y_{k+1}^{(n+1)}$ depends only linearly on $x^{(n+1)}$ and $y_k^{(n+1)}$. Finally, $y_0^{(n+1)} = 0$ is linear in $x$, so the conclusion follows by induction.

With these two facts in mind, we verify that for each $k$, there exists a $\theta_k \in \mathbb{R}^d$, such that
$$g(x, y, k) = g(x, 0, k) + y = \langle \theta_k, x \rangle + y$$
for all $x, y$. It follows from definition that $g(x, y, 0) = y$, so that $\langle \theta_0, x \rangle = g(x, y, 0) - y = 0$, so that $\theta_0 = 0$.

We now turn our attention to the third crucial fact: for all $i$,
$$g(x^{(i)}, y^{(i)}, k) = y_k^{(i)} = \left\langle \theta_k, x^{(i)} \right\rangle + y^{(i)}$$

To see this, suppose that we let $x^{(n+1)} := x^{(i)}$ for some $i \in 1, \ldots, n$. Then

$$y_{k+1}^{(i)} = y_k^{(i)} - \frac{1}{n} \sum_{j=1}^{n} x^{(i)^T} A_k x^{(j)} y_k^{(j)}$$

$$y_{k+1}^{(n+1)} = y_k^{(n+1)} - \frac{1}{n} \sum_{j=1}^{n} x^{(n+1)^T} A_k x^{(j)} y_k^{(j)},$$

thus $y_{k+1}^{(i)} = y_{k+1}^{(n+1)}$ if $y_k^{(i)} = y_k^{(n+1)}$, and the induction proof is completed by noting that $y_0^{(i)} = y_0^{(n+1)}$ by definition. Let $\bar{X} \in R^{d \times n}$ be the matrix whose columns are $x^{(1)}, \ldots, x^{(n)}$, leaving out $x^{(n+1)}$. Let $\bar{Y}_k \in \mathbb{R}^{1 \times n}$ denote the vector of $y_k^{(1)}, \ldots, y_k^{(n)}$. Then it follows that
$$\bar{Y}_k = \bar{Y}_0 + \theta_k^T \bar{X}.$$

Using the above fact, the update formula for $y_k^{(n+1)}$ can be written as

$$y_{k+1}^{(n+1)} = y_k^{(n+1)} - \frac{1}{n} \left\langle A_k X^\top Y_k, x^{(n+1)} \right\rangle$$

$$\Rightarrow \quad \left\langle \theta_{k+1}, x^{(n+1)} \right\rangle = \left\langle \theta_k, x^{(n+1)} \right\rangle - \frac{1}{n} \left\langle A_k \bar{X} \left( \bar{X}^T \theta_k + \bar{Y}_0 \right), x^{(n+1)} \right\rangle$$

$$= \left\langle \theta_k, x^{(n+1)} \right\rangle - \frac{1}{n} \left\langle A_k \bar{X} \left( \bar{X}^T \left( \theta_k + w_\star \right) \right), x^{(n+1)} \right\rangle$$

Since the choice of $x^{(n+1)}$ is arbitrary, we get the more general update formula

$$\theta_{k+1} = \theta_k - \frac{1}{n} A_k \bar{X} \bar{X}^T \left( \theta_k + w_\star \right).$$

We can treat $A_k$ as a preconditioner. Let $f(\theta) := \frac{1}{2n} (\theta + w_\star)^T \bar{X} \bar{X}^T (\theta + w_\star)$, then

$$\theta_{k+1} = \theta_k - \frac{1}{n} A_k \nabla f(\theta).$$

Finally, let $w_k^{\text{gd}} := -\theta_k$. We verify that $f(-w) = R_{w_\star}(w)$, so that

$$w_{k+1}^{\text{gd}} = w_k^{\text{gd}} - \frac{1}{n} A_k \nabla R_{w_\star}(w_k^{\text{gd}}).$$

We also verify that for any $x^{(n+1)}$, the prediction of $y_k^{(n+1)}$ is

$$g\left( x^{(n+1)}, y^{(n+1)}, k \right) = y^{(n+1)} - \left\langle \theta, x^{(n+1)} \right\rangle = y^{(n+1)} + \left\langle w_k^{\text{gd}}, x^{(n+1)} \right\rangle.$$

This concludes the proof.

## C.2 Reformulating the in-context loss

In this section, we will develop a re-formulation in-context loss, defined in (5), in a more convenient form (see Lemma 5).

For the entirety of this section, we assume that the transformer parameters $\{P_i, Q_i\}_{i=0}^k$ are of the form defined in (11), which we reproduce below for ease of reference:

$$P_i = \begin{bmatrix} B_i & 0 \\ 0 & 1 \end{bmatrix}, \quad Q_i = \begin{bmatrix} A_i & 0 \\ 0 & 0 \end{bmatrix}.$$

Recall the update dynamics in (4), which we reproduce below:

$$Z_{i+1} = Z_i + \frac{1}{n} P Z_i M Z_i^\top Q Z_i, \tag{56}$$

where $M$ is a mask matrix given by $M := \begin{bmatrix} I_{n \times n} & 0 \\ 0 & 0 \end{bmatrix}$. Let $X_k \in \mathbb{R}^{d \times n+1}$ denote the first $d$ rows of $Z_k$ and let $Y_k \in \mathbb{R}^{1 \times n+1}$ denote the $(d+1)^{th}$ (last) row of $Z_k$. Then the dynamics in (56) is equivalent to

$$X_{i+1} = X_i + \frac{1}{n} B_i X_i M X_i^T A_i X_i$$

$$Y_{i+1} = Y_i + \frac{1}{n} Y_i M X_i^T A_i X_i. \tag{57}$$

We present below an equivalent form for the in-context loss from (5):

**Lemma 5.** *Let $p_x$ and $p_w$ denote distributions over $\mathbb{R}^d$. Let $x^{(1)}, \ldots, x^{(n+1)} \overset{iid}{\sim} p_x$ and $w_\star \sim p_w$. Let $Z_0 \in \mathbb{R}^{d+1 \times n+1}$ be as defined in (1):*

$$Z_0 = \begin{bmatrix} x^{(1)} & x^{(2)} & \cdots & x^{(n)} & x^{(n+1)} \\ y^{(1)} & y^{(2)} & \cdots & y^{(n)} & 0 \end{bmatrix} \in \mathbb{R}^{(d+1) \times (n+1)}.$$

*Let $Z_k$ denote the output of the $(k-1)^{th}$ layer of the linear transformer (as defined in (56), initialized at $Z_0$). Let $f\left(\{P_i, Q_i\}_{i=0}^k\right)$ denote the in-context loss defined in (5), i.e.*

$$f\left(\{P_i, Q_i\}_{i=0}^k\right) = \mathbb{E}_{(Z_0, w_\star)}\left[\left([Z_k]_{(d+1),(n+1)} + w_\star^\top x^{(n+1)}\right)^2\right]. \tag{58}$$

*Let $\overline{Z}_0$ be defined as*

$$\overline{Z}_0 = \begin{bmatrix} x^{(1)} & x^{(2)} & \cdots & x^{(n)} & x^{(n+1)} \\ y^{(1)} & y^{(2)} & \cdots & y^{(n)} & y^{(n+1)} \end{bmatrix} \in \mathbb{R}^{(d+1) \times (n+1)},$$

*where $y^{(n+1)} = \langle w_\star, x^{(n+1)} \rangle$. Let $\overline{Z}_k$ denote the output of the $(k-1)^{th}$ layer of the linear transformer (as defined in (56), initialized at $\overline{Z}_0$). Assume $\{P_i, Q_i\}_{i=0}^k$ be of the form in (11). Then the loss in (5) has the equivalent form*

$$f\left(\{A_i, B_i\}_{i=0}^k\right) := f\left(\{P_i, Q_i\}_{i=0}^k\right) = \mathbb{E}_{(\overline{Z}_0, w_\star)}\left[\mathrm{Tr}\left((I-M)\overline{Y}_{k+1}^\top \overline{Y}_{k+1}(I-M)\right)\right],$$

*where $\overline{Y}_{k+1} \in \mathbb{R}^{1 \times n+1}$ is the $(d+1)^{th}$ row of $\overline{Z}_k$.*

Before proving Lemma 5, we first establish an intermediate result (Lemma 6 below). To facilitate discussion, let us define a function $F_X\left(\{A_i, B_i\}_{i=0}^k, X_0, Y_0\right)$ and $F_Y\left(\{A_i, B_i\}_{i=0}^k, X_0, Y_0\right)$ to be the outputs, after $k$ layers of linear transformers respectively. I.e.

$$F_X\left(\{A_i, B_i\}_{i=0}^k, X_0, Y_0\right) = X_{k+1}$$

$$F_Y\left(\{A_i, B_i\}_{i=0}^k, X_0, Y_0\right) = Y_{k+1},$$

as defined in (57), given initialization $X_0, Y_0$.

We now prove a useful lemma showing that $[Y_0]_{n+1} = y^{(n+1)}$ influences $X_i, Y_i$ in a very simple manner:

**Lemma 6.** *Let $X_i, Y_i$ follow the dynamics in (57). Then*

1. *$[X_i]$ is are independent of $[Y_0]_{n+1}$.*

2. *For $j \neq n+1$, $[Y_i]_j$ is independent of $[Y_0]_{n+1}$.*

3. *$[Y_i]_{n+1}$ depends additively on $[Y_0]_{n+1}$.*

*In other words, for $C := [0,0,0,\ldots,,0,c] \in \mathbb{R}^{1 \times (n+1)}$,*

$$1 : F_X\left(\{A_i, B_i\}_{i=0}^k, X_0, Y_0 + C\right) = F_X\left(\{A_i, B_i\}_{i=0}^k, X_0, Y_0\right)$$

$$2+3 : F_Y\left(\{A_i, B_i\}_{i=0}^k, X_0, Y_0 + C\right) = F_Y\left(\{A_i, B_i\}_{i=0}^k, X_0, Y_0\right) + C$$

*Proof of Lemma 6.* The first and second items follows directly from observing that the dynamics for $X_i$ and $Y_i$ in (57) do not involve $[Y_i]_{n+1}$, due to the effect of $M$.

The third item again uses the fact that $[Y_{i+1} - Y_i]_{n+1}$ does not depend on $[Y_i]_{n+1}$. $\qquad\square$

We are now ready to prove Lemma 5

*Proof of Lemma 5.* Let $Z_0$, $Z_k$, $\overline{Z}_0$, $\overline{Z}_k$ be as defined in the lemma statement. Let $\overline{X}_k$ and $\overline{Y}_k$ denote first $d$ rows and last row of $\overline{Z}_k$. Then by Lemma 6, $\overline{X}_{k+1} = X_{k+1}$ and $\overline{Y}_{k+1} = Y_{k+1} + \begin{bmatrix} 0 & 0 & \cdots & 0 & \langle w_\star, x^{(n+1)} \rangle \end{bmatrix}$. Therefore, (58) is equivalent to

$$\mathbb{E}_{(\overline{Z}_0, w_\star)}\left[\left([\overline{Z}_{k+1}]_{(d+1),(n+1)}\right)^2\right]$$

$$= \mathbb{E}_{(\overline{Z}_0, w_\star)}\left[\left([\overline{Y}_{k+1}]_{(n+1)}\right)^2\right]$$

$$= \mathbb{E}_{(\overline{Z}_0, w_\star)}\left[\left\|(I - M)\overline{Y}_{k+1}^\top\right\|^2\right]$$

$$= \mathbb{E}_{(\overline{Z}_0, w_\star)}\left[\text{Tr}\left((I - M)\overline{Y}_{k+1}^\top \overline{Y}_{k+1}(I - M)\right)\right].$$

This concludes the proof. $\qquad\square$

# D   Additional experimental results

In this section, we present a few addition experimental results. We first present in Figure 5 a visualization of learned weights $A_0, A_1, A_2$ for the setting of Theorem 4. One can see that the weight pattern matches the stationary point analyzed in Theorem 4; hence, combining Figure 4 and Figure 5, we corroborate our results from Theorem 4. Interestingly, it appears that the transformer implements a tiny gradient step using $X_0$ (as $A_0$ is small), and a large gradient step using $X_2$ (as $A_2$ is large). We believe that this is due to $X_2$ being better-conditioned than $X_1$, due to the effects of $B_0, B_1$.

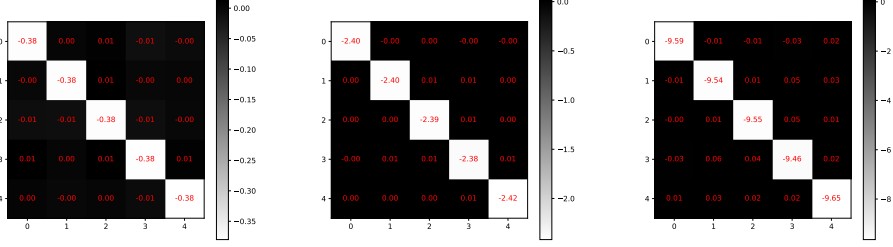

(a) Visualization of $\Sigma^{1/2} A_0 \Sigma^{1/2}$ (b) Visualization of $\Sigma^{1/2} A_1 \Sigma^{1/2}$ (c) Visualization of $\Sigma^{1/2} A_2 \Sigma^{1/2}$

Figure 5: Visualization of learned weights for the setting of Theorem 4. One can see that the weight pattern matches the stationary point analyzed in Theorem 4.

We next present some additional experiments that investigates the properties of the learned predictors of various algorithms. First, we plot the **test losses against the number of examples provided in the prompt** ("the number of ICL examples"). We compare four different algorithms: (i) the predictor learned by a three-layered of linear transformer, (ii) three steps of GD, (iii) three steps of preconditioned GD, and (iv) the ordinary least-squared solution (OLS). For GD and preconditioned GD, the optimal stepsizes are found by gridsearch. For preconditioned GD, preconditioner is fixed to be $\Sigma^{-1}$ for comparison. In all cases, the dimension $d = 5$, and for each $N$, the linear Transformer is trained using Adam. The result is presented in Figure 6.

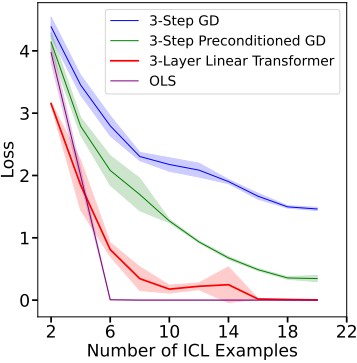

Figure 6: Test loss comparison between (i) the predictor learned by a three-layered of linear transformer, (ii) three steps of GD, (iii) three steps of preconditioned GD, and (iv) the ordinary least-squared solution (OLS).

Lastly, in Figure 7, we plot the **test losses against the number of layer** $L$ (or the number of steps in the case of gradient-based algorithms). For $L = 1, 2, 3, 4$, we compare between (i) the predictor learned by $L$-linear transformer and (i) $L$-steps of GD, (ii) $L$-steps of preconditioned GD. Again, the optimal stepsize is found by gridsearch, and for preconditioned GD, the preconditioner is fixed to be $\Sigma^{-1}$. In all cases, the dimension $d = 5$, and context length $N = 20$. The linear transformer is trained with Adam.

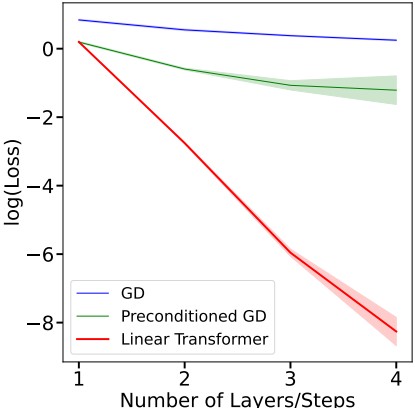

Figure 7: Test loss comparison between (i) the predictor learned by a $L$-layered linear transformer and (i) $L$-steps of GD, (ii) $L$-steps of preconditioned GD, for $L = 1, 2, 3, 4$.

