# Supplementary Materials

## A  Proofs for the single layer case (Theorem 1)

In this section, we prove our characterization of global minima for the single layer case (Theorem 1). We begin by simplifying the loss into a more concrete form.

### A.1  Rewriting the loss function

Recall the in-context loss $f(P, Q)$ defined in (6):

$$f(P,Q) = \mathbf{E}_{Z_0, w_\star} \left[ \left[ Z_0 + \frac{1}{n}\mathrm{Attn}_{P,Q}(Z_0) \right]_{(d+1),(n+1)} + w_\star^\top x^{(n+1)} \right]^2$$

Using the notation $Z_0 = [z^{(1)}\ z^{(2)}\ \cdots\ z^{(n+1)}]$, one can rewrite $Z_1$ as follows:

$$Z_1 = Z_0 + \frac{1}{n}\mathrm{Attn}_{P,Q}(Z_0)$$

$$= [z^{(1)}\ \cdots\ z^{(n+1)}] + \frac{1}{n}P[z^{(1)}\ \cdots\ z^{(n+1)}]M\left([z^{(1)}\ \cdots\ z^{(n+1)}]^\top Q[z^{(1)}\ \cdots\ z^{(n+1)}]\right).$$

Thus, the last token of $Z_1$ can be expressed as

$$z^{(n+1)} + \frac{1}{n}\sum_{i=1}^{n} P z^{(i)}(z^{(i)\top}Q z^{(n+1)}) = \begin{bmatrix} x^{(n+1)} \\ 0 \end{bmatrix} + \frac{1}{n}P\sum_{i=1}^{n} z^{(i)}z^{(i)\top}Q\begin{bmatrix} x^{(n+1)} \\ 0 \end{bmatrix},$$

where note that the summation is for $i = 1, 2, \ldots, n$ due to the mask matrix $M$. Letting $b^\top$ be the last row of $P$, and $A \in \mathbb{R}^{d+1,d}$ be the first $d$ columns of $Q$, then $f(P,Q)$ only depends on $b, A$ and henceforth, we will write $f(P,Q)$ as $f(b, A)$. Then, $f(b, A)$ can be rewritten as

$$f(b,A) = \mathbf{E}_{Z_0, w_\star} \left[ b^\top \underbrace{\frac{1}{n}\sum_{i} z^{(i)}z^{(i)\top}}_{} A x^{(n+1)} + w_\star^\top x^{(n+1)} \right]^2$$

$$=: \mathbf{E}_{Z_0, w_\star} \left[ b^\top \mathcal{M}A x^{(n+1)} + w_\star^\top x^{(n+1)} \right]^2 = \mathbf{E}_{Z_0, w_\star} \left[ (b^\top \mathcal{M}A + w_\star^\top) x^{(n+1)} \right]^2, \quad (13)$$

where we used the notation $\mathcal{M} := \frac{1}{n}\sum_{i} z^{(i)}z^{(i)\top}$ to simplify. We now analyze the global minima of this loss function.

To illustrate the proof idea clearly, we begin with the proof for the simpler case of isotropic data.

## A.2 Warm-up: proof for the isotropic data

As a warm-up, we first prove the result for the special case where $x^{(i)}$ is sampled from $\mathcal{N}(0, I_d)$ and $w_\star$ is sampled from $\mathcal{N}(0, I_d)$.

## Step 1: Decomposing the loss function into components

Writing $A = [a_1 \ a_1 \ \cdots \ a_d]$, and use the fact that $\mathbf{E}[x^{(n+1)}[i]x^{(n+1)}[j]] = 0$ for $i \neq j$, we get

$$f(b, A) = \sum_{j=1}^{d} \mathbf{E}_{Z_0, w_\star} \left[ b^\top \mathcal{M} a_j + w_\star[j] \right]^2 \mathbf{E}[x^{(n+1)}[j]^2] = \sum_{j=1}^{d} \mathbf{E}_{Z_0, w_\star} \left[ b^\top \mathcal{M} a_j + w_\star[j] \right]^2 .$$

Hence, we first focus on characterizing the global minima of each component in the summation separately. To that end, let us formally define each component in the summation as follows.

$$f_j(b, A) := \mathbf{E}_{Z_0, w_\star} \left[ b^\top \mathcal{M} a_j + w_\star[j] \right]^2 = \mathbf{E}_{Z_0, w_\star} \left[ \mathrm{Tr}(\mathcal{M} a_j b^\top) + w_\star[j] \right]^2$$
$$= \mathbf{E}_{Z_0, w_\star} \left[ \langle \mathcal{M}, b a_j^\top \rangle + w_\star[j] \right]^2 ,$$

where we use the notation $\langle X, Y \rangle := \mathrm{Tr}(XY^\top)$ for two matrices $X$ and $Y$ here and below.

## Step 2: Characterizing global minima of each component

To characterize the global minima of each objective, we prove the following result.

**Lemma 6.** *Suppose that $x^{(i)}$ is sampled from $\mathcal{N}(0, I_d)$ and $w_\star$ is sampled from $\mathcal{N}(0, I_d)$. Consider the following objective ($\langle X, Y \rangle := \mathrm{Tr}(XY^\top)$ for two matrices $X$ and $Y$)*

$$f_j(X) = \mathbf{E}_{Z_0, w_\star} \left[ \langle \mathcal{M}, X \rangle + w_\star[j] \right]^2 .$$

*Then a global minimum is given as*

$$X_j = -\frac{1}{\left( \frac{n-1}{n} + (d+2)\frac{1}{n} \right)} E_{d+1, j} ,$$

*where $E_{i_1, i_2}$ is the matrix whose $(i_1, i_2)$-th entry is 1, and the other entries are zero.*

**Proof of Lemma 6.** Note first that $f_j$ is convex in $X$. Hence, in order to show that a matrix $X_0$ is the global optimum of $f_j$, it suffices to show that the gradient vanishes at that point, in other words,

$$\nabla f_j(X_0) = 0 .$$

To verify this, let us compute the gradient of $f_j$:

$$\nabla f_j(X_0) = 2\mathbf{E} \left[ \langle \mathcal{M}, X_0 \rangle \mathcal{M} \right] + 2\mathbf{E} \left[ w_\star[j] \mathcal{M} \right] ,$$

where we recall that $\mathcal{M}$ is defined as

$$\mathcal{M} = \frac{1}{n} \sum_i \begin{bmatrix} x^{(i)} x^{(i)^\top} & y^{(i)} x^{(i)} \\ y^{(i)} x^{(i)^\top} & y^{(i)^2} \end{bmatrix} .$$

To verify that the gradient is equal to zero, let us first compute $\mathbf{E}\left[ w_\star[j]\mathcal{M} \right]$. For each $i = 1, \ldots, n$, note that $\mathbf{E}[w_\star[j]x^{(i)}x^{(i)^\top}] = O$ because $\mathbf{E}[w_\star] = 0$. Moreover, $\mathbf{E}[w_\star[j]y^{(i)^2}] = 0$ because $w_\star$ is symmetric, i.e., $w_\star \stackrel{d}{=} -w_\star$, and $y^{(i)} = \langle w_\star, x^{(i)} \rangle$. Lastly, for $k = 1, 2, \ldots, d$, we have

$$\mathbf{E}[w_\star[j]y^{(i)}x^{(i)}[k]] = \mathbf{E}[w_\star[j] \langle w_\star, x^{(i)} \rangle x^{(i)}[k]] = \mathbf{E}\left[ w_\star[j]^2 x^{(i)}[j]x^{(i)}[k] \right] = \mathbb{1}_{[j=k]} \quad (14)$$

because $\mathbf{E}[w_\star[i]w_\star[j]] = 0$ for $i \neq j$. Combining the above calculations, it follows that

$$\mathbf{E}\left[ w_\star[j]\mathcal{M} \right] = E_{d+1, j} + E_{j, d+1} . \tag{15}$$

We now compute compute $\mathbf{E}\left[ \langle \mathcal{M}, E_{d+1, j} \rangle \mathcal{M} \right]$. Note first that

$$\langle \mathcal{M}, E_{d+1, j} \rangle = \sum_i \langle w_\star, x^{(i)} \rangle x^{(i)}[j] .$$

Hence, it holds that

$$\mathbf{E}\left[\langle \mathcal{M}, E_{d+1,j}\rangle \left(\sum_i x^{(i)} x^{(i)\top}\right)\right] = \mathbf{E}\left[\left(\sum_i \langle w_\star, x^{(i)}\rangle x^{(i)}[j]\right)\left(\sum_i x^{(i)} x^{(i)\top}\right)\right] = O\,.$$

because $\mathbf{E}[w_\star] = 0$. Next, we have

$$\mathbf{E}\left[\langle \mathcal{M}, E_{d+1,j}\rangle \left(\sum_i y^{(i)2}\right)\right] = \mathbf{E}\left[\left(\sum_i \langle w_\star, x^{(i)}\rangle x^{(i)}[j]\right)\left(\sum_i y^{(i)2}\right)\right] = 0$$

because $w_\star \stackrel{d}{=} -w_\star$. Lastly, we compute

$$\mathbf{E}\left[\langle \mathcal{M}, E_{d+1,j}\rangle \left(\sum_i y^{(i)} x^{(i)\top}\right)\right]\,.$$

To that end, note that for $j \neq j'$,

$$\mathbf{E}\left[\langle w_\star, x^{(i)}\rangle x^{(i)}[j]\langle w_\star, x^{(i')}\rangle x^{(i')}[j']\right] = \begin{cases} \mathbf{E}[\langle x^{(i)}, x^{(i')}\rangle x^{(i)}[j] x^{(i')}[j']] = 0 & \text{if } i \neq i', \\ \mathbf{E}[\|x^{(i)}\|^2 x^{(i)}[j] x^{(i)}[j']] = 0 & \text{if } i = i', \end{cases}$$

and

$$\mathbf{E}\left[\langle w_\star, x^{(i)}\rangle x^{(i)}[j]\langle w_\star, x^{(i')}\rangle x^{(i')}[j]\right] = \begin{cases} \mathbf{E}[x^{(i)}[j]^2 x^{(i')}[j]^2] = 1 & \text{if } i \neq i', \\ \mathbf{E}\left[\langle w_\star, x^{(i)}\rangle^2 x^{(i)}[j]^2\right] = d + 2 & \text{if } i = i', \end{cases} \tag{16}$$

where the last case follows from the fact that the 4th moment of Gaussian is 3 and

$$\mathbf{E}\left[\langle w_\star, x^{(i)}\rangle^2 x^{(i)}[j]^2\right] = \mathbf{E}\left[\|x^{(i)}\|^2 x^{(i)}[j]^2\right] = 3 + d - 1 = d + 2.$$

Combining the above calculations together, we arrive at

$$\mathbf{E}\left[\langle \mathcal{M}, E_{d+1,j}\rangle \mathcal{M}\right] = \frac{1}{n^2} \cdot (n(n-1) + (d+2)n)(E_{d+1,j} + E_{j,d+1})$$

$$= \left(\frac{n-1}{n} + (d+2)\frac{1}{n}\right)(E_{d+1,j} + E_{j,d+1})\,. \tag{17}$$

Therefore, combining (15) and (17), the results follows. □

## Step 3: Combining global minima of each component

Now we finish the proof. From Lemma 6, it follows that

$$X_j = -\frac{1}{\left(\frac{n-1}{n} + (d+2)\frac{1}{n}\right)} E_{d+1,j}\,,$$

is the unique global minimum of $f_j$. Hence, $b$ and $A = [a_1\ a_1\ \cdots\ a_d]$ achieve the global minimum of $f(b, A) = \sum_{j=1}^{d} f_j(b, A_j)$ if they satisfy

$$b a_j^\top = -\frac{1}{\left(\frac{n-1}{n} + (d+2)\frac{1}{n}\right)} E_{d+1,j} \quad \text{for all } i = 1, 2, \ldots, d.$$

This can be achieve by the following choice:

$$b^\top = \mathbf{e}_{d+1}, \quad a_j = -\frac{1}{\left(\frac{n-1}{n} + (d+2)\frac{1}{n}\right)} \mathbf{e}_j \quad \text{for } i = 1, 2, \ldots, d\,,$$

where $\mathbf{e}_j$ is the $j$-th coordinate vector. This choice precisely corresponds to

$$b = \mathbf{e}_{d+1}, \quad A = -\frac{1}{\left(\frac{n-1}{n} + (d+2)\frac{1}{n}\right)} \begin{bmatrix} I_d \\ 0 \end{bmatrix}\,.$$

428 **Proof of uniqueness:** Suppose $X_1$ and $X_2$ are two minimizers of $f_j$, then $\langle \mathcal{M}, X_1 \rangle = \langle \mathcal{M}, X_2 \rangle$
429 almost surely for all $\mathcal{M}$. If $\langle \mathcal{M}, X_1 \rangle \neq \langle \mathcal{M}, X_2 \rangle$, then $f_j(\frac{1}{2} X_1 + \frac{1}{2} X_2) < \min f_j$ holds since the
430 1-dimensional quadratic function is strongly convex in its input. This concludes that the minimizer of
431 $f_j$ are a linear combination of $E_{j,d+1}$ with its transpose. Since the constraint $X = ba_j^\top$ ensures $X$ is
432 rank-one, then there are two possible solutions for $X$: $E_{j,d+1}$ or $E_{d+1,j}$. Given $b$ is shared among all
433 $f_j$, the only unique solution for $X$ is $E_{d+1,j}$. This ensures the uniqueness of solutions for $b$ and $a_j$
434 up to scaling.

435 We next move on to the non-isotropic case.

### A.3 Proof for the non-isotropic case

## Step 1: Diagonal covariance case

438 We first consider the case where $x^{(i)}$ is sampled from $\mathcal{N}(0, \Lambda)$ where $\Lambda = \mathrm{diag}(\lambda_1, \ldots, \lambda_d)$ and $w_\star$
439 is sampled from $\mathcal{N}(0, I_d)$. We prove the following generalization of Lemma 6.

440 **Lemma 8.** *Suppose that $x^{(i)}$ is sampled from $\mathcal{N}(0, \Lambda)$ where $\Lambda = \mathrm{diag}(\lambda_1, \ldots, \lambda_d)$ and $w_\star$ is*
441 *sampled from $\mathcal{N}(0, I_d)$. Consider the following objective*

$$f_j(X) = \mathbf{E}_{Z_0, w_\star} \left[ \langle \mathcal{M}, X \rangle + w_\star[j] \right]^2 .$$

442 *Then a global minimum is given as*

$$X_j = -\frac{1}{\frac{n+1}{n} \lambda_j + \frac{1}{n} \cdot \left( \sum_k \lambda_k \right)} E_{d+1,j} ,$$

443 *where $E_{i_1, i_2}$ is the matrix whose $(i_1, i_2)$-th entry is 1, and the other entries are zero.*

444 **Proof of Lemma 8.** Similarly to the proof of Lemma 6, it suffices to check that

$$2\mathbf{E} \left[ \langle \mathcal{M}, X_0 \rangle \mathcal{M} \right] + 2\mathbf{E} \left[ w_\star[j] \mathcal{M} \right] = 0 ,$$

445 where we recall that $\mathcal{M}$ is defined as

$$\mathcal{M} = \frac{1}{n} \sum_i \begin{bmatrix} x^{(i)} x^{(i)\top} & y^{(i)} x^{(i)} \\ y^{(i)} x^{(i)\top} & y^{(i)2} \end{bmatrix} .$$

446 A similar calculation as the proof of Lemma 6 yields

$$\mathbf{E} \left[ w_\star[j] \mathcal{M} \right] = \lambda_j (E_{d+1,j} + E_{j,d+1}). \tag{18}$$

447 Here the factor of $\lambda_j$ comes from the following generalization of (14):

$$\mathbf{E}[w_\star[j] y^{(i)} x^{(i)}[k]] = \mathbf{E}[w_\star[j] \left\langle w_\star, x^{(i)} \right\rangle x^{(i)}[k]] = \mathbf{E} \left[ w_\star[j]^2 x^{(i)}[j] x^{(i)}[k] \right] = \lambda_j \mathbb{1}_{[j=k]} .$$

448 Next, we compute $\mathbf{E} \left[ \langle \mathcal{M}, E_{d+1,j} \rangle \mathcal{M} \right]$. Again, we follow a similar calculation to the proof of
449 Lemma 6 except that this time we use the following generalization of (16):

$$\mathbf{E} \left[ \left\langle w_\star, x^{(i)} \right\rangle x^{(i)}[j] \left\langle w_\star, x^{(i')} \right\rangle x^{(i')}[j] \right] = \begin{cases} \mathbf{E}[x^{(i)}[j]^2 x^{(i')}[j]^2] = \lambda_j^2 & \text{if } i \neq i', \\ \mathbf{E} \left[ \left\langle w_\star, x^{(i)} \right\rangle^2 x^{(i)}[j]^2 \right] = \lambda_j \sum_k \lambda_k + 2\lambda_j^2 & \text{if } i = i', \end{cases}$$

450 where the last line follows since

$$\mathbf{E} \left[ \left\langle w_\star, x^{(i)} \right\rangle^2 x^{(i)}[j]^2 \right] = \mathbf{E} \left[ \left\| x^{(i)} \right\|^2 x^{(i)}[j]^2 \right] = \mathbf{

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

_jU_{\Sigma})\right] = G(X_0,A_j+t\mathbf{E}_U\left[U_{\Sigma}^{\top}R_jU_{\Sigma}\right])$.

## B.3 Proof of Theorem 5

The proof of Theorem 5 is similar to that of Theorem 4, and with a similar setup. However to keep the proof self-contained, we will restate the setup. Once again, we drop the factor of $\frac{1}{n}$ which was present in the original update (51). This is because the constant $1/n$ can be absorbed into $A_i$'s. Doing so does not change the theorem statement, but reduces notational clutter.

Let us consider the reformulation of the in-context loss $f$ presented in Lemma 9. Specifically, let $\overline{Z}_0$ be defined as

$$\overline{Z}_0 = \begin{bmatrix} x^{(1)} & x^{(2)} & \cdots & x^{(n)} & x^{(n+1)} \\ y^{(1)} & y^{(2)} & \cdots & y^{(n)} & y^{(n+1)} \end{bmatrix} \in \mathbb{R}^{(d+1)\times(n+1)},$$

where $y^{(n+1)} = \langle w_{\star}, x^{(n+1)}\rangle$. Let $\overline{Z}_i$ denote the output of the $(i-1)^{th}$ layer of the linear transformer (as defined in (51), initialized at $\overline{Z}_0$). For the rest of this proof, we will drop the bar, and simply denote $\bar{Z}_i$ by $Z_i$.[3] Let $X_i \in \mathbb{R}^{d\times n+1}$ denote the first $d$ rows of $Z_i$ and let $Y_i \in \mathbb{R}^{1\times n+1}$ denote the $(d+1)^{th}$ row of $Z_k$. Under the sparsity pattern enforced in (11), we verify that, for any $i \in \{0...k\}$,

$$X_{i+1} = X_i + B_iX_iMX_i^{\top}A_iX_i$$

$$Y_{i+1} = Y_i + Y_iMX_i^{\top}A_iX_i = Y_0\prod_{\ell=0}^{i}\left(I + MX_{\ell}^TA_{\