# OpenReview forum: "Transformers learn to implement preconditioned gradient descent for in-context learning"
_NeurIPS.cc/2023/Conference — NeurIPS 2023 poster_

### Official Review · Reviewer_vRHd · 2023-06-17

**Soundness:** 4 excellent
**Presentation:** 3 good
**Contribution:** 3 good
**Rating:** 7
**Confidence:** 4

**Summary:**

The paper tries to tackle an important and relevant problem in LLM training: whether in-context learning properties can be explained by a form of gradient descent?

For the setting, the authors have chosen to focus specifically on the linear transformer with a single or multiple layers with an input that is given by the data generated using a linear model. Within this setting, the authors were able to show a series of results for various specially constructed transformers.

Specifically, they look at a (1) single-layer transformer, (2) multi-layer with sparse parameters and (3) multi-layer with more general parameters. For (1), they have provided a characterization of the unique global optimum of a single-layer transformer with linear attentions. They showed that foer the optimal parameters found, the transformer implements a single step of preconditioned gradient descent. For (2), they have shown that global minimizers of the training objective of a two-layer linear transformer over isotropic regression instances. For the most general  scenario (3), they found that certain critical points implement preconditioning using inverse-data-covariance of the input.

Experimentally, the authors have confirmed using a toy-example that optimizing a three layer model indeed results in a minimizer that converges to an inverse of a covariance matrix.

**Strengths:**

The paper provides an interesting analysis that can be useful to understand the intriguing properties of in-context learning. It studies a series of transformer models, ranging from a simple single layer to a more general multi-layer ones and shows that for all the settings the in-context learning implements the preconditioned gradient descent. While by no means comprehensive, this paper can be useful as a stepping stone for any researcher that wants to further study intriguing properties of in-context learning.

The paper is easy to read and easy to understand. The assumptions and the math mainly seem to check out.

**Weaknesses:**

Generally, the paper restricts itself to a very narrow setting of a specific data that is centered at zero, linear model generating the labels, linear transformer, specific sparsity of the parameter matrix.

The transformers that they study are linear with a single head only. While I do appreciate the value of the analysis and I understand that this is just a step of many, it would be great to include a short paragraph outlining what the authors think about generalization of the proposed approach when the assumptions are removed.

Also, given the series of the assumptions, I would make sure that they are stated early on in the abstract and introduction.

I also think that the experimental validation can be slightly improved by considering different choice of layers, dimension of x, conditioning factor of the input etc. It would also be great to have an empirical plot demonstrating how close A is to the Gram matrix as the number of points increase (the results from the end of section 3)

**Questions:**

The equations for the gradient do not immediately pop out from the eq 7. I would encourage the authors to expose the equations for the gradient to the reader (maybe after eq 5) so that the connection between the gradient and the preconditioner would become more apparent.

What are the properties of A in sections 3 and 4? Can a transformer find a positive definite matrix through training?

For the evaluation of the Theorem 4 and 5, it is a bit hard to interpret the numbers, since the used distance metric is close to zero, but not exactly zero. The obtained value of ~0.1-0.2 is quite hard to interpret, apart from the fact that it is smaller that the distance with respect to the identity.

Also, the metric that the authors are using involves the minima over the space of scalars $\alpha$. It would be interesting if the authors can actually provide the scalars $a_i$ from Theorem 4 that they have found empirically. Do these scalars remain the same over multiple restarts of the algorithm?

How come the same $\Sigma$ matrix is used for both data and the weights (theorem 4 and 5)? It doesn't sound very realistic that the data and the weights use the same covariance matrix.

Would be great to add more information to Section 4.1 about adaptive coordinate-wise step sizes. If I understood correctly, Theorem 3 talks about the existence of the global minimizer under certain conditions. How exactly does adaptive coordinate-wise step sizes help find this solution?

I would encourage the authors to add a simple table at the end of the introduction clearly stating the assumptions that each section has. As written, It is quite hard to understand, especially when the notation change from section to section.

Small comments:
- Typo l.5: “Can transformers can”
- L.83: dimension of Z doesn’t match the dimension of W_k, and W_q in eq (1).
- In Lemma 2 z^(1) etc are mentioned, but not properly defined nor used.
- Gamma_k in l.155 plays a similar role to a_i from l.184. Maybe call both of these variables gamma?
- In fig.1 (d) the log(loss) converges to -1.0, which is not a zero value, as the text suggests in l.213.
- How come the authors choose L-BFGS as an optimizer? It is a fairly specialized algorithm that is not often used these days. What properties of L-BFGS are desirable in the settings that authors used it for? Can the same results be achieved with SGD or Adam?

**Limitations:**

It would be important to highlight limitations of the analysis. E.g. same covariance matrix, singe-head linear attention...

---

> ### Author Rebuttal · Authors · 2023-08-09
>
>
>
> **Regarding the setting** We agree with the reviewer that our setting is theoretical. However, we emphasize that despite our assumptions, the setting is expressive enough for ICL, and has been studied in a number of other papers.
> - [1,2,3] use linear attentions as linear attentions are expressive to implement gradient descent on linear regression.
> - [1-5] use centered data and generated labels
> - [1,2,3,5] focus on linear models
>
> In Section 5, we relax the sparsity constraint. As noted by reviewer `k8UL`: "It is quite a neat idea to consider sparsity driven constraints that permit tractability." Without the minimal sparsity constraint in Eq. 10, the analysis becomes very difficult.
>
>
> **Generalization of our results**
>
> - **Multi-head attention.** We show here that learning Multi-head _linear_ attentions reduces to learning a single head attention: Recall notation $Attn_{P,Q}(Z)$ in Eq. 2. Due to the linearity of the attention in $P$ and $Q$, we have
>  $\sum_{i} Attn_{P_i,Q_i}(Z) = Attn_{\sum_i P_i,\sum_{i}Q_i}(Z)$.  Thus the reparameterization $P' := \sum_{i} P_i$ and $Q' = \sum_{i} Q$ casts the problem to learning a single head attention.
> - **Linear attentions.** We elaborate on this concern in the [joint response](https://openreview.net/forum?id=LziniAXEI9&noteId=p362P0UrGI) and provide other references which also consider the same settings due to the expressively of linear attentions for linear models. Furthermore, we provide experimental and theoretical results for non-linear attentions.
>
> **Assumptions**
>
> In the abstract, we narrow our scope to "linear transformers trained over random instances of linear regression". Our introduction further clarifies "Akin to recent work, we too focus on the setting of linear regression encoded via ICL, for which we train a  transformer architecture with multiple layers of single-head self-attentions without softmax". Linear regression encoded via ICL is introduced in [1,4,5] which encompasses the settings of generated labels, linear regression, centered input data and specific encoding of regression samples. We will gladly add more required details.
>
> **Additional experiments**
>
> Thank you for your suggestions. We provide additional experiments in the PDF under [joint response](https://openreview.net/forum?id=LziniAXEI9&noteId=p362P0UrGI) above. In particular, in Figure 3, we plot "Distance of A to Identity" against "number of ICL samples".
>
> **Questions**
>
> - Regarding the scalars in Theorem 4, we have a theoretical result in Lemma 7 in the Appendix proving permutations of scalars obtains the same function value. Thus independent runs will not recover the same scalars.
>
> -  In experiments, we observe gradient descent converges to a positive definite matrix which depends on the covariance matrix of inputs (characterized in Theorems 4 and 5). We take steps towards proving this observation showing that the characterized parameters in Theorem 4 and 5 are stationary points of gradient descent. An interesting follow-up paper proves the convergence of gradients to the global optimum (in Theorem 1) for a single layer transformer [2]. (though the convergence for multi-layer transformer is still an open problem.)
>
> - We agree about the error ambiguity. In Figures 2 and 4, we visualize the outputs (after training) to illustrate the diagonal dominance of matrices which matches the result of Theorems 4 and 5. The error can be reduced by using a larger set of training set.
>
> -  In Theorem 1, we used different covariance matrices for data and $w^*$. However, the analysis for multi-layer attention is significantly more difficult when covariates are dependent to each other. We motivate this assumption in the practical scenario when "observations of covariates are distorted by $\Sigma^{1/2}$  (see line 190 of paper). The same covariance structure was also considered in a few other papers, e.g. [3].
>
> - Here, we do not analyze the convergence of adaptive coordinate-wise stepsizes to find the global optimum characterized in Theorem 3 (though it be an interesting future direction). Instead, we analyze the structure of a global minimizer of training objective and provide an algorithm interpretation for that.
>
> - We will improve readability: (i) we will add a table at the end of introduction and outline assumptions and main results of different sections (ii) add gradient formula.
>
> **Small comments:**
>
> - LBFGs enjoys considerably faster convergence rate compared to Adam in our settings. [1] has experiments with AdamW without sparsity constraints of Section 5. [2] proves the global convergence of gradient descent for a single-head attention. We will compare convergence rates of these algorithms to justify our choice for the optimizer.
>
> - In discussions, we included limitations of our analysis where we talked about linearity of attention. We will add more limitations such as covariance matrix structure, and single-head analysis.
>
> Thank you once again for taking the time to review our paper. We sincerely hope that the response to your concerns helps assuage your concerns, and view this paper in a more favorable light.
>
>  **References**
>
> [1] Von Oswald, Johannes, et al. “Transformers learn in-context by gradient descent.”
>
> [2] Zhang, Ruiqi, Spencer Frei, and Peter L. Bartlett. “Trained Transformers Learn Linear Models In-Context.”
>
> [3] Mahankali, Arvind, et al. “One Step of Gradient Descent is Provably the Optimal In-Context Learner with One Layer of Linear Self-Attention.”
>
> [4] Garg, Shivam, et al. "What can transformers learn in-context? a case study of simple function classes."
>
> [5] Akyürek, Ekin, et al. "What learning algorithm is in-context learning? investigations with linear models."
>
> [6] Imanol Schlag, et al. Linear transformers are secretly fast weight programmers.

---

> > ### Comment · Reviewer_vRHd · 2023-08-18
> >
> > I'm grateful for the authors for their detailed answer and commitment to modify the paper according to the feedback provided. Given that and the feedback from other reviewers, I would be happy to increase the score to get more aligned with the score of other reviewers.

---

### Official Review · Reviewer_iy8f · 2023-07-04

**Soundness:** 3 good
**Presentation:** 3 good
**Contribution:** 2 fair
**Rating:** 6
**Confidence:** 2

**Summary:**

This paper studies linear transformer and validates its ability in solving linear problems via applying conditioned gradient descent. The authors firstly state that there exists a unique global minimum for single attention layer. Then they explore multi-layer linear transformer and demonstrate that stationary points can be found that perform preconditioned gradient descent algorithm.

**Strengths:**

This paper theoretically studies that a $k$-layer linear transformer can perform as a $k$-step preconditioned gradient descent algorithm when forwarding a linear ICL sequence.

**Weaknesses:**

1. Related work is absent in this paper.
2. This paper simplifies the problem a lot by treating all components as linear. However, for a standard transformer, the softmax and activation functions are two key components and introduce nonlinearity. A discussion on the relationship between linear and standard transformers is missing.
3. Compared to the previous work (von Oswald et al., 2022), the novelty is insufficient except the introduction of non-isotropic data distribution.
4. Although simulations are provided in this paper, it is not sufficient. It would be interesting to know: How performance varies with the number of layers? How a $k$-layer linear transformer performs compared with a direct gradient descent approach with different step sizes or a standard transformer (during training or after sufficiently trained)? ...

**Questions:**

1. Although $w^\star$ is well-defined in the paper, it is not clear how $w$ relates to the model parameters or data.

minor:

2. Presuming you are studying least-squares loss, Eq.(5) may miss a square symbol.
3. There seem to be notation typos: $k\to i$ in Eq. (9) and $A_k\to A_i$ in Lemma 2.

**Limitations:**

While the authors did not explicitly discuss limitations, they provide potential future directions in the discussion section that are related to their work and have not been explored.

---

> ### Author Rebuttal · Authors · 2023-08-09
>
>
>  Thank you for your constructive comments! We address your comments one by one as below.
>
>  > Compared to the previous work (von Oswald et al., 2022), the novelty is insufficient except the introduction of non-isotropic data distribution.
>
> Our results in Theorems 1, 2, and 3 are novel even for isotropic inputs. Remarkably, [von Oswald et al. 2022] proves the recurrence of linear attentions is expressive enough to implement gradient descent. However, such expressivity result **do not imply anything about whether the trained transformers also attain such properties**. Here, we  theoretically address the conjecture of [von Oswald et al. 2022] which postulates that the optimization of parameters leads to the implementation of gradient descent.
>
> The importance of going beyond expressivity can be highlighted by considering the setting of anisotropic data distribution: the network is still expressive to implement gradient descent but the optimal transformer (with a single-layer) is not one that implements gradient descent. This is proved in Theorem 1. Thus, it is important to analyze the landscape of training to characterize the outcome of training.
>
>
> > This paper simplifies the problem a lot by treating all components as linear. However, for a standard transformer, the softmax and activation functions are two key components and introduce nonlinearity. A discussion on the relationship between linear and standard transformers is missing.
>
>
> Our problem settings is taken from [1] since we are building on empirical observations in [1]. More recently, [2] and [3] have also considered similar settings
>  - [MLP block with non-linear activations] We argue that MLP block are not necessary for learning linear models. Zeroing out the weights of MLP blocks allows skipping these blocks over residual connections. To implement gradient descent on ridge regression, one can omit MLP blocks and only rely on the attention.
>  - [Linear attentions] We elaborate on this concern in the [joint response](https://openreview.net/forum?id=LziniAXEI9&noteId=p362P0UrGI) and provide other references which also consider the same settings due to the expressively of linear attentions for linear models. Furthermore, we provide experimental and theoretical results for non-linear attentions.
>
>
>
> > Although simulations are provided in this paper, it is not sufficient. It would be interesting to know: How performance varies with the number of layers?
>
>
> To address your concern, we actually conducted more experiments and share them in the [joint response](https://openreview.net/forum?id=LziniAXEI9&noteId=p362P0UrGI) -- In Figure 2, we compare the linear transformer against Gradient Descent and Preconditioned Gradient Descent (with optimally chosen stepsize). We plot test loss against number of layers of linear transformer/steps of gradient descent.
>
> We would like to also highlight that there have already been several **empirical works** in the literature [1,4,5] showing various properties of the model learned for in-context learning. Hence, our main focus was to provide "**theoretical footing**" for their interesting empirical observations. In particular, we refer the reviewer to these works for more empirical investigations.
>
>
> > Although $w^*$ is well-defined in the paper, it is not clear how $w$ relates to the model parameters or data.
>
> $w^*$ is a random vector, independent of the data distribution. Since the model is trained over random instances, the parameters of optimal model is independent of individual random $w^*$ and only depends on their distribution. We established the connection between distribution of $w^*$ and optimal model in Theorem 1. Although $w^*$ does not show up physically in the weights _the transformer's predicted label is close to the predicted label using $w^*$_ (and as number of layers $\to \infty$, the transformer's prediction converges to $y^{(n+1)} = -<w^*, x^{(n+1)}>$)
>
>   > Related work is absent in this paper.
>
> We reviewed the related works in the introduction. To address your concern, we will add a broader scope of related works in the final version. Also, we will include any other related works that the reviewer recommend.
>  > Presuming you are studying least-squares loss, Eq.(5) may miss a square symbol.
> > There seem to be notation typos:
>  in Eq. (9) and
>  in Lemma 2.
>
>  Thank you very much for catching these typos and your comments.
>
>
>
>
> Thank you once again for taking the time to review our paper. We sincerely hope that the response to your concerns, as well as the overall response to other reviewers’ concerns, helps assuage your concerns, and view this paper in a more favorable light.
>
>  **References**
>
>  [1] Von Oswald, Johannes, et al. “Transformers learn in-context by gradient descent.” International Conference on Machine Learning. PMLR, 2023.
>
>
> [2] Zhang, Ruiqi, Spencer Frei, and Peter L. Bartlett. “Trained Transformers Learn Linear Models In-Context.” arXiv preprint arXiv:2306.09927 (2023)
>
>
> [3] Mahankali, Arvind, Tatsunori B. Hashimoto, and Tengyu Ma. “One Step of Gradient Descent is Provably the Optimal In-Context Learner with One Layer of Linear Self-Attention.” arXiv preprint arXiv:2307.03576 (2023)
>
>
> [4] Garg, Shivam, et al. "What can transformers learn in-context? a case study of simple function classes." Advances in Neural Information Processing Systems 35 (2022)
>
>
> [5] Akyürek, Ekin, et al. "What learning algorithm is in-context learning? investigations with linear models." arXiv preprint 2022.

---

> > ### Comment · Reviewer_iy8f · 2023-08-15
> >
> > Thank you to the authors for clarifying the contributions and providing additional experiments. Now the distinctiveness of this work compared to the prior studies is more evident to me. I have made the decision to increase my evaluation score.

---

### Official Review · Reviewer_k8UL · 2023-07-08

**Soundness:** 4 excellent
**Presentation:** 4 excellent
**Contribution:** 4 excellent
**Rating:** 7
**Confidence:** 4

**Summary:**

LLMs trained with transformers have shown remarkable abilities that have led to a lot of recent theoretical research towards understanding the origins of these abilities. In context learning is one such ability that refers to a model's ability to adapt in-context to new tasks without weight updates. In the work of Garg et al., the authors proposed a simple and powerful data generation process to understand the mechansims behind in-context learning. In Garg et al., it was empirically shown that transformers can learn to imitate ordinary least squares  algorithm. Since that work several works have taken steps to understand why and how transformers implement these algorithms. In recent work from Oswald et al., it was empirically shown that linear transformers implement one step of (modified) gradient descent per self-attention layer. In this work, the authors take an important step to explaining why do transformers implement gradient descent. The authors show that for shallow transformers pre-conditioned gradient descent is the global minima of the in-context training loss. For multi-layer transformers, the authors establish that pre-conditioned gradient descent form the critical points of the loss landscape.

**Strengths:**

This is a very good paper. I was very happy to read it as it solves an important unresolved problem that has to come to fore in the recent investigations on in-context learning.

On the different axes, I would rate the paper as follows.

Originality: I really liked the approach that the authors have taken. It is quite a neat idea to consider sparsity driven constraints that permit tractability.

Quality: This is a very good paper. The results resolve some very important questions towards understanding ICL.

Clarity: The paper is very well written and a joy to read.

Significance: . This paper serves as a crucial step towards the goal of understanding ICL is one of the central problems towards demistifying LLMs

**Weaknesses:**

I have some concerns that I would highly appreciate if authors can throw light on in the paper.

i) The authors have studied linear attention mechanisms, which I very well understand are themselves non-trivial to understand. If we were to consider softmax attention and carry out the same numerical experiments with sparsity constraints of the form of equation 8, then do we continue to achieve global minima. If authors can add such an experiment, then this can serve as very solid motivation for someone to study ICL with softmax attention (and sparsity constraint) and develop theory for the same.

ii) The authors reparametrize the query and key matrix as $Q= W_k^{\top} W_q$. This is not standard in transformers. What would happen if we were to consider the standard parametrization. Do we still obtain stationary points in the general case? Does the update dynamics still converge to the global minima (the verification step of Theorem 4 and 5)?

iii) The authors use LBFGS to learn the parameters. Do the results change under standard gradient descent or SGD or Adam?

iv) In the most general case of linear transformers, when $P$ and $Q$ are not constrained to be sparse at all, do the authors conjecture that similar results as the ones shown in the paper continue to hold.



**Questions:**

Please see the weakness section.

**Limitations:**

The authors discuss future directions, which stem from the limitations of the current study. The authors can elaborate further on limitations of the theory further and highlight some other potential gaps that can be addressed in future work.

---

> ### Author Rebuttal · Authors · 2023-08-09
>
>
>
> For the linear attention vs softmax attention comment, we provide an extended response in the [joint response](https://openreview.net/forum?id=LziniAXEI9&noteId=p362P0UrGI) above. Thank you for pointing it out!
>
> > What would happen if we were to consider the standard parametrization. Do we still obtain stationary points in the general case? Does the update dynamics still converge to the global minima (the verification step of Theorem 4 and 5)?
>
> There are experimental and theoretical results confirming the optimization of reparameterized network will recovers the same solution characterized in this paper: $W_k W_q^\top$ converges to $Q^*$ characterized in Theorems 1, 2 and 3 when $W_q$ and $W_k$ are square matrices. Experiments in [1] shows this convergence for single-layer and multi-layer linear attention. Theoretically, [2] recently proves the global convergence of GD to the optimal solution when  $W_q = W_k$ for a single attention layer.
>
> > The authors use LBFGS to learn the parameters. Do the results change under standard gradient descent or SGD or Adam?
>
> We conjecture that various algorithms will converge to the same solution characterized in the Theorems. This is proven for SGD on single layer attention by (Zhang et al. 2023) [2]. Furthermore, [1] includes experimental results for AdamW on multilayer attention without sparsity constraint. We specifically chose LBFGS because it can get to _very small loss_ much more quickly than Adam, which is important as the predicted sparsity pattern only becomes apparent when the loss is very low (see e.g. Figure 3). Based on preliminary experiments, we believe that, SGD and Adam will both recover the same solutions given enough iterations (and small enough learning rate).
>
> > In the most general case of linear transformers, when $P$ and $Q$ are not constrained to be sparse at all, do the authors conjecture that similar results as the ones shown in the paper continue to hold.
>
> [1] conjectures the similar results hold without sparsity constraint based on experimental observations. In fact, in our Section 5, we did our best to relax such technical sparsity assumption required for theoretical proofs.  We will clarify this in the paper.
>
>
>
> Thank you once again for taking the time to review our paper. We sincerely hope that the response to your concerns, as well as the overall response to other reviewers’ concerns, helps assuage your concerns, and view this paper in a more favorable light.
>
>
>
> **References**
>
> [1] Von Oswald, Johannes, et al. “Transformers learn in-context by gradient descent.”
>
> [2] Zhang, Ruiqi, Spencer Frei, and Peter L. Bartlett. “Trained Transformers Learn Linear Models In-Context.”

---

> > ### Comment · Reviewer_k8UL · 2023-08-16
> > **Thanks**
> >
> > I have read the response from the authors. The response has clarified some of the confusions I had. I would encourage authors to make changes to the paper in light of the above discussions. I am happy to maintain my score.

---

### Official Review · Reviewer_m6sU · 2023-07-10

**Soundness:** 4 excellent
**Presentation:** 3 good
**Contribution:** 4 excellent
**Rating:** 7
**Confidence:** 4

**Summary:**

The paper presents an in-depth analysis of in-context learning (ICL) in transformers, particularly focusing on linear transformers. The authors propose a novel interpretation of ICL as an implicit form of '**preconditioned**' gradient descent. They further extend this analysis to multi-layer transformers and provide insights into the optimal parameters for the in-context loss under certain constraints. The paper also includes empirical results on synthetic datasets to support their findings.

**Strengths:**

1. The paper provides a novel perspective -- the preconditioning/adaptive perspective -- on in-context learning in transformers, which could potentially lead to new ways of understanding and improving these models. Given the growing interest in understanding ICL from a theoretical perspective, the `*preconditioning gradient descent*' can be served as an important building block. This preconditioning perspective is also new in the literature (as far as I know).

2. The authors provide a rigorous mathematical analysis to support their claims, including single and multi-layer cases. The analysis tools developed in this paper could be helpful for future research on understanding ICL.

3. The paper is well-structured and includes visualizations to help understand the concepts and results.

**Weaknesses:**

1. (minor) In the experiments, I did not find the ICL experiments --- x-axis represents the number of ICL samples, and the y-axis represents the test loss (similar to the experiments in Figure 2 of Garg et al. (2022)). It would be interesting to check those results besides the current ones on training dynamics.

**Questions:**

1. In the current setup, I find there is no MLP block in the architecture, empirically that seems to be an important component in the transformer and ICL, could the authors provide more discussions/explanations on the roll of MLP (could also be listed as future directions)?
2. Typo: missing '.' at the end of line189.

---

> ### Author Rebuttal · Authors · 2023-08-09
>
> > I did not find the ICL experiments --- x-axis represents the number of ICL samples, and the y-axis represents the test loss (similar to the experiments in Figure 2 of Garg et al. (2022)). It would be interesting to check those results besides the current ones on training dynamics.
>
> Thank you for the suggestion. We conducted additional experiments on ICL, plotting test loss against number of ICL samples, for Linear Transformer vs Gradient Descent vs Preconditioned Gradient Descent. You can find it in Figure 1 of the rebuttal PDF in the [joint response](https://openreview.net/forum?id=LziniAXEI9&noteId=p362P0UrGI) above.
>
>
>
>
> As noted by Oswald et al., the network analyzed in this paper is sufficiently expressive to implement gradient descent on the least-squares objective. Hence, one can zero-out all parameters of the MLP block and still implement gradient descent on ridge regression. Thus, in this work, we omit MLP block; notably, this approach was also taken in other recent works in the literature [1,2,3]. Note that ICL of kernel regression requires MLP blocks as shown in [1,4,5]. Extending our results to kernel regression is an interesting research topic for future work. We will add a detailed discussion of this in our final version as per your suggestion.
>
> Thank you for catching the typos, we will correct them in the final version.
>
>
>
> Thank you once again for taking the time to review our paper. We sincerely hope that the response to your concerns, as well as the overall response to other reviewers’ concerns, helps assuage your concerns, and view this paper in a more favorable light.
>
> **References.**
>
> [1] Von Oswald, Johannes, et al. “Transformers learn in-context by gradient descent.” International Conference on Machine Learning. PMLR, 2023.
>
> [2] Zhang, Ruiqi, Spencer Frei, and Peter L. Bartlett. "Trained Transformers Learn Linear Models In-Context." arXiv preprint arXiv:2306.09927 (2023)
>
> [3] Mahankali, Arvind, Tatsunori B. Hashimoto, and Tengyu Ma. "One Step of Gradient Descent is Provably the Optimal In-Context Learner with One Layer of Linear Self-Attention." arXiv preprint arXiv:2307.03576 (2023)
>
> [4] Garg, Shivam, et al. "What can transformers learn in-context? a case study of simple function classes." Advances in Neural Information Processing Systems 35 (2022)
>
> [5] Akyürek, Ekin, et al. "What learning algorithm is in-context learning? investigations with linear models." arXiv preprint 2022.

---

> > ### Comment · Reviewer_m6sU · 2023-08-17
> >
> > Thanks to the authors for the response and additional experiments. My questions/concerns have been resolved.

---

### Author Rebuttal · Authors · 2023-08-09

We thank all the reviewers for very constructive comments.
- We thank **Reviewer m6sU** recognizing the novelity of our work:  _"The paper provides a novel perspective -- the preconditioning/adaptive perspective -- on in-context learning in transformers... Given the growing interest in understanding ICL from a theoretical perspective, the "preconditioning gradient descent" can be served as an important building block. The analysis tools developed in this paper could be helpful for future research on understanding ICL."_
- We thank **Reviewer k8UL** for appreciating the value of our approach: _"It is quite a neat idea to consider sparsity driven constraints....The results resolve some very important questions towards understanding ICL....This paper serves as a crucial step towards the goal of understanding ICL is one of the central problems towards demistifying LLMs"_.
- We thank **Reviewer vRHd** for appreciating the analysis in our paper:  _"The paper provides an interesting analysis that can be useful to understand the intriguing properties of in-context learning...this paper can be useful as a stepping stone for any researcher that wants to further study intriguing properties of in-context learning."_

We begin our response with the summary of our main contributions as summarized by the reviewers above.

**Primary technical achievements:**

- The inspiring previous works have shown that by setting the weights of transformers carefully, one can ICL the task of linear regression. We follow up on these works and show that in fact when you train your the transformer over the random instances of linear regression, **the global minima (and stationary points) correspond to interesting algorithms** (such as "preconditioned" gradient descent).
- At a more technical level, **we propose an analytical techniques to analyze the training loss of transformer architecture**.
- We would also like to underscore the relevance of this work to this community by briefly mentioning concurrent works that were posted shortly after our submission. First, Zhang et al. [2] have shown that under a random initialization, gradient descent converges to the global minimum characterized in this work, contributing to our understanding training transformers. Moreover, Arvind et al. [3] establish a similar set of results to ours for the single layer case.
We thus believe that the **question we attempt to answer in this work is timely and relevant to the community,** and would hopefully spark further interests.

We next address a shared concern raised by the reviewers:

**Beyond Linear Attention:**

As noted by reviewers, we focus on theoretical analysis of transformers with _linear attention and no MLP layers_. Nevertheless, such networks are expressive enough for solving linear regression, and they have become the focus of theoretical studies in various recent works [1,2,3]. In particular, Lemma 2 proves these networks are expressive enough to implement a family of optimization methods.

We observe empirically that softmax attention behaves similarly to the linear attention (consistent with observations in [Appendix A.9] in Oswald et al.) In fact, under the two-headed softmax attention, we empirically observe that the learned algorithm is similar to that of linear attention. Without going into details, the key intuition is the following linearization trick $\frac{1}{2} (e^x-e^{-x}) \approx x$. Indeed, in our experiments, we observe that the weights of the two attention heads have approximately opposite sign to each other.

Since theoretical analysis of softmax is rather difficult, we instead analyze a variant where softmax is replaced by ReLU as suggested by [4]. Experimental observation, presented in the attached PDF showing that the sparsity pattern of the solution is the same as those in Thm. 1. Here, we provide a theoretical analysis for this observation.

Following the theoretical settings of [1], we introduce a variant of non-linear attention where softmax is replaced by ReLU, i.e. $Attn_{Q}(Z) =  P ZM g(
Z^\top Q Z)$
where $g$ is the ReLU activation function, and $P = [0,\dots, 0, 1]^{\otimes 2} \in \mathbb{R}^{(d+1)\times (d+1)}$ and $Q = \begin{bmatrix} A & 0\\\\
0 & 0\end{bmatrix}$.
The in-contex loss is defined as
$$f(A) = \mathbb{E} \left( \sum_{i} \langle x^{(i)}, w^* \rangle    g(x^\top A x^{(i)} )- \langle x, w^* \rangle \right)^2 $$
where we use the compact notation $x = x^{(n+1)}$. We omit the constant $1/N$ for the simplicity of notations.

**Lemma.** Suppose that the data and $w^*$ distribution is Gaussian and isotropic. Then there is a constant $c$ such that $c I = \arg\min_A f(A)$.

This is a theoretical result derived in the limited rebuttal time, and hence has the following limitations (in addition to requiring further proof-checking):

- We fixed the parameter $P$ in the attention and only optimized $Q$.
- We only analyzed ReLU instead of softmax
- We only analyzed isotropic inputs.

The proof is based on an application of Stein's Lemma. [5] uses this technique to prove the solution of general linearized linear models for various non-linear functions and Gaussian inputs. Due to response limits, we could not include the proof. Upon request, we can add the proof in an official comment.

---
**References.**

[1] Von Oswald, Johannes, et al. "Transformers learn in-context by gradient descent."

[2] Zhang, Ruiqi, Spencer Frei, and Peter L. Bartlett. "Trained Transformers Learn Linear Models In-Context."

[3] Mahankali, Arvind, Tatsunori B. Hashimoto, and Tengyu Ma. "One Step of Gradient Descent is Provably the Optimal In-Context Learner with One Layer of Linear Self-Attention."

[4] Zhao, H., Panigrahi, A., Ge, R., & Arora, S.. Do Transformers Parse while Predicting the Masked Word?

[5] Erdogdu, M. A., Dicker, L. H., & Bayati, M. (2016). Scaled least squares estimator for GLMs in large-scale problems.

---

### Decision · Program_Chairs · 2023-09-21

**Decision:**

Accept (poster)

**Comment:**

The paper studies in-context learning, a phenomenon where Transformers can adjust their predictions based on additional data given in the input sequence itself. Prior work, e.g. von Oswald, showed that in-context learning in Transformers can approximate gradient-based learning within its forward pass.

The reviewers were in general rather positive about the paper, mentioning it provides an interesting perspective with a rigorous analysis. Some reviewers asked about the novelty compared to prior work, which in my view is adequately addressed in the rebuttal. For instance, the analysis of von Oswald is a constructive proof, stating a result about the potential expressivity of Transformers, but not what they learn. The paper therefore takes a significant step in the direction of showing that the optimization of parameters leads to the implementation of gradient descent.

Finally, the discussion between authors and reviewers has led several reviewers to increase their scores. Overall, the paper is clearly of interest to the community and I therefore recommend acceptance. I do encourage the authors to revise the paper according to the comments of the reviewers.